# Population-scale sequencing resolves determinants of persistent EBV DNA

Sherry S. Nyeo[1,2], Erin M. Cumming[1], Oliver S. Burren[3], Meghana S. Pagadala[1,4], Jacob C. Gutierrez[1], Thahmina A. Ali[1,2], Laura C. Kida[1], Yifan Chen[5,6], Hoyin Chu[1,2], Fengyuan Hu[3], Xueqing Zoe Zou[3], Benjamin Hollis[3], Margarete A. Fabre[3], Stewart MacArthur[3], Quanli Wang[3], Leif S. Ludwig[7,8], Kushal K. Dey[1], Slavé Petrovski[3✉], Ryan S. Dhindsa[6,9,10✉] & Caleb A. Lareau[1✉]

Epstein–Barr virus (EBV) is an endemic herpesvirus implicated in autoimmunity, cancer and neurological disorders. Although primary infection is often subclinical, persistent EBV infection can drive immune dysregulation and long-term complications. Despite the ubiquity of infection, the determinants of EBV persistence following primary exposure remain poorly understood, although human genetic variation partially contributes to this phenotypic spectrum[1–3]. Here we demonstrate that existing whole genome sequencing (WGS) data of human populations can be used to quantify persistent EBV DNA. Using WGS and health record data from the UK Biobank ($n = 490,560$) and All of Us ($n = 245,394$), we uncover reproducible associations between blood-derived EBV DNA quantifications and respiratory, autoimmune, neurological and cardiovascular diseases. We evaluate genetic determinants of persistent EBV DNA via genome association studies, revealing heritability enrichment in immune-associated regulatory regions and protein-altering variants in 148 genes. Single-cell and pathway level analyses of these loci implicate variable antigen processing as a primary determinant of EBV DNA persistence. Further, relevant gene programs were enriched in B cells and antigen-presenting cells, consistent with their roles in viral reservoir and clearance. Human leukocyte antigen genotyping and predicted viral epitope presentation affinities implicate major histocompatibility complex class II variation as a key modulator of EBV persistence. Together, our analyses demonstrate how re-analysis of human population-scale WGS data can elucidate the genetic architecture of viral DNA persistence, a framework generalizable to the broader human virome[4].

In 1964, Anthony Epstein, Yvonne Barr and Burt Achong observed actively replicating viral particles from Burkitt lymphoma, discovering the virus that now bears their names: the Epstein–Barr virus (EBV)[5]. EBV was subsequently recognized as the first known human oncogenic virus[6], the cause of infectious mononucleosis[6], and an agent in developing and exacerbating multiple autoimmune diseases[7]. Despite these wide-ranging pathogenic roles, EBV infection is nearly ubiquitous, infecting >90% of adults worldwide, with most individuals remaining asymptomatic[8]. EBV primarily transmits via saliva, infecting oral epithelial cells, spreading to B cells and establishing persistent infections in the human host that can last for a lifetime[9]. Why clinical outcomes of EBV infection—ranging from asymptomatic infection to severe disease—vary so widely remains poorly understood. The most severe manifestation, EBV-triggered cancers, collectively account for 130,000–200,000 annual deaths worldwide[10]. By contrast, immunocompetent individuals may harbour latent EBV within peripheral memory B cells, where the virus expresses a minimal gene program[4,11]. As with other herpesvirus infections, EBV can reactivate sporadically or in response to acute stressors or host immunosuppression, resulting in expanded viral reservoirs and potentially lethal clinical complications[11,12]. This vast phenotypic spectrum following acute and chronic infection underscores extensive individual variability, which can be partially attributed to host genetic variation[1–3]. However, genetic association studies of common infections with complex phenotypes such as EBV have been underpowered owing to small cohort sizes[13], motivating new approaches to study infection, viral persistence and host–phenotype associations.

[1]Computational and Systems Biology Program, Memorial Sloan Kettering Cancer Center, New York, NY, USA. [2]Tri-Institutional Program in Computational Biology, Weill Cornell School of Medicine, New York, NY, USA. [3]Centre for Genomics Research, Discovery Sciences, BioPharmaceuticals R&D, AstraZeneca, Cambridge, UK. [4]Department of Radiation Oncology, Memorial Sloan Kettering Cancer Center, New York, NY, USA. [5]Medical Scientist Training Program, Baylor College of Medicine, Houston, TX, USA. [6]Jan and Dan Duncan Neurological Research Institute, Texas Children's Hospital, Houston, TX, USA. [7]Berlin Institute of Health at Charité—Universitätsmedizin Berlin, Berlin, Germany. [8]Max-Delbrück-Center for Molecular Medicine in the Helmholtz Association (MDC), Berlin Institute for Medical Systems Biology (BIMSB), Berlin, Germany. [9]Department of Pathology and Immunology, Baylor College of Medicine, Houston, TX, USA. [10]Department of Molecular and Human Genetics, Baylor College of Medicine, Houston, TX, USA. ✉e-mail: slav.petrovski@astrazeneca.com; Ryan.Dhindsa@bcm.edu; lareauc@mskcc.org

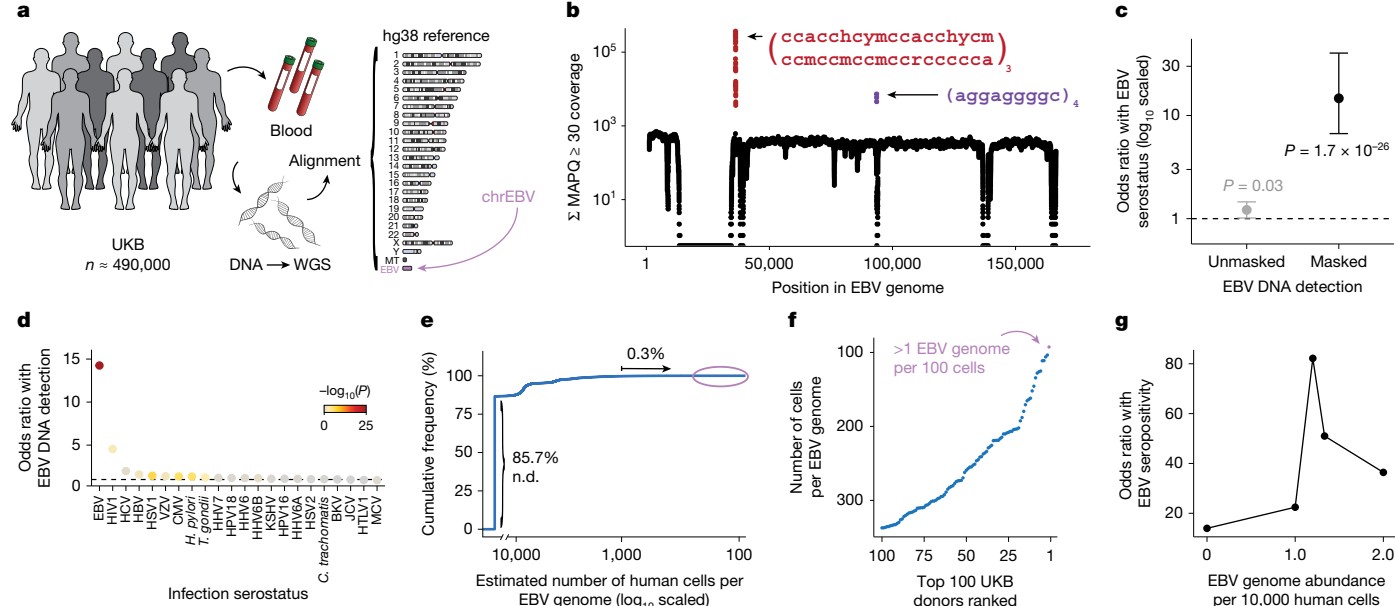

**Fig. 1 | Retrospective quantification of EBV DNA in the UKB. a**, Schematic of the approach. WGS libraries from peripheral blood were aligned to the hg38 reference genome, which contains an EBV reference contig (chrEBV). Reads mapping to chrEBV were extracted for downstream analyses. **b**, Sum of per-base read coverage of high-confidence EBV-mapping reads. Two repetitive regions with inflated coverage are noted in red and purple (following IUPAC convention: h = A/C/T, y = C/T, m = A/C, r = A/G; subscripts indicate the number of repeats). **c**, Association summary of individual-level serostatus and EBV DNA quantification with variable region masking. Statistical test: two-sided Fisher's exact test. Error bars represent 95% confidence intervals for the point effect

estimate (centre dot). **d**, Summary of EBV DNA detection with serostatus of 22 infectious agents. Statistical test: two-sided Fisher's exact test. HHV-6 was partitioned into strains HHV-6A and HHV-6B. **e**, Empirical cumulative distribution of detected EBV DNA across the entire cohort (85.7% of individuals had no detectable (n.d.) EBV DNA; 0.3% had EBV DNA at a copy number of 1+ EBV genome per 1,000 human cells). **f**, Top 100 individuals on the basis of EBV DNA copy number, from the circled population in **e**. **g**, Association between EBV seropositivity and EBV DNA detection thresholds at variable levels. Statistical test: two-sided Fisher's exact test. Sample size of full UKB cohort: $n$ = 490,560. The images in panel **a** were adapted from ref. 19, Springer Nature Ltd.

Beyond its role in human disease, EBV has been instrumental in advancing human population genetics research. EBV can transform primary B lymphocytes from healthy individuals into immortalized lymphoblastoid cell lines (LCLs)[14], critical resources that historically enabled long-term storage and large-scale genetic studies[15]. Consequently, immortalized LCLs were the primary material used in the HapMap[16] and 1000 Genomes[15] projects to profile genetic variation across the globe. These foundational efforts laid the groundwork for more expansive population-scale cohorts, such as the UK Biobank (UKB)[17] and All of Us (AOU)[18], which include sequencing and phenotypic data from hundreds of thousands of individuals: a scale that can interrogate the genetic underpinnings of complex phenotypes following infection.

As modern biobanks perform whole genome sequencing (WGS) on peripheral blood rather than on LCLs, we posited that EBV DNA reflecting EBV persistence in circulating cells could be captured and quantified in these libraries. Building on recent work that quantifies viral nucleic acids in petabyte-scale datasets to infer host–virus interactions retrospectively[19,20], we sought to develop a scalable computational pipeline to estimate individual-level EBV DNA loads. By leveraging the inclusion of the EBV genome as a contig in the human reference genome, we demonstrate how ordinarily excluded sequencing reads can be reanalysed to create a new molecular feature for genome-wide and phenome-wide association studies at petabase-scale.

## Biobank WGS data harbour EBV DNA

To address the high levels of EBV DNA present in the LCL-derived libraries—including those used in foundational efforts such as the 1000 Genomes Project[15]—the EBV genome (chrEBV, NC_007605) was incorporated into the human reference genome assembly (as of hg38)[21].

This alternative contig was designated as a sink for viral nucleic acids to improve variant calling and interpretation in the human genome[21]. We hypothesized that reads mapping to this contig from blood-derived WGS data would reflect persistence of EBV DNA following a primary infection. We thus extracted all sequencing reads from the aligned .cram files that mapped to chrEBV, enabling a quantification of the per-individual, per-base EBV DNA coverage across 490,560 individuals in the UKB (Fig. 1a,b). In addition to regions with low coverage corresponding to poor mappability, we identified two distinct loci with disproportionately high read depths, which corresponded to repetitive sequences (Fig. 1b and Methods). As these regions were covered at levels that were orders of magnitude higher than the median coverage of the viral contig, we reasoned that they would confound EBV DNA quantification. As an orthogonal measure of past infection, we used EBV serostatus ascertained on a subset of 9,687 individuals from the UKB, noting that EBV seropositivity requires sufficient antibody titres for at least two of four EBV antigens. We observed a nominal association between presence of EBV DNA and seropositivity when including these two repetitive regions (Fisher's exact test odds ratio = 1.2, $P$ = 0.03; Fig. 1c and Methods); however, discarding these two repetitive regions revealed that >40% of the UKB cohort only had aligned reads in these regions, and masking these regions before binarizing individuals resulted in a markedly stronger association (Fisher's exact test odds ratio = 14.6, $P$ = $1.7 \times 10^{-26}$; Fig. 1c). The next strongest association of detected EBV DNA with serostatus was for human immunodeficiency virus (HIV) 1 (Fisher's exact test odds ratio = 4.6, $P$ = 0.0023), consistent with reports of EBV DNA detection in blood following immunosuppression due to HIV[22] (Fig. 1d). Taken together, our sequencing-based approach readily scales to hundreds of thousands of individuals: a more than a 100-fold increase in sample size compared with serology-based association studies[13].

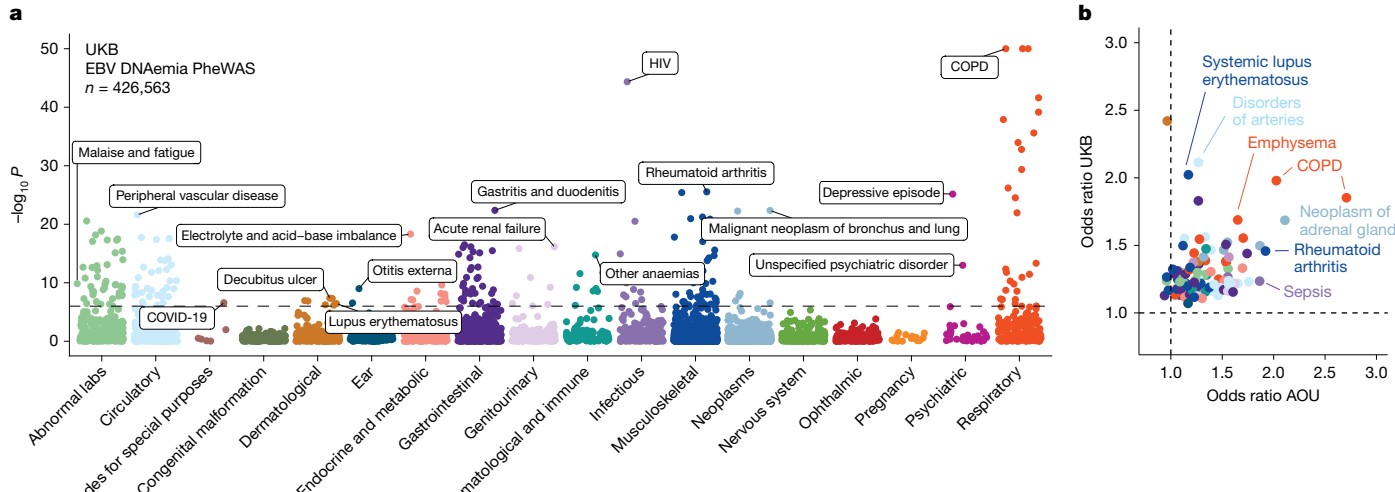

**Fig. 2 | EBV DNA is a biomarker of complex traits. a**, Summary of associations between EBV DNAemia and binary phenotypic traits in the UKB, with individuals of broadly NFE ancestry. The horizontal dashed line represents the phenome-wide significant *P*-value threshold ($3.3 \times 10^{-6}$). The *y* axis is capped at $-\log_{10}(P) = 50$, with those exceeding this threshold plotted at 50. Union phenotypes are plotted to reduce redundancy. Selected traits are highlighted based on biological interest. Statistical test: Wald test from logistic regression model (two-sided). **b**, Effect sizes for matching ICD-10 codes between the UKB and AOU, with individuals of European ancestry in AOU. Dotted lines at odds ratio = 1 represent null associations. Sample size: *n* = 426,563 UKB NFE individuals; *n* = 133,578 for AOU European ancestry individuals. The colours are the same as in **a**.

To further interpret our metric, we estimated the EBV DNA copy number per 1,000 cells by normalizing read counts between viral and host genome sizes. At the extremes of the distribution, 85.7% of individuals had no detectable bias-corrected EBV DNA, whereas 0.3% exhibited EBV DNA copy numbers of at least one viral genome per 1,000 human cells, including one individual with at least one EBV genome per 100 human cells (Fig. 1e,f). This range—which is derived from predominantly healthy individuals—is consistent with past quantitative polymerase chain reaction (qPCR)-based measurements of EBV DNA copy numbers in healthy populations, which reported upper ranges of one copy per 200 cells[23] (Methods). Using serostatus as a ground truth and accounting for standard covariates, a cutoff of 1.2 viral genomes per $10^4$ human cells yielded the strongest concordance with seropositivity (odds ratio = 82.2, $P = 2.2 \times 10^{-16}$), noting that all donors with detectable EBV DNA had at least one positive response against the four tested EBV antigens (Fig. 1g, Extended Data Fig. 1b and Methods). We classified 47,452 (9.7%) individuals with EBV DNAemia (defined as detectable EBV DNA levels >1.2 genomes per $10^4$ cells) for subsequent analyses (Extended Data Fig. 1c). As the proportion of individuals with EBV DNAemia (9.7%) is lower than the seropositivity rate (>90%) in the UKB, we interpret our metric as capturing the subset of individuals with the highest levels of circulating EBV DNA at the time of WGS sampling. Indeed, simulated data from a censored log-normal distribution of per-person EBV DNA levels closely approximated the empirical distribution (Extended Data Fig. 1d–g and Methods).

Next, we sought to better understand the profile of individuals with EBV DNAemia in the UKB cohort (Supplementary Table 1). Annotating each individual by birth location, we observed a higher proportion of EBV DNAemia in individuals born in more northern latitudes in the UK, consistent with previous reports of increased EBV infection further from the equator[24] (Extended Data Fig. 1h). We also observed a sex-biased (higher in male) and age-associated increase in EBV DNAemia rates, the latter consistent with EBV serology (Extended Data Fig. 1i). EBV DNAemia rates also differed among genetic ancestries and had a modest increase among individuals taking immunosuppressive medications (Extended Data Fig. 1j,k and Supplementary Table 2). We performed parallel analyses in the AOU cohort, spanning 245,394 individuals with blood-derived WGS (Extended Data Fig. 2a and Methods). Results from the independent analyses of AOU replicated key attributes of the UKB data, including a clear repetitive region that was similarly masked, yielding 11.9% of individuals with EBV DNAemia and consistent associations with age, sex, genetic ancestry and prescription of immunosuppressive drugs (Extended Data Fig. 2b–f).

As primary EBV infection occurs earlier in life[25], we hypothesized that donors with EBV DNAemia probably reflected a previous infection that persisted until sampling. Conversely, lytic herpesvirus infection would be concomitant with viral transcription, including in peripheral blood[11,19]. As the UKB and AOU collected DNA but not RNA-seq data, we reprocessed bulk and single-cell RNA-seq from the OneK1K[26] and Genotype-Tissue Expression[27] consortia to assess for EBV transcription in peripheral blood cells (Supplementary Note 1 and Supplementary Fig. 1a,f). Across these 1,663 donors, we detected minimal evidence of EBV transcripts, suggesting that the vast majority of blood-derived EBV DNA from our cohorts probably reflects latent infection, which is concordant with the lack of EBV lytic reactivation gene expression detected in peripheral B cells of healthy individuals[28] (Supplementary Note 1 and Supplementary Fig. 1c,g). Furthermore, analysed saliva-derived WGS samples for another set of 48,899 AOU participants showed a markedly higher rate of EBV DNAemia (50.9%), reflecting a distinct environmental and cellular reservoir for EBV (Supplementary Note 2 and Supplementary Fig. 2a–d). Together, our findings demonstrate that EBV DNA can be retrospectively quantified from existing large-scale WGS datasets with reproducible signals, including sequences collected from different anatomical sites.

## Associations with complex traits

Next, we investigated whether our WGS-enabled measure of EBV DNAemia could serve as a biomarker of complex disease. To assess this, we performed a phenome-wide association study (PheWAS) to map systematic outcomes catalogued via International Classification of Diseases, 10th revision (ICD-10) codes with EBV DNAemia as an exposure (Methods). Using individuals from the UKB of predominantly non-Finnish European (NFE) genetic ancestry (*n* = 426,563) as a discovery cohort, we tested for the association between EBV DNAemia and 13,290 binary phenotypes as well as 1,931 quantitative phenotypes, following our previously described PheWAS workflow[29] (Supplementary Table 3 and Methods). Among binary traits, we observed

271 significant ($P < 3.3 \times 10^{-6}$) ICD-10 codes, including well-established associations with splenic diseases and Hodgkin lymphoma. We also observed significant associations with rheumatoid arthritis[11], chronic obstructive pulmonary disease (COPD[30]) and systemic lupus erythematosus[28], each of which has been previously associated with EBV using orthogonal approaches (Fig. 2a). Past case studies have anecdotally reported associations between EBV infection and various conditions in small-scale studies relative to our population-scale cohorts. Our analyses reinforced evidence for these relationships, including chronic ischemic heart disease (odds ratio = 1.19, $P = 2.8 \times 10^{-18}$), acute kidney failure (odds ratio = 1.21, $P = 1.4 \times 10^{-16}$), depressive episodes (odds ratio = 1.19, $P = 4.0 \times 10^{-26}$) and stroke (odds ratio = 1.20, $P = 6.1 \times 10^{-13}$). We emphasize that these associations may also reflect a general state of immunosuppression, and additional work is required to determine which of these associations are causal rather than correlational.

Statistically significant quantitative associations ($n = 156$) included leukocyte count, neutrophil percentage, smoking pack years, telomere length and compositions of omega-3 fatty acids, consistent with previous observations of lipogenesis induction following EBV infection[31] (Extended Data Fig. 3a). We also detected an association with malaise and fatigue (odds ratio = 1.27, $P = 2.06 \times 10^{-10}$), noting that EBV has long been hypothesized as a risk factor for myalgic encephalomyelitis/chronic fatigue syndrome (ME/CFS)[32]. We also identified significant associations with decreased levels of phosphatidylcholine ($P = 2.9 \times 10^{-9}$) and total choline ($P = 5.9 \times 10^{-9}$), consistent with metabolic studies in patients with ME/CFS[33]. Our results reinforce a potential relationship between EBV and ME/CFS that warrants further examination.

We sought to replicate these associations using the AOU cohort (Fig. 2b). As the underlying electronic health record data vary between cohorts, we focused on 141 significantly associated ICD-10 codes in the UKB that had sufficient representation in AOU (minimum $n = 24$ cases). Of these, 87 (62%) were replicated in AOU ($P < 0.05$; odds ratio directionally concordant with UKB statistics), resulting in a set of traits that we examined more closely (Methods, Supplementary Table 3 and Supplementary Note 3). These phenotypes included rheumatoid arthritis, COPD and lung neoplasms, as well as less-established phenotypes such as peripheral vascular disease, emphysema and tachycardia, some of which may be attributable to the association between smoking and EBV reactivation. We also considered two traits that were previously linked to EBV but were not significant in either cohort (Methods and Extended Data Fig. 3b). For multiple sclerosis, we observed nominal associations that did not survive multiple testing corrections (UKB, odds ratio = 2.1, $P = 0.019$; AOU, odds ratio = 0.73, $P = 0.0087$), consistent with a past report that did not detect a significant association using ICD-based viral exposure measures[34]. For gammaherpesviral mononucleosis, a primary manifestation of EBV infection, the association was in the expected direction (UKB, odds ratio = 2.55, $P = 0.23$; AOU, odds ratio = 5.86, $P = 1.1 \times 10^{-6}$) but underpowered owing to low sample sizes ($n = 11$ in the UKB, $n = 42$ in AOU), noting infectious mononucleosis primarily affects younger individuals.

In addition to phenotypes that replicated between cohorts, we noted instances of neurological conditions that were nominally associated with EBV DNAemia in the UKB but lacked sufficient case numbers to be assessed in AOU ($P < 0.05$; Extended Data Fig. 3c). These included all-cause dementia (odds ratio = 1.16, $P = 6.0 \times 10^{-5}$); rarer phenotypes such as neuromyelitis optica, which is a rare autoimmune disease with similar clinical presentation as multiple sclerosis (odds ratio = 6.31, $P = 2.7 \times 10^{-3}$); and acute disseminated demyelination (odds ratio = 6.31, $P = 5.3 \times 10^{-3}$). Although further work is required to implicate the role of EBV in these phenotypes, our scalable approach enables systematic association studies across a broad range of conditions, including rare diseases for which very large cohorts such as the UKB and AOU are essential.

## Genetic variation underlies EBV DNAemia

Past studies have established that manifestations of viral infections are a polygenic trait controlled by dozens of loci in the human genome[13,35]. Hence, we reasoned that genetic variation would similarly influence the variable degree of EBV persistence across the population. We thus conducted a genome-wide association study (GWAS) on individuals of NFE ancestry (~94% of the UKB cohort) to identify loci associated with EBV DNAemia (Methods). Using array-based genotype data followed by imputation from 426,563 NFE individuals in the UKB, we identified 22 independent loci associated with EBV DNAemia that reached genome-wide significance ($P < 5 \times 10^{-8}$; Fig. 3a, Methods and Supplementary Table 4). Overall, the single nucleotide polymorphism (SNP)-based heritability ($h^2$) determined by LDscore regression (LDSC) was 2.21% (± 0.85%) with limited evidence of genomic inflation ($\lambda_{GC} = 1.1$; LDSC intercept = 1.03 ± 0.008; Supplementary Note 4). Partitioned heritability analyses showed an enrichment at conserved and non-coding loci marked by enhancer and super-enhancer annotations (Extended Data Fig. 4a), consistent with other complex trait associations[36].

The strongest associations emerged near human leukocyte antigen (HLA) genes on chromosome 6 that encode the major histocompatibility complex (MHC) class I and II proteins (Fig. 3a). Major histocompatibility complex molecules are critical in differentiating between self and non-self proteins and have been widely associated with autoimmune traits[13,37]. We conducted an exome-wide association study (ExWAS), which included protein-coding variants observed at least six times (that is, with a minor allele count of greater than five) in the NFE cohort[29], to refine association signals at the MHC and other associated loci (Methods). Associations at alleles assayed by either technology were concordant (Extended Data Fig. 4b). Among the 1,102 variants significantly associated with EBV DNAemia ($P < 5 \times 10^{-8}$, cases > 20), 686 were missense variants spanning 148 genes. These missense variants facilitated the annotation of putative causal variants at 9 of the 22 implicated loci (Fig. 3a, Methods and Supplementary Table 4). Consistent with our GWAS results, the protein-coding variants with the largest effect sizes were near the MHC locus, where 148 MHC class I, 113 MHC class II and 7 non-classical HLA protein-altering variants were significantly associated with EBV DNAemia (Fig. 3b).

We used the AOU cohort to replicate the biological plausibility and pleiotropy of genetic associations in the UKB. Repeating our GWAS framework on $n = 131,938$ people with European (EUR) ancestry in AOU for 12,099,305 common variants (1% minor allele frequency), we observed concordant associations at implicated loci. Globally, 40,675 variants were genome-wide significant ($P < 5 \times 10^{-8}$) in the UKB and passed quality control filters in AOU, noting that many were from the HLA region. Of these genome-wide significant variants, 91.4% of variants were replicated in the AOU GWAS (nominal $P < 0.05$; odds ratio concordant; Fig. 3c). Further, 12 of the 19 (63%) assayed index GWAS variants replicated in the AOU GWAS (nominal $P < 0.05$, odds ratio directionally concordant; Supplementary Table 4). These included loci near well-established immune-regulatory genes, including *CTLA4, EOMES, LNPEP, PTPN22* and *SLAMF7* (Supplementary Note 4 and Supplementary Fig. 3c–f). Although these analyses primarily focused on individuals of European ancestry, additional meta-analyses from the diverse ancestries of the UKB and AOU revealed an additional 23 loci surpassing genome-wide significance, including variants near *BIM, GSDMB, TERT, BCL11A, MYC* and *CD160* (Supplementary Note 5 and Supplementary Fig. 4a,b). Together, our results indicate that persistence of EBV DNA is a polygenic trait that can be quantified from multiple population-scale WGS datasets, and loci underlying EBV DNAemia are reproducible across continents.

Given the well-described associations between EBV and immune-mediated phenotypes, we sought to systematically evaluate similarities between the genomic architectures of EBV DNAemia and immune-mediated diseases (IMDs). We used cupcake[38], a framework

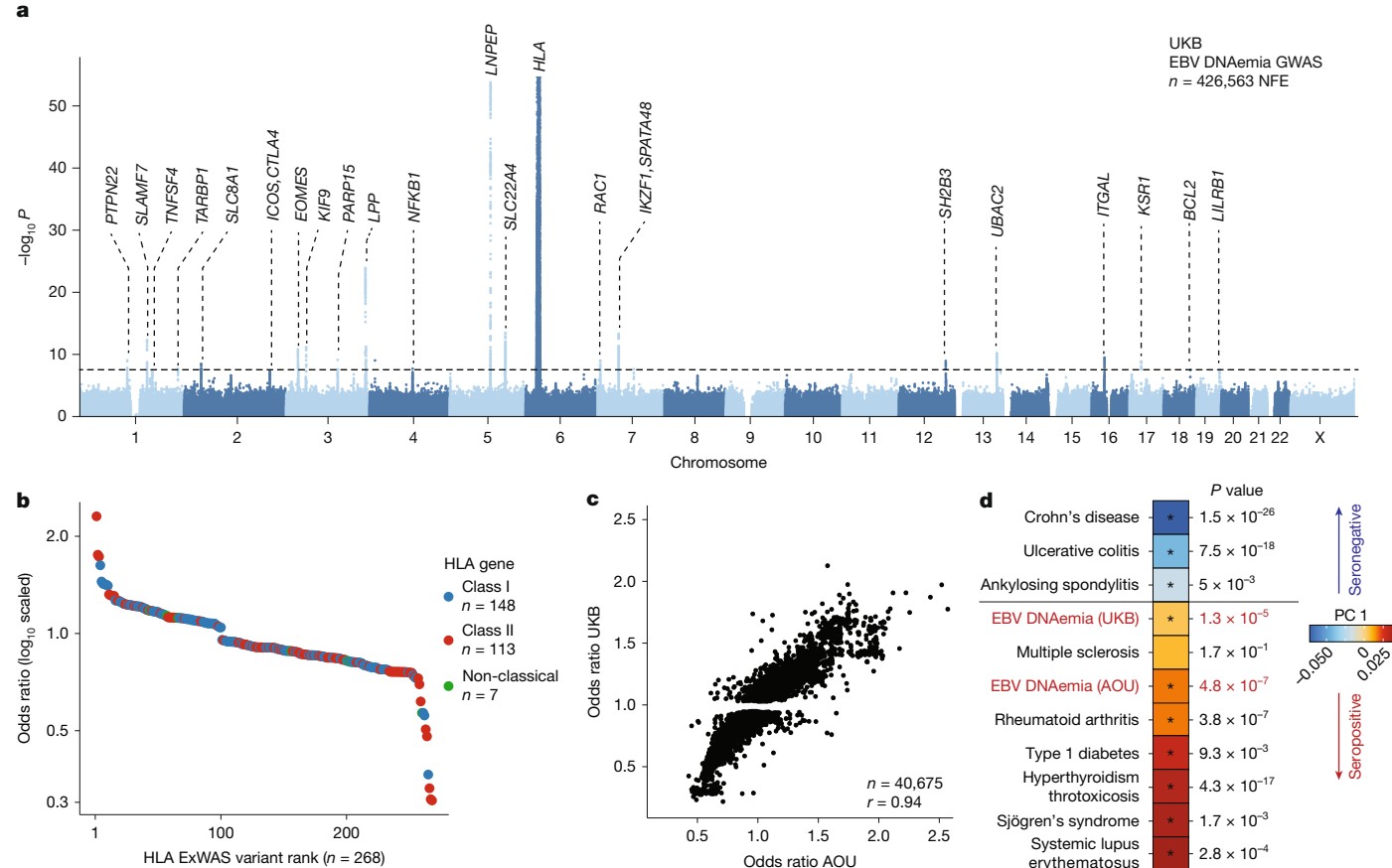

**Fig. 3 | Genetic architecture of EBV DNAemia. a**, Manhattan plot summarizing the genome-wide association statistics for EBV DNAemia for 426,563 individuals of predominantly NFE ancestry in the UKB. Genes proximal to genome-wide significant associations ($P < 5 \times 10^{-8}$) are annotated. Statistical test: likelihood ratio test from logistic regression model (two-sided). **b**, Summary of protein-altering variants in HLA genes. **c**, Replication of UKB-associated variants in the AOU European ancestry cohort. The Pearson correlation coefficient of variant effect sizes is noted. **d**, PCA and projection of EBV summary statistics on complex immune-mediated diseases via cupcake[38]. An asterisk indicates a significant PC projection score after multiple testing correction. Statistical test: $Z$-test (two-sided).

that accounts for the shared components of non-HLA genetic architecture across 13 IMDs using a shrinkage approach to adjust for linkage disequilibrium, allele frequency and differential sample size via principal component analysis (PCA), where principal component (PC)1 captures an IMD genetic axis characterized by autoantibody seropositivity[38]. Consistent with our PheWAS and past reports of EBV pathogenesis, we observed a cupcake PC1 score that reflects a shared component of genetic architecture between EBV DNAemia and autoimmune diseases such as rheumatoid arthritis, systemic lupus erythematosus and T1D from both the UKB ($P = 1.3 \times 10^{-5}$) and AOU ($P = 4.8 \times 10^{-7}$; Fig. 3d). Noting that initial EBV infections are most prevalent in adolescence[25] and generally precede onset of autoimmunity[39], our data refine a potential model in which a component of genetic architecture shared by seropositive IMDs may first determine the persistence of EBV after primary infection that, in turn, may trigger complications characteristic of disease.

As a contrast to our blood-derived EBV DNAemia biomarker, we conducted analogous genome-wide analyses of binarized EBV serology (seropositivity) from the UKB[40] and saliva-derived EBV DNAemia from AOU[18]. Serostatus from 8,669 individuals of NFE ancestry resulted in zero genome-wide significant loci (Supplementary Note 5 and Supplementary Fig. 4c). Furthermore, although we observed markedly higher levels of EBV DNA in AOU saliva WGS samples, including 51% DNAemia in 32,745 saliva EUR ancestry donors, the only genome-wide significant association for these individuals under three different candidate models was at the MHC locus (Supplementary Note 2 and Supplementary Fig. 2f,g). We attribute the disparity in the number of significant loci

to the underlying biology of EBV DNAemia in peripheral blood, distinct from the site of transmission[11]. Whereas EBV serostatus reflects a history of any past infection, which is largely independent of genetic variation, EBV DNAemia identifies the subset of infected individuals with the highest levels of persistent viral DNA.

## Cell type and pathway level analyses

To further evaluate the role of EBV DNAemia-associated immunomodulatory genes, we examined the expression of the 148 genes that harboured at least one significant ExWAS variant as a signature score in a multi-modal dataset of 211,000 human peripheral blood mononuclear cells (PBMCs)[41] (Fig. 4a). As expected, the EBV signature score was enriched in B cells, consistent with the known viral tropism of EBV infection and latency[8,9] (Fig. 4b,c). This enrichment was corroborated in the non-coding genome, as genome-wide significant variants were enriched in B cell-specific accessible chromatin from fluorescent-activated cell-sorting-isolated populations profiled via the Assay for Transposase Accessible Chromatin using sequencing[42] (ATAC-seq; Extended Data Fig. 5a and Methods). We also observed a similar enrichment in subsets of antigen-presenting cells, particularly conventional dendritic cells (Extended Data Fig. 5b,c), although dendritic cells are most likely not directly infected by EBV[43]. To resolve the potential biological processes linked to this genetic architecture, we performed gene set analyses using the Gene Ontology biological processes and Kyoto Encyclopedia of Genes and Genomes (KEGG) pathway

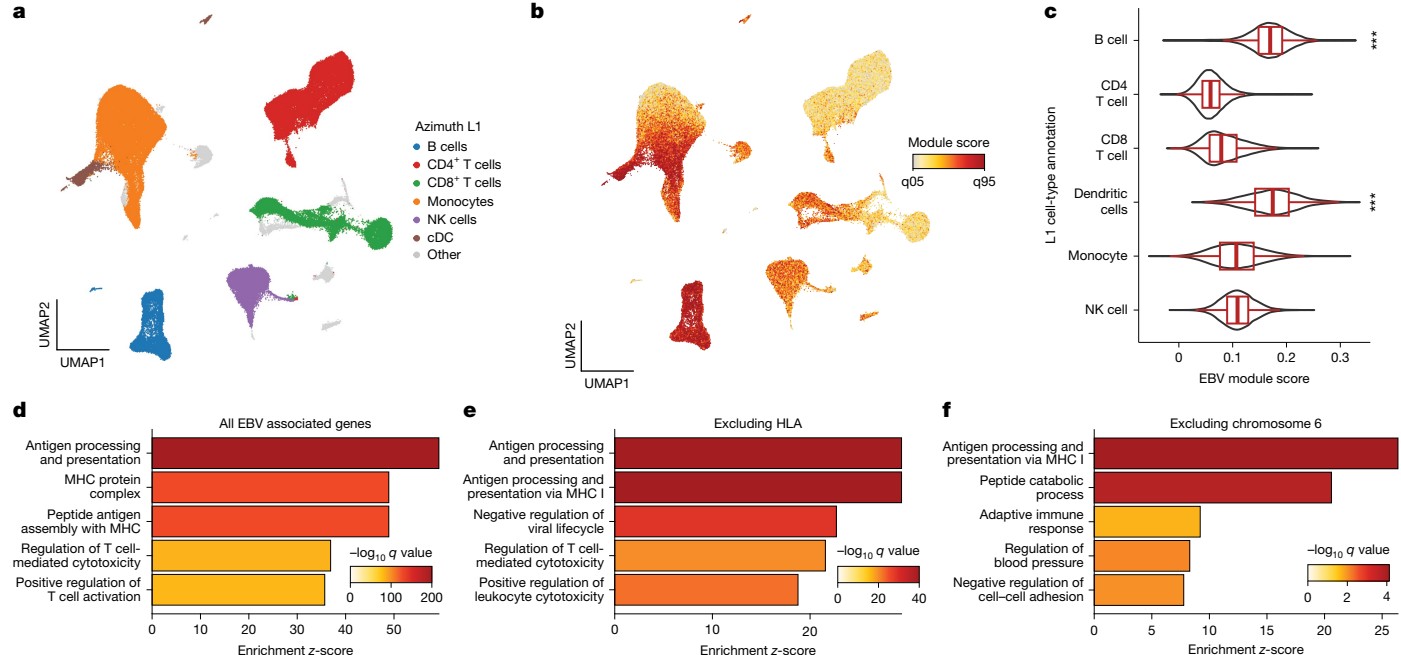

**Fig. 4 | EBV DNAemia gene associations at cell and pathway resolutions.**
**a**, Uniform manifold approximation and projection (UMAP) embedding of 211,000 PBMCs. The broad cell type label (Azimuth L1; ref. 41) annotates major populations. NK, natural killer; cDC, conventional dendritic cells. **b**, Module score of EBV ExWAS associations, highlighting populations with the highest enrichment. **c**, Summary of EBV ExWAS scores in major populations. ***Indicates statistical significance ($P < 2.2 \times 10^{-16}$ relative to held-out cell types; $P$-value threshold reflects machine precision; one-sided Wilcoxon rank-sum test; $n = 211,000$ cells). Boxplots: centre line, median; box limits, first and third quartiles; whiskers, 1.5× interquartile range. **d**, Summary of the top five enriched Gene Ontology biological process terms identified by gene set enrichment analysis of EBV DNAemia-associated genes. **e**, Same as **d** except excluding annotated HLA genes. **f**, Same as **d** but excluding all genes mapping to chromosome 6.

analyses. Among the Gene Ontology biological-process-enriched terms, the top pathways involved antigen processing and presentation, MHC protein complex and assembly, and regulation of T cells (Fig. 4d, Extended Data Fig. 5d and Supplementary Table 5). From the KEGG enrichments, we observed disease-associated annotations that included viral myocarditis, rheumatoid arthritis, herpes simplex virus 1 (HSV-1) infection and, reassuringly, EBV infection (Extended Data Fig. 5e). As the strong linkage disequilibrium on chromosome 6 could drive this association, we refined these enrichments by further removing all HLA-associated genes or all genes on chromosome 6 (Methods). Regardless, antigen processing and presentation remained the most enriched term in our Gene Ontology biological processes analyses, underscoring the critical role of this pathway in controlling viral infection and clearance (Fig. 4e,f). Together, these analyses indicate that B cells and antigen-presenting cells are the primary cell types affected by the genetic architecture of EBV DNAemia, with viral antigen processing and presentation predominantly influencing the emergence and persistence of EBV DNA, a characterization consistent with the known roles of these immune cells in regulating herpesvirus infections.

## HLA-EBV peptide binding predictions

Although the HLA locus is pervasively associated with immune-mediated complex traits, these associations are challenging to resolve owing to allelic diversity, heterogeneity between human populations and lack of well-estimated (auto-) antigens that can mediate complex trait manifestation[37]. In our setting, the EBV proteome defines the set of candidate antigens variably presented by these alleles that would, in turn, variably yield EBV DNAemia. Hence, we reasoned that explicit modelling of HLA variation could refine our understanding of genetic variation underlying viral persistence.

To assess this, we first assembled four-digit HLA alleles across all donors in the UKB and AOU with NFE or EUR ancestry (Extended Data

Fig. 6a and Methods). Using these per-donor genotypes and similar covariates to our GWAS, we performed a multivariate regression to assess whether each HLA allele was associated with variable rates of EBV DNAemia (Methods). We identified a total of 42 associated HLA alleles, including 18 class I and 24 class II alleles (nominal $P < 0.05$ in both cohorts; Extended Data Fig. 6b,c, Supplementary Table 6 and Methods). One of the strongest risk alleles for EBV DNAemia was HLA-A*03:01 (UKB, $P = 0.0060$; AOU, $P = 9.63 \times 10^{-25}$), previously linked with increased risk of multiple sclerosis[44]. Conversely, a protective allele against EBV DNAemia, HLA-DRB1*12:01 (UKB, $P = 4.6 \times 10^{-18}$; AOU, $P = 3.9 \times 10^{-4}$), has been associated with less severe multiple sclerosis[45]. We also observed two other negatively associated HLA alleles, HLA-B*35:01 (UKB, $P = 1.3 \times 10^{-28}$; AOU, $P = 2.6 \times 10^{-18}$) and HLA-B*55:01 (UKB, $P = 7.3 \times 10^{-29}$; AOU, $P = 2.8 \times 10^{-14}$), that present known immunodominant epitopes from the EBV proteome[46,47]. These results collectively suggest that strong peptide presentation may underlie decreased EBV DNAemia.

Motivated by these findings, we hypothesized that systematic predictions of EBV peptide display and processing could further characterize variation in population-level EBV DNAemia. We used NetMHC (NetMHCpan and NetMHCIIpan)[48] to infer the binding affinity of all potential EBV epitopes in the viral proteome with all HLA alleles observed in the UKB NFE cohort (Fig. 5a and Methods). Following past works in which candidate singular immunodominant epitopes were prioritized[49,50], we summarized the top-ranking peptide per allele from NetMHC for both class I and II alleles. The top-predicted epitopes prioritized by NetMHC were corroborated by previously identified EBV antigens in the Immune Epitope Database (IEDB)[51], including 9 of 83 (10.8%) class I peptides and 7 of 110 (6.4%) class II peptides (Fig. 5b, Extended Data Fig. 6d and Supplementary Table 7). These overlaps were significantly enriched over a random set of peptides for both class I ($P = 3.5 \times 10^{-23}$; binomial test) and class II ($P = 0.047$; binomial test), verifying the capacity for NetMHC to predict viral peptide processing and presentation across HLA alleles. We also observed that predicted immunodominant

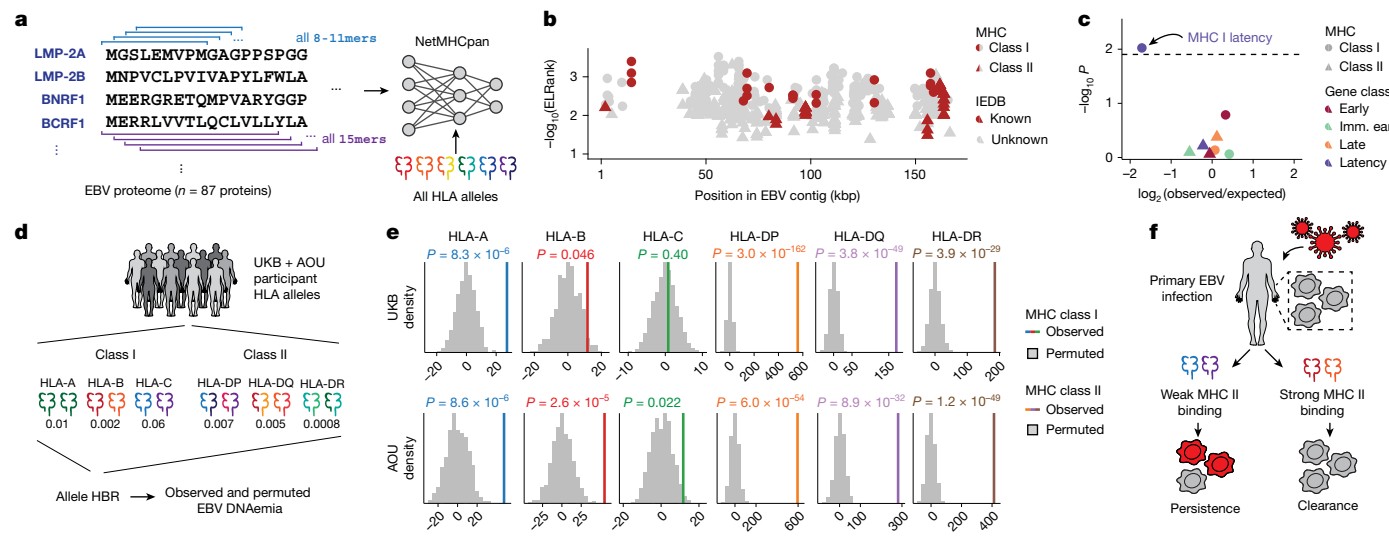

**Fig. 5 | Variable predicted EBV antigen presentation underlies EBV DNA persistence. a**, Schematic of scoring the EBV proteome for antigen presentation against all HLA alleles observed in the UKB NFE cohort using NetMHCpan and NetMHCIIpan[48]. **b**, Summary of top antigens bound per MHC class, coloured by whether the peptide is experimentally confirmed in IEDB[51]. **c**, Enrichment analyses of immunodominant peptides as a function of EBV gene functional class. The highlighted dot shows MHC I peptides are depleted for latency-associated genes. The significance cutoff was drawn at $P = 0.05/4$, to adjust for the four gene classes tested. Statistical test: two-sided Fisher's exact test. **d**, Schematic of HBR per HLA allele, which is used as input for downstream analyses. **e**, Summary of the change in comparing individuals with and without EBV DNAemia. $P$ values are the result of a permutation test ($n = 100$ permutations; two-sided). **f**, Overview of an inferred model of antigen processing and presentation via MHC, resulting in persistence or clearance of EBV DNA following infection. The images in panels **d**,**f** were adapted from ref. 19, Springer Nature Ltd.

peptides were depleted in latency-associated EBV genes specifically for MHC class I peptides, reflecting potential viral evolution to evade host immunity during latency[52] (Fig. 5c).

Recent work has shown that aggregation of immunodominant epitopes of the NetMHC scores via a harmonic mean of the best ranked peptide (HBR) is predictive of immune response, including to neoantigens in tumours[49,50]. We reasoned that analogous measures could predict the immune processing and recognition of viral epitopes and therefore summarized the per-person, per-allele HBR for class I and II MHC (Fig. 5d). We developed two heuristics to assess the ability of these HBR scores in predicting EBV DNAemia, using both a permutation-based and regression-based framework (Methods). We compared the mean difference in HBR for individuals with and without EBV DNAemia for 100 permutations. For class I presentation, HLA-A ($P = 8.3 \times 10^{-6}$) and HLA-B ($P = 0.046$), but not HLA-C ($P = 0.40$), were associated with individual persistence of EBV DNA (Fig. 5e). Conversely, for class II presentation, each allele was strongly associated (HLA-DP, $P = 3.0 \times 10^{-162}$; HLA-DQ, $P = 3.8 \times 10^{-49}$; HLA-DR, $P = 3.9 \times 10^{-29}$), consistent with the role of CD4-mediated immunity of viral infections via class II antigen presentation by B cells and dendritic cells[53]. These enrichments were concordant with identical analyses in the AOU EUR cohort (Fig. 5e). An orthogonal statistical regression framework that accounted for potential confounders, including the full HLA haplotype per individual, produced concordant results (Extended Data Fig. 6e–g and Methods). Together, these results demonstrate that computational modelling of interactions between host alleles and the viral proteome is predictive of the incidence of EBV DNAemia and support a model where individual genetic variation, predominantly in MHC class II, controls EBV DNA persistence in blood (Fig. 5f).

## Genetic diversity in EBV genomes

A longstanding hypothesis is that genetic variation in EBV genomes could explain the diversity in host responses ranging from tolerance to pathogenesis[54]. However, recent reports have shown that variants in

EBV previously attributed to oncogenicity were more closely tied to geographic origin than functional variation[55]. Distinguishing geographic structure from true oncogenic potential is critical, as EBV-driven tumours display pronounced regional enrichments, including nasopharyngeal carcinoma (NPC), which is prevalent in southeast China, northern Africa and other regions in southern Asia. We reasoned that our composite measure of the circulating genetic variation of EBV in ostensibly healthy individuals could stratify functional EBV variants of unknown significance (VUS) in tumour samples (Methods). After verifying reproducible viral genetic variation in both biobanks, we examined 31 previously reported EBV protein-altering mutations from patients with NPC[55] (Extended Data Fig. 7a,b, Supplementary Note 6 and Methods). We annotated these VUS on the basis of our observed EBV allele frequencies in the UKB and AOU. Notably, all but four variants were detected in one or both cohorts at an allele frequency of ≥10% (Extended Data Fig. 7c and Supplementary Table 8). The other 27 variants previously detected in NPC genomes are unlikely to be sufficient for pathogenesis, based on their prevalence in healthy individuals in the UK and USA. Hence, these 27 VUS probably either reflect geographical drift or require an epistatic effect for driving malignancy. When assessing the four VUS exceptions, our viral proteome NetMHC workflow suggested that these four VUS are unlikely to alter peptide presentation (or thereby enable immune evasion), indicating that these variants, if indeed functional, may modulate viral-intrinsic functions (Extended Data Fig. 7d,e and Methods). In total, our approach of synthesizing pieces of viral genomes from excluded WGS reads of hundreds of thousands of individuals provides an alternative to low-throughput amplification and sequencing of healthy control individuals[55,56] to resolve potential functional variation in the EBV genome.

## Discussion

The exponential rise in population-scale sequencing has transformed our understanding of the genetic determinants of complex phenotypes[57]. Although these biobanking efforts were originally genotyped

using DNA microarrays, more recent exome and whole-genome sequencing cohorts have discovered a diversity of rare genetic variants underlying complex traits[17,29]. Here we show that these same large-scale sequencing libraries contain sufficient EBV nucleic acid content to derive a new molecular biomarker, once corrected for low complexity and biased regions. Our analyses show that host genetic variation significantly contributes to the persistence of EBV DNA following infection, which in turn associates with a variety of both known and speculated phenotypic outcomes. Beyond confirming established associations between EBV and respiratory and autoimmune diseases, analyses of EBV DNAemia nominated various neurological indications, including rare conditions that have been anecdotally tied to EBV infection in past work. Further, we characterize EBV DNAemia as a polygenic trait regulated by genetic loci affecting antigen presentation, as well as both adaptive and innate immune signalling. We also identify individual HLA alleles linked to heterogeneous autoimmune diseases that modulate risk of EBV DNAemia, observing an overall trend where predicted viral peptide presentation strength was negatively correlated with viral persistence. Finally, the aggregate of viral-derived WGS reads reflected the circulating EBV strains, enabling studies that contrast the composition of viral heterogeneity across population-scale cohorts. Collectively, our framework extends evaluations of endogenous HHV-6 that have nominated loci in linkage with germline integration[58], whereas our analyses reveal that acquisition of viral DNA over a lifetime is a trait subject to genetic regulation at the population level.

Despite >90% EBV seropositivity among adults in the UK and USA, we identify a distinct population of 9.7–11.9% of individuals with detectable EBV DNA in peripheral blood, suggesting that past infection is necessary but not sufficient for EBV DNAemia. Instead, simulations imply that the EBV DNAemia population reflects a tail of exposed individuals with the highest EBV DNA levels. We hypothesize that others in these cohorts are carriers of EBV DNA from past infections, but at levels below our limit of detection. Further, as viral DNA levels can fluctuate longitudinally[59], these WGS measurements represent a snapshot of a complex process that marked individuals with potentially transiently high levels of EBV DNA.

Our work provides a scalable framework for repurposing population-scale WGS to define genetic determinants of viral persistence. Although this study focused on EBV in cohorts from the UK and USA, our approach may extend to a broad range of viruses including phages and eukaryotic viruses that comprise the human virome worldwide, including viral species from the Polyomaviridae, Adenoviridae, Parvoviridae and *Anelloviridae* families. A limitation of blood-derived WGS is that it only detects pathogens that persist in the peripheral circulation. Characterizing host responses for other pathogens, including RNA viruses, will require cohorts that integrate DNA- and RNA-sequencing across diverse anatomic sites. Equally important will be expanding large-scale sequencing to geographic regions with variable persistent viruses, enabling representation of viral diversity that is currently under-sampled. Together, these current and future efforts will yield a tissue-resolved map of viral reservoirs and infection while clarifying how human genetic diversity shapes lifelong interactions with our viromes.

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

## Methods

### Rationale of EBV detection

The 171,823-nucleotide EBV genome (NC_007605.1) was first included in December 2013 (hg38 version GCA_000001405.15) as a sink for off-target reads that are often present in sequencing libraries, to account for pervasive EBV reads present from the immortalization of LCLs (as with the 1000 Genomes Project and related consortia). Importantly, WGS in the UKB and AOU consortia was performed on whole blood[18,60], reflecting that EBV reads detected would derive from viral DNA from past infections.

### WGS data and cohort analyses in the UKB

For the UKB, we obtained per-base abundance of EBV DNA of the 490,560 WGS libraries by extracting reads aligning to chrEBV in the hg38 human genome reference that had a read mapping quality (MAPQ) ≥ 30 (q30) via the SAMtools view command[61]. To quantify EBV DNA abundance for each position, we summed the coverage of each base in the EBV genome across all libraries (per-base abundance). The resulting coverage across the viral contig was approximately flat, supporting that EBV DNA detection from WGS reads was real viral DNA, with two key exceptions (Fig. 1b). First, a total of 27,692 positions had low to no coverage (per-base abundance ≤10) due to low mappability of the EBV contig. Second, two regions (positions 36,390–36,514 and 95,997–96,037) had orders-of-magnitude higher coverage (per-base abundance of ≥$10^3$ at these 166 positions). On further examination, the sequences were highly repetitive. Hence, we reasoned that these two regions may confound EBV DNA quantification. To assess this, we calculated EBV DNA abundance per person before and after masking, by summing MAPQ ≥ 30 coverage either across all $J = 171,823$ bases, or only across the remaining $J' = 143,965$ well-covered bases (10 < per-base abundance < $10^3$ for each base). The per-individual EBV sum unmasked was computed over all $J$ bases, whereas the masking was performed over $J'$ bases.

We then used a two-sided Fisher's exact test to test for association between EBV DNA presence (EBV DNA coverage > 0) and EBV serostatus, recorded in the UKB as 'EBV seropositivity for Epstein–Barr Virus' (data field 23053). Before masking, EBV DNA presence had a weak but insignificant positive association with EBV seropositivity (odds ratio = 1.2, $P = 0.03$). Conversely, after masking these repetitive regions and recomputing donor detection status, the association between EBV DNA detection and seropositivity was much stronger (odds ratio = 14.6, $P = 1.7 \times 10^{-26}$) (Fig. 1c). These analyses demonstrate that masking highly repetitive regions in the viral contig is required to perform valid inferences from whole genome sequencing data, as evidenced by statistical overlap with EBV serostatus.

### Contig mappability analyses

To confirm that regions of the EBV contig that were not detected were attributable to poor mapping quality of those regions, we generated synthetic reads of length 101 bases by tiling the reference EBV contig. Next, each synthetic read was aligned using bowtie2 v.2.5.1 (ref. 62). We define mappability as the percentage of reads overlapping a position with a map quality score exceeding ten. This analysis reproduced regions depleted from the pseudobulk abundance (Extended Data Fig. 1a), indicating that low detection in these regions was due to homology in the hg38 reference rather than variable DNA presence from past infection.

### EBV DNA copy number estimation, simulation and thresholding

To calculate EBV DNA abundance per person, we summed the coverage over the well-covered, non-biased bases ($J'$). We normalized this value against the effective EBV genome size (143,965 bases) to obtain an estimate of the coverage per EBV genome. Next, we used the 30× human WGS coverage and accounted for the diploid human

genome to compute an estimate of EBV DNA copy number per human cell, which resulted in approximately 1 in 1,000–10,000 cells in individuals with detectable EBV DNA (that is, our limit of detection was approximately 1 EBV genome per 10,000 cells). To contextualize these values, the upper range of EBV copy numbers in healthy individuals measured using qPCR was $10^3$ EBV genomes per 1 µg DNA, or 1 EBV genome per 200 cells[23]. The latter number was estimated with the assumption that $10^5$ cells produce 0.5 µg DNA. Although a previous study similarly used EBV reads in a cohort of ~8,000 donors, this analysis did not correct for the repetitive, biased DNA abundances that significantly skewed the resulting quantification[4]. After quantifying per-person EBV DNA abundance, 85.7% of individuals in the UKB had no detectable EBV DNA.

In the UKB cohort, over 90% of individuals are seropositive, yet only 14.3% of individuals have non-zero EBV DNA levels detected. Therefore, we conducted a simulation study to better characterize the discrepancy. Using maximum likelihood estimation, we estimated values for the mean and standard deviation of a log-normal distribution to initialize the simulation and subsequently modified these values to (1) account for a mixture including 10% zeros (representing the individuals who were not infected with EBV) and (2) adjust the mean for a round, interpretable number. The final values used in the simulation (Extended Data Fig. 1d) were set to zero for 50,000 individuals, whereas the remaining 450,000 individuals were simulated via a log-normal distribution, with a mean of 0.2 EBV genome copies per 10,000 cells, a standard deviation of 0.62 and a censored value of 0.71. We emphasize that this simulation does not test an explicit statistical question but is designed primarily for illustrative purposes, to show that a single underlying component can explain many features of the empirical data (rather than requiring a second condition).

The extreme skew of the EBV levels distribution (Extended Data Fig. 1f) motivated our transformation of EBV DNA copy number to a binary trait, which we define as EBV DNAemia, since a quantitative trait otherwise assumes a dose-dependent relationship when testing for associations. To binarize our data for downstream analyses, we used a series of two-sided Fisher's exact test to survey different cutoffs against association with EBV serostatus (Fig. 1g). Our goal was to determine an optimal EBV copy number threshold. We observed the most significant positive association with a threshold of 1.2 EBV copies per $10^4$ human cells (odds ratio = 82.17, $P \approx 0$) after accounting for standard covariates used in a GWAS analysis (age, sex, age × sex, and ancestry PCs 1–15). This corresponded to having a per-person abundance of at least 302 bases covered on the EBV genome, which in turn corresponded to a full paired-end sequencing read (2 × 151 bp) with no soft-clipping. There were 47,452 people (9.67%) with EBV copy numbers greater than this threshold, which was used for all downstream analyses.

For the 9,607 individuals with both EBV serology and WGS available, there were 919 individuals (9.57%) that had EBV DNAemia. Only two (0.2%) of these 919 individuals were seronegative. One donor had an EBV DNA load of 1.36 EBV genomes per $10^4$ cells (just above our EBV DNAemia cutoff) with a high VCAp18 titre, but low titres for the other three EBV antigens. The other donor had an EBV DNA load of 3.34 EBV genomes per $10^4$ cells, with a positive titre for EA-D but low titres for the other antigens. In other words, among the 347 donors with no seropositivity against any antigens, none were annotated as individuals with EBV DNAemia (Extended Data Fig. 1b).

### EBV DNA detection in AOU

We obtained per-base abundance of EBV DNA for 245,394 people in AOU with WGS data similarly by extracting reads that mapped to chrEBV in the hg38 human genome reference with MAPQ ≥ 30. To quantify EBV DNA abundance per base, we summed the q30 coverage of each base in the 171,823 bp EBV genome across all people. We again observed an overall uniform coverage; 23,513 positions had no coverage (per-base abundance = 0), and four regions (positions

# Article

36,389–36,516; 52,012–52,034; 95,997–96,037 and 163,596–163,617) had abnormally high coverage (per-base abundance of >1,000 at 214 positions; Extended Data Fig. 2b). The effective EBV genome size was the remaining 148,096 bases (>0 but <$10^3$ for each base). Although the largest repetitive region was the same in both the UKB and AOU, differences in the other regions with variable bias could be attributed to differences in the alignment software for either cohort, noting that all analyses used the existing mappings from either cohort.

We quantified the EBV copy number per person in AOU with a similar approach to the one used for the UKB. In brief, we quantified EBV DNA loads after masking and normalized them by the effective EBV genome size, then by the average genome coverage (30× human WGS) provided by AOU metadata. A total of 51,459 people (21%) had detectable EBV DNA (Extended Data Fig. 2c). The top EBV DNA load harboured was ~1 EBV copy per 1.4 cells (or 7,046 EBV copies per $10^4$ cells).

Using the same EBV DNA copy number thresholds as in the UKB, a total of 29,249 people (11.9%) had EBV copy numbers greater than the threshold of 1.2 EBV copies per $10^4$ human cells (Extended Data Fig. 2c). The overall higher EBV loads in AOU compared to in the UKB may be due to a difference in the recruitment criteria and demographics of the two cohorts: relative to the general population (as in AOU), the UKB shows a 'healthy volunteer bias' where participants were less likely to have self-reported health conditions[57]. In comparison, the maximum copy number described in a previous paper was a few orders of magnitude higher (2,404,531 EBV copies per $10^5$ human cells), potentially due to our exclusion of abnormally high coverage regions[4].

## Phenome-wide association studies

We conducted PheWAS using the UKB as a discovery cohort to test for the association between EBV DNAemia and 13,290 binary phenotypes and 1,931 quantitative phenotypes amongst participants with broadly NFE as in the GWAS (refer to the following section). We used logistic regression with Firth correction, including sex and age as covariates. Using a Bonferroni correction, we defined $0.05/15,221 = 3.3 \times 10^{-6}$ as our significance threshold. To ensure that the PheWAS was not confounded by immunosuppressive drugs, we ran a secondary analysis in which we included immunosuppressive drug status as an additional covariate in the regression. Because a majority of blood samples used for WGS were drawn at the time of enrollment, we identified these individuals on the basis of medication taken at the time of their initial assessment visit (UKB data field 20003). A full list of the 169 medications used for annotating immunosuppressed individuals is reported in Supplementary Table 2.

As validation in AOU, we obtained unique RxNorm codes for 53 of the 169 drugs (Supplementary Table 2) and queried for individuals that had any of these drug exposures, along with the exposure start and end dates. We annotated each individual as immunosuppressed only when the biosample collection date for WGS fell between the drug exposure start and end dates (or after start dates, if no end date was recorded). We observed a positive but not significant association between immunosuppressive drug exposure at the time of WGS collection and EBV DNAemia (odds ratio = 1.03, $P = 0.54$) (Extended Data Fig. 2f).

We replicated PheWAS associations using the AOU cohort of individuals with European ancestry via Fisher's exact tests for association between EBV DNAemia and each representative ICD-9 or ICD-10CM code in AOU. As recommended in the AOU workbench, we defined a representative ICD code as a code appearing at least twice in a person and 20 instances across all participants. The top results were predominantly being HIV positive, having immunodeficiencies, or receiving organ transplants, which we also observed in the UKB. To compare effect sizes between hits in the UKB and AOU, we matched AOU ICD-10CM codes to a corresponding UKB ICD-10 code by taking the first four characters of the ICD-10CM code, as codes >4 characters do not exist in the ICD-10 ontology used in UKB. For the two traits linked to EBV discussed in the main text, multiple sclerosis was queried using the ICD-10CM code 'G35' in AOU, and gammaherpesviral mononucleosis was queried using the ICD-10CM code 'B27.00'.

## Genetic associations with EBV DNAemia in the UKB

For UKB individuals of broadly NFE ancestry, array-based imputed genotypes with good genome-wide coverage in the common (>5%) and low-frequency (1–5%) MAF ranges were available[17]. Genotyping arrays capture genome-wide genetic variations (SNPs and indels) within both coding and noncoding regions, allowing imputation of genotypes and tests for association between genotypes and a specified trait. To avoid confounding results due to differences in ancestral background, we stratified the cohort across six broad genetic ancestries (African, AFR; Hispanic or Latin American, AMR; Ashkenazi Jewish, ASJ; East Asian, EAS; non-Finnish European, NFE; and South Asian, SAS) before testing for associations between EBV DNAemia and UKB-imputed genotypes, which resulted in a total of 450,032 individuals with array imputed genotype data available, including 426,563 individuals of NFE ancestry. We then used REGENIE v.3.5 (ref. 63) to examine associations between EBV DNAemia and imputed genotypes, using a logistic model with covariates and applying Firth correction: EBV DNAemia ~ age + sex + age × sex + $age^2$ + $age^2$ × sex + batch + ancestry PCs 1–20, as previously described[64]. The input to REGENIE includes directly genotyped variants (MAF > 1%, MAC > 100, genotyping rate per variant >99%, and genotyping rate per individual >80%). We pruned these variant sets using PLINK2 (--indep-pairwise 1000 100 0.8) as input to REGENIE's step1 analyses. This step produces a whole genome regression model to fit to the binary trait of EBV DNAemia and outputs a set of genomic predictions.

For REGENIE step2, we further filtered out SNPs that had 0.99 'missingness', imputation INFO < 0.7, and p.HWE > $1 \times 10^{-5}$. This step fits a logistic model to imputed data, using the genomic predictions from step1. To estimate heritability of SNPs and genomic inflation, we performed linkage disequilibrium score regression (LDSC) by applying the ldsc package (v.1.0.1). In brief, we used munge_stats.py on the cleaned summary stats, then used ldsc.py to estimate $h^2$ using the supplied 1KG Genomes linkage disequilibrium score matrices (Supplementary Note 4). Identical steps were applied to conduct the EBV serology GWAS on the subset of UKB participants for whom EBV serostatus was measured[40].

To annotate variant loci, we focused on significant variants ($P < 5 \times 10^{-8}$) and created genomic intervals of ±1 Mb around each variant. As variants on chromosome 6 often exhibit linkage disequilibrium with MHC, we created a custom interval (chr6: 25,500,000 to 34,000,000) for the HLA region. We then combined overlapping intervals using the GenomicRanges reduce function and selected the most significant variant per interval as the index variant. In the case of ties, we selected the variant closest to the midpoint of the region. We applied the reduce function again to ensure we had a set of non-redundant index variants. Finally, we annotated each variant by the closest gene, using Ensembl v.111 (Jan 2024) gene annotations and selecting the gene whose midpoint was closest to the index variant. For visualization of specific loci, we used the canonical hg38 reference genome isoforms. Linkage disequilibrium was determined via LDlink[65] for the regions noted (Supplementary Note 4). Zoom plots were from the array-based GWAS associations in the UKB, and the linkage disequilibrium reference panel in LDLink[65] used all European populations.

We complemented our GWAS with an exome-wide association analyses (ExWAS), leveraging the whole genome sequencing data available in the UKB. Specifically, we tested for associations between EBV DNAemia and protein-coding variants observed in at least six participants of NFE ancestry in the UKB. We applied our previously described protocol to generate variant-level statistics[29,66]. Variants were required to pass the following quality control criteria: coverage ≥10x; ≥0.20 of reads with the alternate allele for heterozygous genotype calls; binomial test of alternate allele proportion departure from 50% in heterozygous state $P \geq 1 \times 10^{-6}$; GQ ≥ 20; Fisher Strand Bias ≤ 200 for indels and ≤ 60 for SNVs;

root-mean-square mapping quality (MQ) ≥ 40; QUAL ≥ 30; read position rank sum score (RPRS) ≥ −2; mapping quality rank score (MQRS) ≥ −8; DRAGEN variant status = PASS; and ≤10% of the cohort with missing genotypes. Additional out-of-sample quality control filters were also imposed based on the gnomAD v2.1.1 exomes (GRCh38 liftover) dataset[67]. The sites of all variants were required to have ≥10x coverage in ≥30% of gnomAD exomes and, if present, each variant was required to have an allele count ≥50% of the raw allele count. Variants with missing values for any filter were retained unless they failed another metric. Variants failing quality control in >20,000 people were also removed. *P* values were generated via Fisher's exact two-sided test. Three distinct genetic models were studied for binary traits: allelic (A versus B allele), dominant (AA + AB versus BB), and recessive (AA versus AB + BB), where A denotes the alternative allele and B denotes the reference allele. ExWAS hits were filtered following: $P < 5 \times 10^{-8}$, nCases >20, and protein-altering Most Damaging Effect ('Stop_lost', 'Stop_gained', 'Start_lost', 'Splice_region_variant', 'Splice_donor_variant', 'Splice acceptor variant', 'Missense_variant', 'Frameshift_variant', 'Disruptive_inframe_insertion', 'Disruptive_inframe_deletion'). For functional variant annotation and interpretation, AlphaMissense[68] was executed on all variants that were statistically significant from the ExWAS analyses using default parameters. If multiple transcripts were associated, only one is reported (the one with the highest AlphaMissense score, if available).

### Replication of UKB EBV DNAemia-associated genotypes

To broadly capture variants in individuals with EUR ancestry in AOU, we used the variant-level metadata for the SNP and indel variants contained in the short read WGS (srWGS) data dictionary. We filtered for variants with an alternative allele frequency (AF) of 0.01 < AF < 0.49 or 0.51 < AF < 0.99 (gvs_eur_af) and at least 100 individuals containing this variant (gvs_eur_sc ≥ 100) in the EUR subpopulation as the input SNPlists to step1 and 2 of the REGENIEv3.2.4 pipeline. This resulted in 16,566,413 variants across chromosomes 1–22. EBV DNAemia was supplied as a binary trait, along with the covariates age, sex, age × sex, and ancestry PCs 1–15. There were 133,578 such individuals that had EBV DNAemia status determined, of which 131,938 had complete covariate data and were included in the analysis, and 12,099,305 total variants had GWAS statistics results.

### Genomic architecture associations

To holistically evaluate genetic architecture similarities between EBV DNAemia and IMDs, we used the R package cupcake[38]. The package was used to define shared components of genetic architecture across 13 IMDs, applying shrinkage to adjust for linkage disequilibrium, allele frequency and differential sample size. Summary statistics of 13 large IMD GWASs were used to define a reduced dimension space using PCA, which served as a common genetic basis that enabled simultaneous comparisons between multiple diseases. The reduced dimension space included 566 driver variants and 13 PCs that were defined as orthogonal genetic risk components[38]. Applying this approach, we extracted summary association statistics for these 566 driver variants from our UKB NFE EBV DNAemia GWAS. After checking and adjusting the effect allele alignment, we used cupcake[38] to project these variants onto the 13 IMD genetic risk bases and assess the significance of association with each component. The output from this projection is a score or delta (δ) for each PC that quantifies the difference between the projected genetic risk for that trait on a particular basis axis and a synthetic control (which has zero effect sizes for all SNPs). This effectively measures how strongly the trait aligns with the risk architecture represented by that component. To account for uncertainty, the variance of δ is calculated using the propagation of error from the input GWAS summary statistics, adjusted for the same shrinkage weights and allele frequency variance as applied in basis construction. With δ and its variance, a *Z*-statistic can be formed for each component, and standard statistical inference can be used to compute a *P* value[38].

### Pathway and single-cell analyses

To evaluate the gene expression program uncovered by our ExWAS associations, we used a high-resolution single-cell cellular indexing of transcriptomes and epitopes by sequencing (CITE-seq) dataset of PBMCs from eight distinct donors with 210,911 quality-controlled cells[41]. The 148 ExWAS-associated genes were input alongside the preprocessed Seurat[41] object into the AddModuleScore function with default hyperparameters. To reduce technical variation, we removed genes mapping to the HLA region as well as ribosome-associated genes from the input gene list (HLA for genetic polymorphisms; ribosome for cell quality) from the module score foreground and background. Downstream association analyses of cell type enrichment were performed using the pre-supplied labels.

Pathway enrichment analyses were performed using the same ExWAS gene set via the clusterProfiler R package[69]. Gene set analyses were performed using the enrichGO (for biological processes) and enrichKEGG functions (for pathways) using the set of 148 genes and all ENSEMBL human genes as a background set. For analyses with HLA (Fig. 4e) and chromosome 6 excluded (Fig. 4f), we removed either HLA or chromosome 6 genes from both the foreground (that is, test set) and background set for statistical analyses. We used the simplify() function in clusterProfiler with a similarity cutoff of 0.7 (the default value) to reduce the number of redundant association terms. Hence, we note that the labels in panels Fig. 4d–f are not identical in name; this result is due to the simplify() function's selection of a single term that is nearly identical to other related terms.

Enrichment analyses for non-coding enrichment in accessible chromatin used 18 fluorescent activated cell sorting (FACS)-isolated immune and hematopoietic populations that were uniformly reprocessed and aggregated using the hg19 reference genome (Extended Data Fig. 5a). To compute enrichment scores, we isolated genome-wide significant variants from the UKB NFE GWAS, lifted over the hg38 coordinates to hg19, and built a RangedSummarizedExperiment object to compute the enrichment. For accessible chromatin enrichments, we used an approach motivated by the chromVAR statistical testing framework adapted for genetic variants. Specifically, 100 background peaks (identified through the same mean and GC content of the ATAC-seq peak) were used as a null distribution, and the mean deviations at peaks variably containing genome-wide significant variants were computed via the abundance of accessible chromatin from each sorted population. The background and observed deviations were used to estimate an empirical *Z*-statistic, which was transformed into a *P*-value using the pnorm() R function.

### HLA haplotype and EBV peptide presentation

We used the four-digit HLA imputation calls processed in the UKB Research Analysis Platform using HLA*IMP:02 (ref. 70). Allele dosage values of >0.7 were used to assign donor haplotypes for a specific four-digit HLA allele. Homozygotes were determined by alleles with values of >1.3. For the AOU cohort, predetermined HLA genotypes were not available in the workbench. Hence, we reconstructed the HLA calls for all individuals of EUR ancestry using the T1K toolkit[71] (v.1.0.8-r237) by extracting reads aligning to the HLA region, which included canonical chr6 HLA region (chr6: 25,500,000 to 34,000,000) and all alternative HLA contigs in the hg38 reference. Using a .bed file of the HLA region coordinates, these alignments were streamed with the GATK PrintReads commands into the T1K genotyper, which was set to default parameters. Following T1K toolkit recommendations, the donor haplotypes were assigned for alleles called with a quality score of >0. Homozygotes were determined by donors with only a single allele and with a quality score of >30.

To determine specific HLA associations with EBV DNAemia, we used the per-person four-digit HLA alleles for both class I and II as predictors in a logistic regression, with EBV DNAemia as an outcome. Models

included standard covariates used throughout the paper (age, sex, genetic PCs and so on). We performed this regression on the 208 HLA class I and 145 HLA class II alleles in UKB NFE individuals. We then repeated the same analysis for 175 class I and 132 class II alleles that were also present in the AOU EUR cohort (Supplementary Table 7).

The amino acid sequences of all 87 unique EBV protein sequences were obtained from the peptide sequence of the nuccore NC_007605. The protein .fasta file was input to NetMHCpan, along with all observed MHC class I (HLA-A, HLA-B or HLA-C) and class II (HLA-DR, HLA-DP or HLA-DQ) alleles in the UKB NFE cohort. Sliding windows of all 8-, 9-, 10- or 11-mers of the provided protein sequences were generated for the prediction of class I allele peptide presentation; sliding windows of size 15-mers were used for class II. The binding scores of these peptides were determined for all observed UKB NFE MHC alleles that could be scored by NetMHCpan4.1 and NetMHCIIpan4.3 (ref. 48).

The NetMHC output reflects the predicted %rank score for each peptide and a given allele, which is a measure of the rank of the predicted affinity of the allele for the peptide compared to a set of 400,000 random natural peptides. For MHC class I, we computed the HBR score per allele by taking the harmonic mean over the two genotyped alleles for each of HLA-A, B and C. For homozygotes, the harmonic mean is equivalent to any individual observation. For individuals missing a single allele, we considered only the genotyped call, and for two missing alleles, the individual was excluded from the per-allele analysis.

For MHC class II analyses, all HLA-DRB alleles were directly applied as input—along with the EBV proteome .fasta file—to generate HLA-peptide presentation scores for all possible 15-mer sliding windows. As HLA-DQ and HLA-DR alleles exist in pairs of alpha and beta alleles within the predictions, we took all HLA-DQ and HLA-DP alleles imputed in the UKB NFE cohort and generated all possible combinations of HLA-DQA/HLA-DQB alleles and all possible combinations of HLA-DPA–HLA-DPB allele pairs. These alpha–beta allele combinations were then used as inputs to NetMHCIIpan, along with the EBV proteome .fasta file. Again, the output file lists each peptide, the protein from which the peptide is derived, a given class II allele (pair) and the predicted %rank_EL score, which is the percentile rank of the eluted ligand prediction score. As HLA-DRA is the only non-variable gene in the population, each individual has only two possible HLA-DR heterodimers. Each individual can form four possible alpha–beta heterodimers from HLA-DP and HLA-DQ (between alpha and beta molecules). Hence, each individual may assemble up to ten unique heterodimeric MHC class II molecules[50].

The per-allele HBR was computed using the harmonic rank of the heterodimers for each allele class and rescaled by a factor of $10^6$ when computing the final ΔHBR score (shown in Fig. 5). The comparisons were only between the NFE/EUR ancestry populations in either cohort. To further verify that our effect was linked to class II presentation strength, we completed regression analyses using the same set of covariates for our genetic association analyses, which verified that other forms of confounding (for example, population stratification or sex) did not explain the associations between the class II predicted presentation strength and EBV DNAemia.

## EBV viral sequence analysis

Raw sequencing reads from chrEBV were merged from all participants from both cohorts. The aggregated .bam file was transformed into a per-base, per-nucleotide count using bam-readcount[72]. For the type 1 and 2 strain analyses, we sought to quantify the abundance directly from the aligned reads to the chrEBV reference (a type 1 EBV strain). Here we performed a multiple-sequence alignment of the EBNA-2 gene (the major difference between strains) for nuccore IDs K03333 (type 1) and K03332 (type 2) and mapped the MSA coordinates back to the chrEBV reference to identify putative regions that would reflect single nucleotide variation, which, in turn, would reflect strain-level differences. We identified nine variants on chrEBV: 36209C>T, 36226T>A,

36251A>G, 36252A>T, 36258C>A, 36275G>T, 36302A>C, 36312T>A and 36320C>T, where the reference allele was type 1-derived and the alternate was type 2-derived. These variants were selected on the basis of: (1) the combined allele frequency being greater than 99% for the reference and alternate alleles and (2) no overlap with the repetitive regions (Fig. 1b).

Next, we analysed a set of 31 protein-altering mutations in EBV (Extended Data Fig. 7c), which was curated from a recent global-scale analyses of EBV genomes[55] derived from individuals with EBV⁺ nasopharyngeal carcinomas. Of these 31 EBV VUS, there were four VUS that were detected at less than 5% pseudobulk in both cohorts. To assess whether these four VUS were potentially involved in immune evasion, we assembled all possible peptides for presentation on both classes I and II, and then scored these peptides with all of the NFE/EUR observed HLA alleles to compute a NetMHC rank score for both the wild-type and mutated forms of the peptides. As both the wild-type and mutated peptides generally had similar values, and few were near the IEDB-validated thresholds (blue dotted lines; Extended Data Fig. 7d,e), we suggest that these VUS—if there is an effect—are probably not mediated via immune evasion but instead via altered function of the viral protein.

## Reporting summary

Further information on research design is available in the Nature Portfolio Reporting Summary linked to this article.

## Data availability

The UKB data are available to qualified researchers (http://www.ukbiobank.ac.uk/register-apply/). The AOU data are available as a featured workspace to registered researchers of the AOU Researcher Workbench (https://www.researchallofus.org/). Summary statistics from the EBV DNAemia discovery GWAS (UKB NFE individuals) are available at https://my.locuszoom.org/gwas/409414/?token=6385c90400414f34b8ed17679bf1495b and have been uploaded to the GWAS catalogue (GCST90572743).

## Code availability

The code to reproduce custom analyses in this manuscript is available online at https://github.com/clareaulab/ebv_biobank_gwas.

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

**Acknowledgements** We are grateful to the participants and staff of the UKB and AOU for enabling the research. We thank T. Rückert, C. Honeycutt, T. Nawy and members of the

Lareau and Dhindsa laboratories for their helpful feedback. This work was directly supported by the Human Virome Program (U01AT012984 to C.A.L. and R.S.D.), P30CA008748 (C.A.L. and K.K.D), a National Academy of Medicine Catalyst award (C.A.L.) and a Michelson Prize Next-Generation Grant (C.A.L.). S.S.N. is supported by the NSF Graduate Research Fellowship. C.A.L is supported by R00HG012579. K.K.D. is supported by R00HG012203 and R01HG014008. R.S.D. is supported by NIH DP5 OD036131, a Longevity Impetus Grant from Norn Group, Hevolution Foundation and Rosenkranz Foundation, and the Texas Children's Hospital Research Vision Scholar program.

**Author contributions** S.S.N., S.P., R.S.D. and C.A.L. conceived and designed the study. S.S.N., E.M.C., O.S.B., R.S.D. and C.A.L. led bioinformatics analyses. M.S.P., J.C.G., T.A.A., L.C.K., Y.C., H.C., F.H., M.A.F., X.Z.Z., B.H., S.M. and Q.W. supported the bioinformatics analyses. L.S.L. and K.K.D. helped with data interpretation. S.S.N., E.M.C., O.S.B., S.P., R.S.D. and C.A.L. wrote the manuscript with input from all authors.

**Competing interests** O.S.B., F.H., X.Z.Z, B.H., M.A.F., S.M., Q.W. and S.P. are current employees and/or stockholders of AstraZeneca. R.S.D. is a paid consultant of AstraZeneca. C.A.L. is a consultant to Cartography Biosciences. The remaining authors declare no competing interests.

**Additional information**
**Correspondence and requests for materials** should be addressed to Slavé Petrovski, Ryan S. Dhindsa or Caleb A. Lareau.

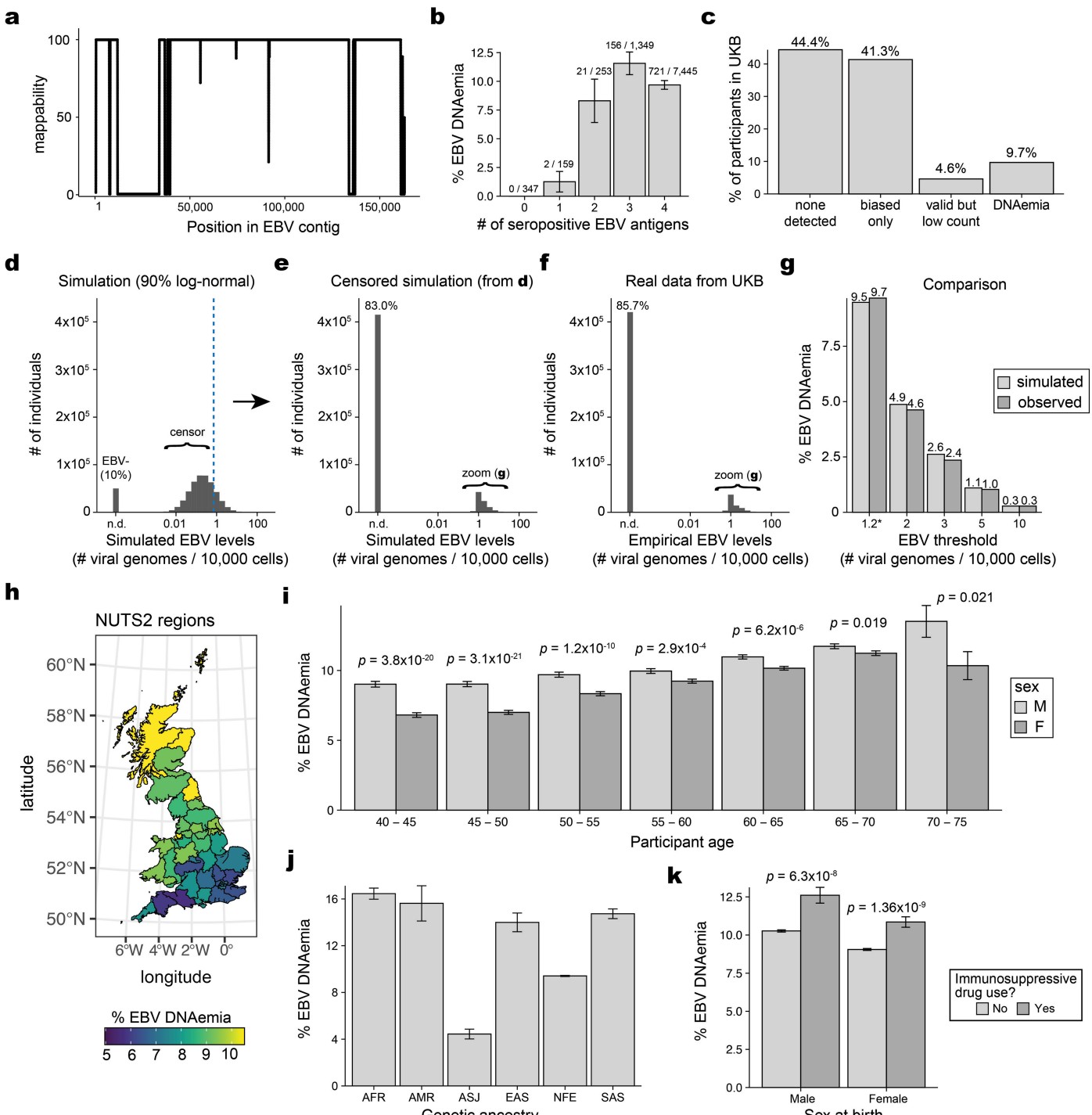

**Extended Data Fig. 1 | Supporting analyses of EBV DNA detected in WGS data from UKB. (a)** Mappability of the EBV contig in the hg38 reference. **(b)** Characterization of EBV DNAemia rates stratified by EBV seropositive antigen number. Overall EBV seropositivity requires 2+ antigens. **(c)** Partition of UKB participants by EBV DNA detection after accounting for biased regions. "Biased only" refers to participants with reads mapping only to the two repetitive regions indicated in Fig. 1b. "Valid and low count" have EBV DNA detected after masking the two biased regions. "DNAemia" exceeds 1.2 EBV copies per $10^4$ human cells. **(d)** Simulated data of a mixture of 10% 0 EBV and 90% log-normal EBV. The dotted line indicates the threshold for data censoring. **(e)** Result of data censoring on simulated data. **(f)** Empirical distribution of observed EBV levels. **(g)** Comparison of donor positivity from simulated and observed EBV levels. The threshold of 1.2 EBV copies per $10^4$ human cells was chosen in the manuscript. **(h)** Geographical distribution of participant birth location coloured by percent EBV DNAemia, split by UK NUTS2 annotations. **(i)** Percent EBV DNAemia resolved by sex and age in UKB. Statistical test: two-sided proportion test comparing sex in the associated age bin. Error bars: standard error of the mean. **(j)** Percent EBV DNAemia resolved by genetic ancestry in UKB. **(k)** Percent EBV DNAemia resolved by sex and immunosuppressive drug use in UKB. Statistical test: two-sided proportion test comparing sex in the associated immunosuppressive drug use bin. For panels b,i,j,k: center is the mean or point estimate of the proportion; error bars: standard error of the mean. Sample size: $n = 490,560$.

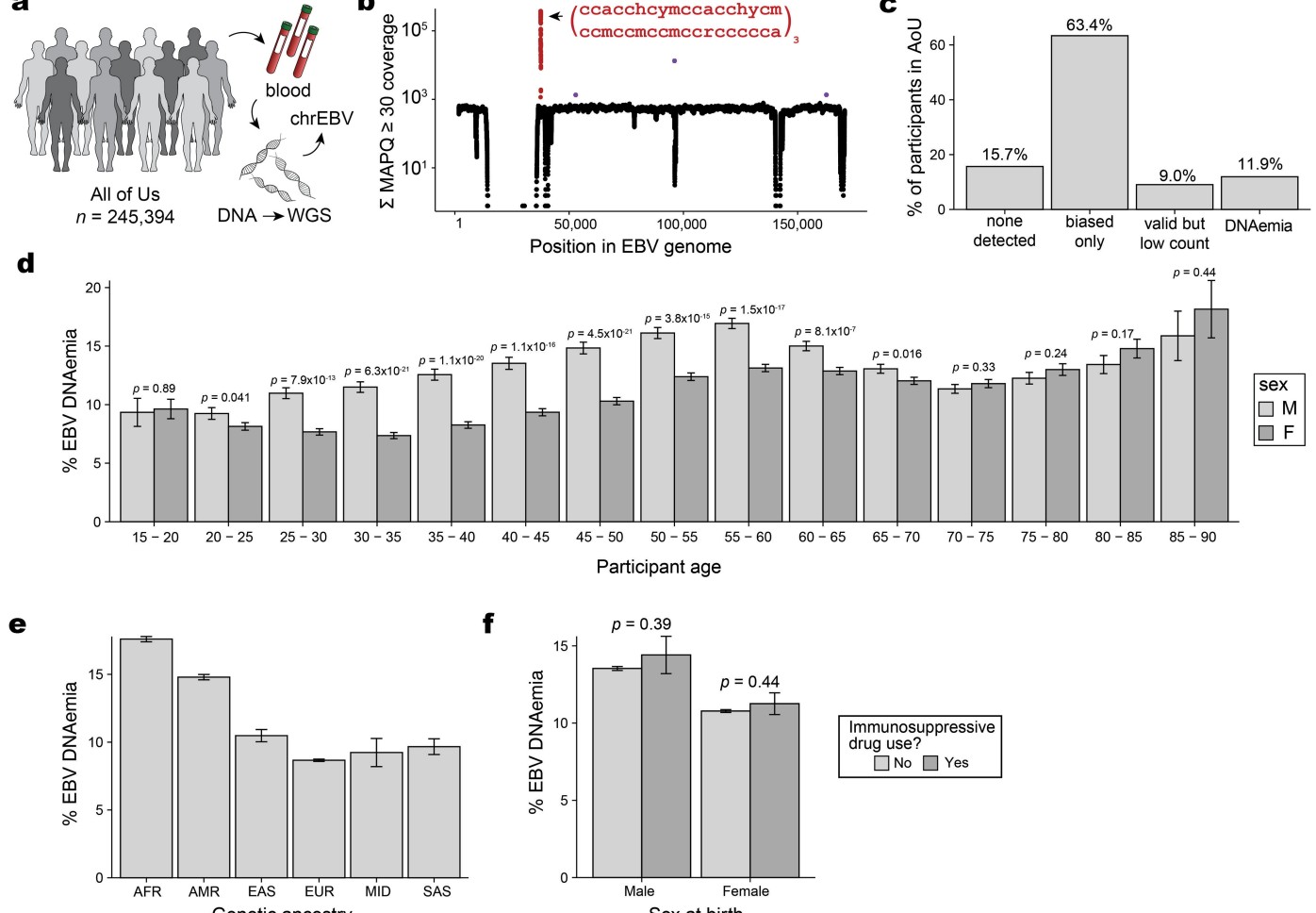

**Extended Data Fig. 2 | Supporting analyses of EBV DNA detected from WGS data from AOU.** (**a**) Schematic of AOU chrEBV extraction from blood-based WGS. (**b**) Sum of per-base read coverage of map quality (MAPQ) score ≥30. (**c**) Partition of AOU participants by EBV DNA detection after accounting for biased regions, the same as in Extended Data Fig. 1c. (**d**) Percent EBV DNAemia resolved by sex and age in AOU. Statistical test: two-sided proportion test. Error bars: standard error of the mean. (**e**) Percent EBV DNAemia resolved by genetic ancestry in AOU. (**f**) Percent EBV DNAemia resolved by sex and immunosuppressive drug use in AOU. Statistical test: two-sided proportion test comparing sex in the associated immunosuppressive drug use bin. Error bars: standard error of the mean. For panels c,d,e,f: center is the mean or point estimate of the proportion; error bars: standard error of the mean. Sample size: $n = 245,394$. The silhouettes in panel **a** were adapted from ref. 19, Springer Nature Ltd.

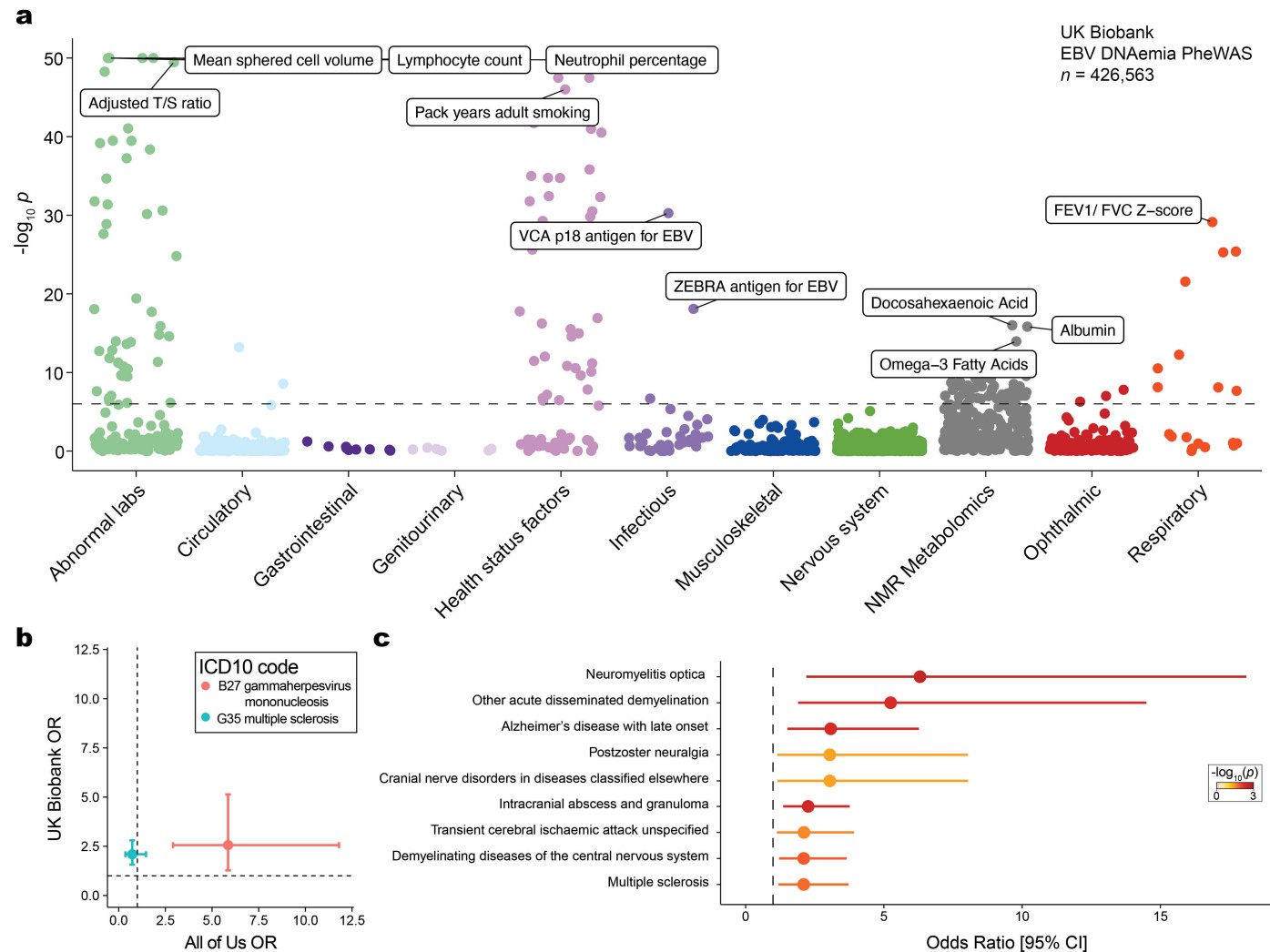

**Extended Data Fig. 3 | Supporting analyses for phenome-wide associations for selected traits.** (**a**) Summary of associations between EBV DNAemia and quantitative traits in UKB individuals of broadly non-Finnish European (NFE) ancestry. The dashed line represents the phenome-wide significant *P* value threshold ($3.3 \times 10^{-6}$). The y-axis is capped at -$\log_{10}(P)$ = 50; all associations are plotted ($n = 1,931$), with those exceeding this threshold plotted at 50. Selected traits are highlighted based on biological interest. Statistical test: Wald test from logistic regression model (two-sided). (**b**) Focused association summary for two ICD-10 codes. (**c**) Top UKB neurological associations, sorted by effect size (odds ratio; OR). Traits were filtered for a minimum of 10 cases and a nominal *P* < 0.05 (logistic regression; two-sided). For (**b,c**), error bars represent the 95% confidence interval of the OR estimate from either cohort. Dotted lines at OR = 1 represent null associations. Center measurement represents point estimate. Sample size: $n = 426,563$ UKB NFE individuals; $n = 133,578$ for AOU European (EUR) ancestry individuals.

**a** EBV DNAemia GWAS partitioned heritability enrichment

**b** GWAS / ExWAS comparison

$r = 1$
$n = 43$ ($p < 1 \times 10^{-5}$)

**Extended Data Fig. 4 | Supporting analyses for NFE genetic association studies from UKB. (a)** Partitioned heritability enrichment via stratified LD score regression. Shown are the 9 genomic features with positive heritability enrichment at a nominal $P < 0.05$. Statistical test: two-sided LD score regression full model. Center of bar: point estimate; error bars: standard errors of mean. Sample size: summary statistics from 426,563 NFE individuals. **(b)** Concordant effect sizes for non-MHC genome-wide suggestive ($P < 1 \times 10^{-5}$; two-sided likelihood ratio test) variants assayed in both the GWAS and ExWAS. Comparison: 43 genetic variants associated at $P < 1 \times 10^{-5}$ in both the ExWAS and the GWAS.

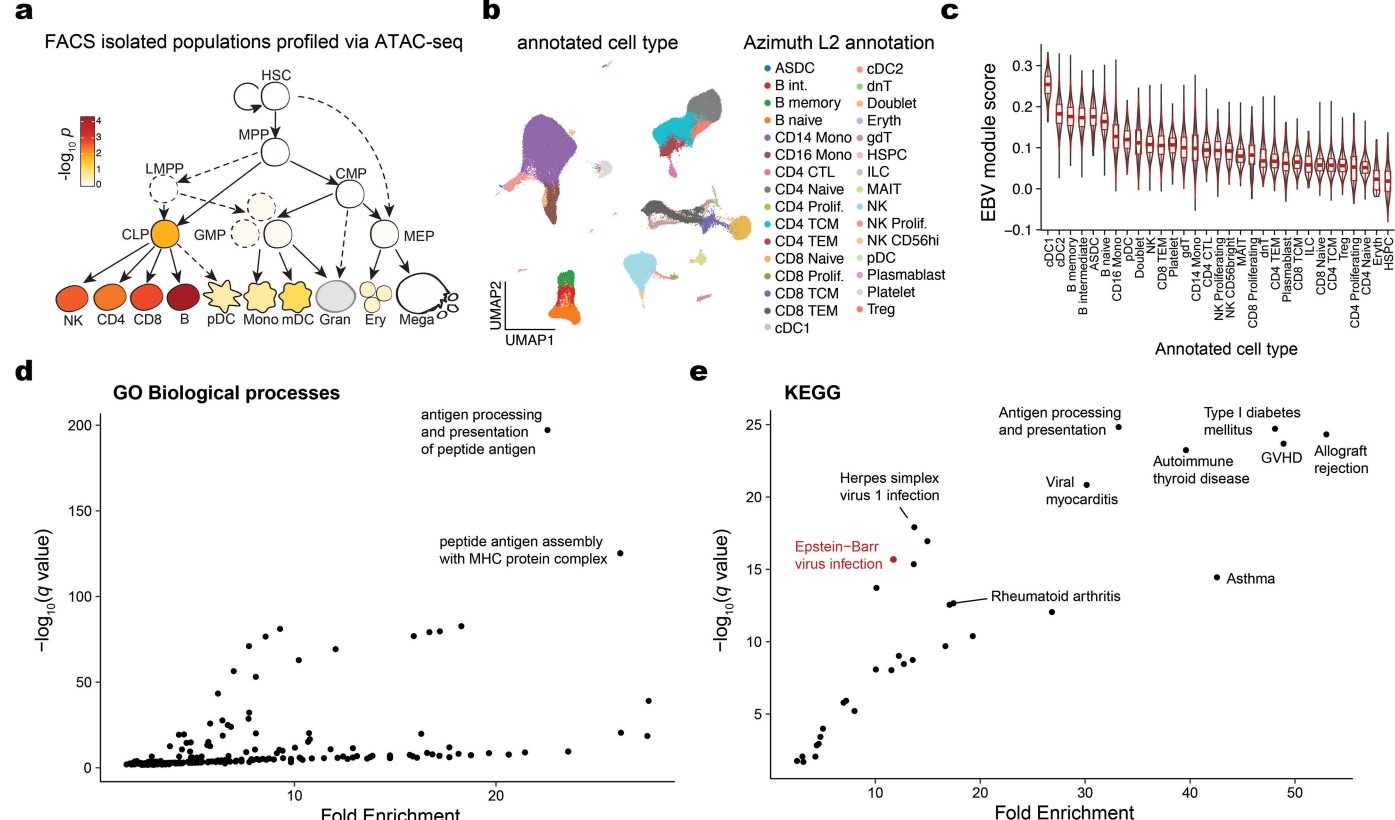

**Extended Data Fig. 5 | Enrichment of EBV DNAemia-associated genetic variants in immune populations and pathways.** (**a**) Schematic of human hematopoietic differentiation, highlighting 18 populations with fluorescent activated cell sorting (FACS)-isolated populations profiled via ATAC-seq. Color: accessible chromatin enrichment at genome-wide significant non-coding loci. Statistical test: empirical permutation test. (**b**) UMAP embedding of 211,000 peripheral blood mononuclear cells. The broad cell type annotation (Azimuth level 2) annotates refined cell types. (**c**) Summary of EBV ExWAS signature stratified by Azimuth L2 cell type, sorted by median score. Boxplots: center line, median; box limits, first and third quartiles; whiskers, 1.5× interquartile range. Sample size: 211,000 cells. (**d**) Summary of GO Biological Processes and (**e**) KEGG gene set analyses. Top pathways based on *q* value and fold enrichment are annotated.

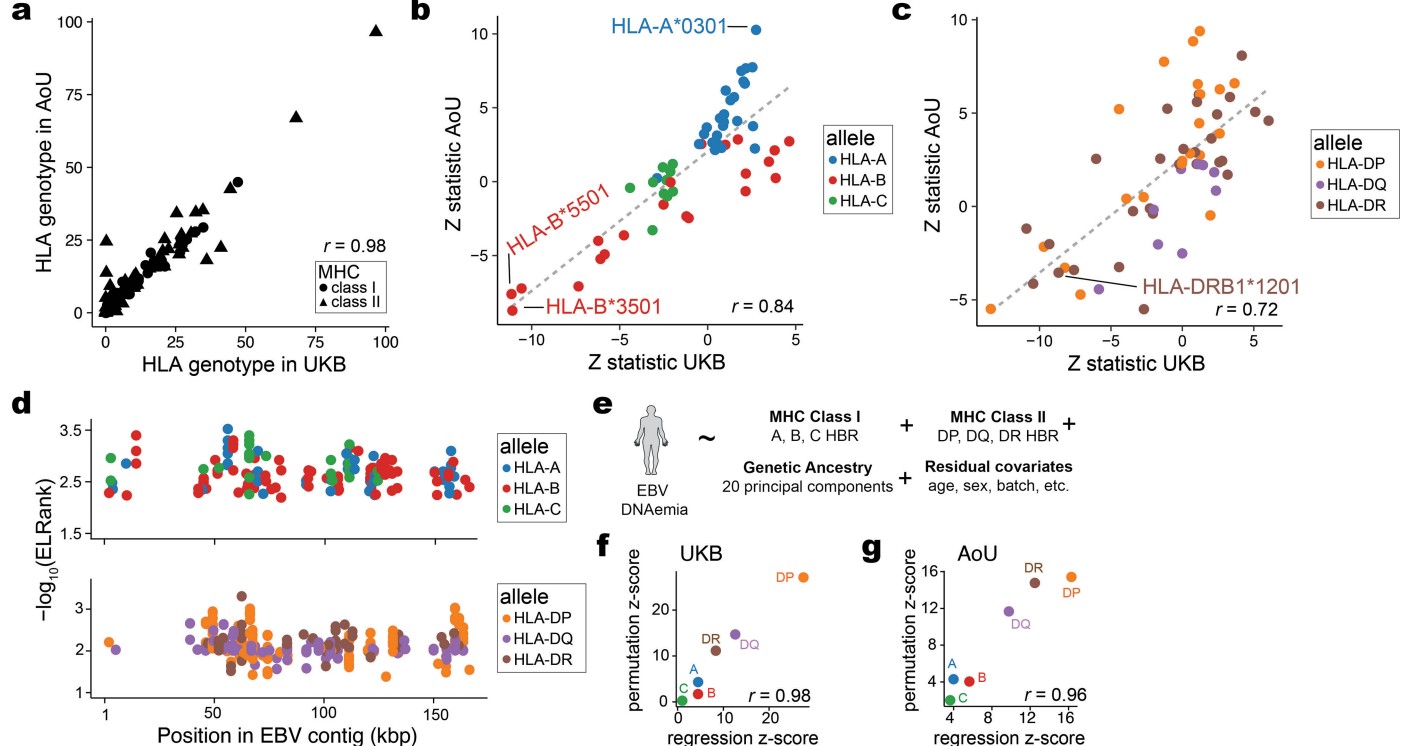

**Extended Data Fig. 6 | Supporting analyses of HLA-specific and antigen-binding associations.** (**a**) Summary of four-digit HLA genotype frequencies across both UKB and AOU individuals of NFE/EUR ancestry. The Pearson correlation of the allele frequency is noted. (**b**) Results of allele-level regression, showing the $Z$-statistic of the two-sided Wald test for individual HLA class I alleles. (**c**) Same as (b) but for class II alleles. (**d**) Annotation of the strongest predicted peptide per HLA allele for class I (top) and class II (bottom).

(**e**) Schematic of logistic regression analyses to assess peptide presentation/ binding scores as a predictor of EBV DNAemia. (**f**) Correspondence between $Z$-statistic per allele for regression and permutation statistical models for the UKB NFE cohort. The Pearson correlation between $Z$-statistic of the alleles is noted. (**g**) Same as in (d) but for the AOU EUR cohort. The silhouette in panel **e** was adapted from ref. 19, Springer Nature Ltd.

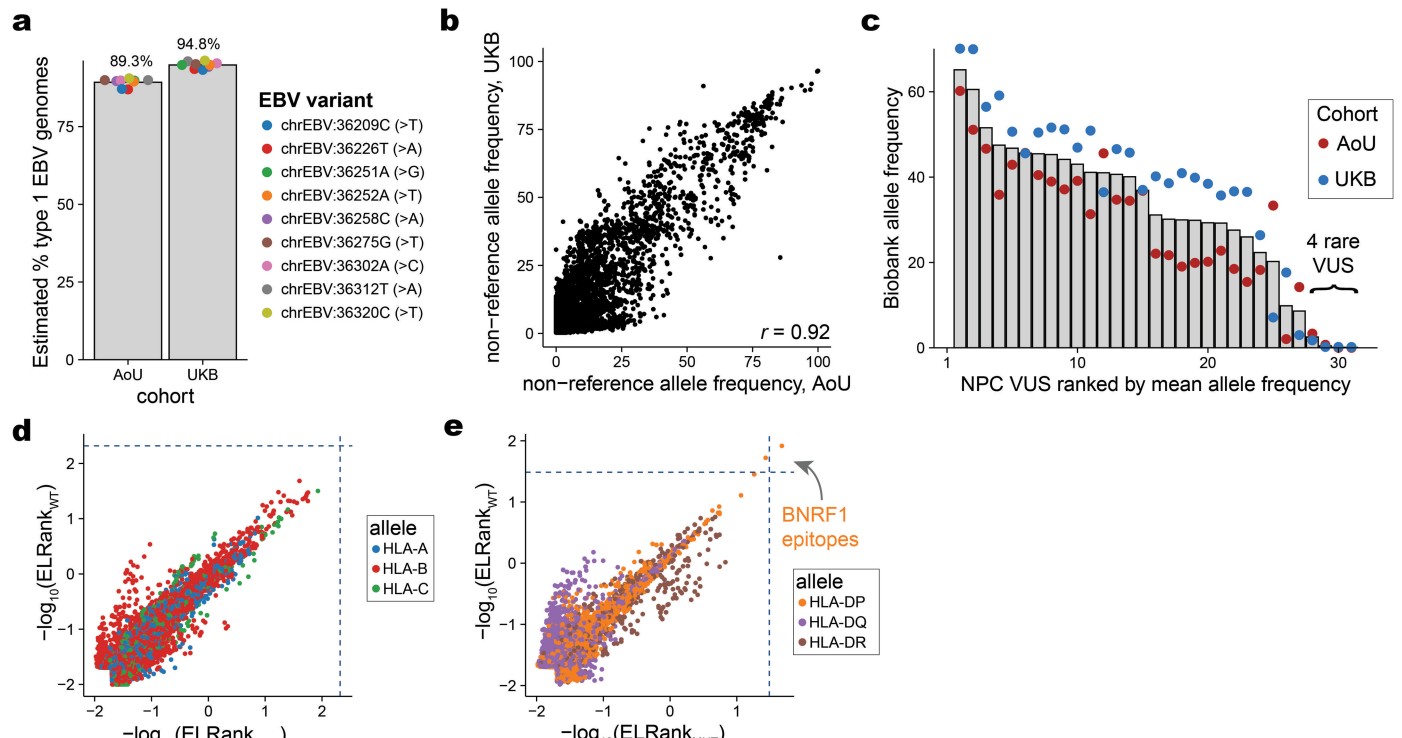

**Extended Data Fig. 7 | Analyses of genetic variation in the EBV genome.**
(**a**) Summary of 9 selected variants that discriminate between type 1 and
type 2 EBV strains. The observed allele frequency for the reference contig
(NC_007605; type I EBV) is plotted, and the corresponding type 2 allele is noted
in parentheses. (**b**) The Pearson correlation of the two allele frequencies is
noted. (**c**) Characterization of EBV variants of unknown significance (VUS)
from cohorts of nasopharyngeal carcinoma (NPC) tumors[55]. All but 4 variants
were detected at ≥10% in one or both cohorts. (**d**) Comparison of predicted
peptide presentation strength for all possible peptides and HLA-A/B/C alleles
containing one of four VUS from (c). (**e**) Same as (d) but for class II alleles. For (d)
and (e), dotted blue lines reflect the weakest epitope nominated by NetMHC
and confirmed to be bound by IEDB for class I and class II.

# Reporting Summary

## Statistics

For all statistical analyses, confirm that the following items are present in the figure legend, table legend, main text, or Methods section.

| n/a | Confirmed | |
|---|---|---|
| ☐ | ☒ | The exact sample size (*n*) for each experimental group/condition, given as a discrete number and unit of measurement |
| ☐ | ☒ | A statement on whether measurements were taken from distinct samples or whether the same sample was measured repeatedly |
| ☐ | ☒ | The statistical test(s) used AND whether they are one- or two-sided<br>*Only common tests should be described solely by name; describe more complex techniques in the Methods section.* |
| ☐ | ☒ | A description of all covariates tested |
| ☐ | ☒ | A description of any assumptions or corrections, such as tests of normality and adjustment for multiple comparisons |
| ☐ | ☒ | A full description of the statistical parameters including central tendency (e.g. means) or other basic estimates (e.g. regression coefficient) AND variation (e.g. standard deviation) or associated estimates of uncertainty (e.g. confidence intervals) |
| ☐ | ☒ | For null hypothesis testing, the test statistic (e.g. *F*, *t*, *r*) with confidence intervals, effect sizes, degrees of freedom and *P* value noted<br>*Give P values as exact values whenever suitable.* |
| ☒ | ☐ | For Bayesian analysis, information on the choice of priors and Markov chain Monte Carlo settings |
| ☒ | ☐ | For hierarchical and complex designs, identification of the appropriate level for tests and full reporting of outcomes |
| ☐ | ☒ | Estimates of effect sizes (e.g. Cohen's *d*, Pearson's *r*), indicating how they were calculated |

*Our web collection on statistics for biologists contains articles on many of the points above.*

## Software and code

Policy information about availability of computer code

| Data collection | Reads mapping to the chrEBV contig were extracted for the UKB using samtools v1.17 and for the AoU cohort using GATK v4.2.6. Four-digit HLA calls were acquired from the UKB RAP web portal. HLA genotypes for AoU were inferred via T1K v1.0.7. For both cohorts, the underlying composition of the viral genomes were determined using bam-readcount v1.0.1 on the merged .bam file of all chrEBV reads. |
|---|---|
| Data analysis | Downstream analyses were performed using bowtie2 v2.5.1, REGENIE v3.2.4 (AoU) and v3.5 (UKB), plink v1.9 (AoU) and 2.0 (UKB), ldsc v1.0.1, GenomicRanges v1.59.0, AlphaMissense v2023.hg38, cupcake v0.1.0, CIBERSORT v1.0.6, kallisto v0.50.0, Seurat v5, clusterProfiler v4.0, chromVAR v1.5.0, NetMHCpan v4.1, and NetMHCIIpan v4.3. Code to reproduce custom analyses in this manuscript is available online at https://github.com/clareaulab/ebv_biobank_gwas. |

For manuscripts utilizing custom algorithms or software that are central to the research but not yet described in published literature, software must be made available to editors and reviewers. We strongly encourage code deposition in a community repository (e.g. GitHub). See the Nature Portfolio guidelines for submitting code & software for further information.

## Data

Policy information about availability of data

All manuscripts must include a data availability statement. This statement should provide the following information, where applicable:

- Accession codes, unique identifiers, or web links for publicly available datasets
- A description of any restrictions on data availability
- For clinical datasets or third party data, please ensure that the statement adheres to our policy

The UK Biobank data are available to qualified researchers (please refer to the details at http://www.ukbiobank.ac.uk/register-apply/). The All of Us data are available as a featured workspace to registered researchers of the All of Us Researcher Workbench (https://www.researchallofus.org/). Summary statistics from the EBV DNAemia discovery NFE GWAS in UKB are available at https://my.locuszoom.org/gwas/409414/?token=6385c90400414f34b8ed17679bf1495b and have been uploaded to the GWAS catalogue (GCST90572743). No new sequencing data was generated as part of this study.

## Research involving human participants, their data, or biological material

Policy information about studies with human participants or human data. See also policy information about sex, gender (identity/presentation), and sexual orientation and race, ethnicity and racism.

| | |
|---|---|
| Reporting on sex and gender | All analyses included males and females. We report that sex was used as a covariate in many downstream analyses and association tests. |
| Reporting on race, ethnicity, or other socially relevant groupings | Both sex assigned at birth and self reported gender of individuals was collected. Sex assigned at birth was used for all relevant analyses, including only individuals where the genetically inferred sex matched sex assigned at birth. |
| Population characteristics | For the discovery cohort, the average age was 57, and 54% of the cohort was female. 94% of the cohort is of European ancestry (UK Biobank). For the All of Us cohort, adults 18 years and older who have the capacity to consent and currently reside in the U.S. or a U.S. territory were eligible. |
| Recruitment | Participants were recruited to the UK Biobank on a voluntary basis. Approx 500K individuals 40-69 years of age in 2006-2010 volunteered. Informed consent was obtained for all participants. It has previously been observed that participants are less likely to live in socioeconomically deprived areas than non-participants, and they tend to be healthier than non-participants, which may impact some of the reporting rates in comparison to what could be observed through random sampling from the UK population. Fry et al (10.1093/aje/kwx246). Recruitment of the All of Us Research Program was described in detail in "The "All of Us" Research Program", NEJM 2019; briefly individuals were recruited through direct participant enrollment or recruitment at one of >340 locations at US healthcare provider organizations or federally qualified community health centers. |
| Ethics oversight | The protocols for UK Biobank are overseen by The UK Biobank Ethics Advisory Committee (EAC), for more information see https://www.ukbiobank.ac.uk/ethics/ and https://www.ukbiobank.ac.uk/wp-content/up1oads/2011/05/EGF20082.pdf Informed consent for the All of Us participants is conducted in person or through an eConsent platform that includes primary consent, HIPAA Authorization for Research EHRs, and Consent for Return of Genomic Results. The protocol was reviewed by the Institutional Review Board (IRB) of the All of Us Research Program. The All of Us IRB follows the regulations and guidance of the NIH Office for Human Research Protections for all studies, ensuring that the rights and welfare of research participants are overseen and protected uniformly. |

Note that full information on the approval of the study protocol must also be provided in the manuscript.

# Field-specific reporting

Please select the one below that is the best fit for your research. If you are not sure, read the appropriate sections before making your selection.

☒ Life sciences ☐ Behavioural & social sciences ☐ Ecological, evolutionary & environmental sciences

For a reference copy of the document with all sections, see nature.com/documents/nr-reporting-summary-flat.pdf

# Life sciences study design

All studies must disclose on these points even when the disclosure is negative.

| | |
|---|---|
| Sample size | No pre-determined sample size was calculated for these analyses as analyses were retrospective from large cohorts. The sample sizes for genetic and phenotypic associations exceeded 490,000 from the UKB (discovery cohort) and 245,000 from AoU (replication cohort) represent the largest cohorts to date to study the genetic basis of EBV (a minimum ~50x increase from any past study), meaning our sample size was substantially larger than any published analysis to date. |
| Data exclusions | No data or individuals with successful generation of genome sequencing data were excluded from these analyses. |
| Replication | The UK Biobank cohort was used for discovery. The All of Us cohort was used for replication studies. The GWAS and PheWAS results showed largely concordant results for variants and phecodes that could be analyzed in both cohorts. For PheWAS, 87 of 141 (62%) significant |

phecodes in UKB that could be remapped to the AoU phecodes replicated in AoU (P < 0.05; OR directionally concordant with UKB statistics). For GWAS, 40,675 variants were genome-wide significant (P < 5×10⁻⁸) in UKB and passed quality control filters in AoU, of which 91.4% were replicated in AoU (nominal P < 0.05; OR concordant).

Randomization | This study is observational. Randomization was not applicable to this study.

Blinding | This study is observational, using coded de-identified data. Blinding was not applicable to this study

# Reporting for specific materials, systems and methods

We require information from authors about some types of materials, experimental systems and methods used in many studies. Here, indicate whether each material, system or method listed is relevant to your study. If you are not sure if a list item applies to your research, read the appropriate section before selecting a response.

## Materials & experimental systems

| n/a | Involved in the study |
|---|---|
| ☒ | Antibodies |
| ☒ | Eukaryotic cell lines |
| ☒ | Palaeontology and archaeology |
| ☒ | Animals and other organisms |
| ☒ | Clinical data |
| ☒ | Dual use research of concern |
| ☒ | Plants |

## Methods

| n/a | Involved in the study |
|---|---|
| ☒ | ChIP-seq |
| ☒ | Flow cytometry |
| ☒ | MRI-based neuroimaging |

## Plants

Seed stocks | N/A

Novel plant genotypes | N/A

Authentication | N/A

