## [Peer Review file · Nature]

Population-scale sequencing resolves determinants of persistent EBV DNA

Corresponding Author: Dr Slavé Petrovski

Version 1:

Reviewer comments:

Referee #1

(Remarks to the Author)

Summary: The authors describe an association study between detection of EBV viral DNA and genetic variants, disease phenotypes and transcriptional profiles using whole genome sequencing and phenotype data from two large biobanks, the UK biobank (discovery) and the All of Us cohort (replication). They first describe a novel method for extracting off target, EBV reads from WGS data and demonstrate that it can be consistently recovered from biobank scale data sets. Next, they perform PheWAS and GWAS analyses that elucidate a potential role for EBV persistence in cancer, autoimmune disorders and respiratory and conditions and show genetic variation in genes involved in peptide presentation (predominantly but not exclusively driven by HLA variation) are a major driver of EBV persistence. This role is further substantiated by investigating gene expression across a large scRNA-seq data set which confirmed a major role for antigen processing and peptide presentation. Through HLA/peptide prediction analyses, they further demonstrate that peptide presentation through HLA class II alleles are key to determining EBV persistence. Overall this is a well performed and well-presented study that has high potential for translatability. I have only minor comments/suggestions.

An obvious limitation of this study is that it restricts to only non-Finnish European individuals. Given the geographic heterogeneity in EBV outcomes, in particular some types of cancers, it would be important to understand how genetic ancestry may influence EBV persistence. Although perhaps beyond the scope of the current work, the authors should comment on whether they believe host or viral factors are likely to impact this heterogeneity. Indeed, there is opportunity in the AoU cohort to investigate this, it would be interesting to know how these persistence levels vary across various ancestry groups.

I'm also interested in the authors perspective on why the association with MS wasn't more obvious in the data. Indeed, the effect sizes per cohort are actually in opposing directions (although non-significant). Do the authors believe this lack of detection is due to power considerations or something about the underlying biology of the DNAemia phenotype?

I was similarly surprised that that HLA class II didn't come up in the serology only GWAS given the overall strong effect size and previous identifications (PMID: 26456283). Again, I suppose this could be a function of the seronegative sample size but I'm curious if the authors believe that DNAemia and seropositivity have similar underpinnings. Related to this, the statement on line 337 "Notably, EBV DNAemia, but not serostatus, is a polygenic trait" suggest the authors believe they have different architectures. I'm not convinced this is supported by the data presented.

Minor:

Line 171 seems incomplete. Was there meant to be an elaboration of the previous sentence?

I caught a couple of other typos; parentheses usage on line 231 and 235, line 233 "...shared genetic architecture *between* EBV DNAemia and ...", line 236 "rick loci", line 245 "EDs" should be DCs?

(Remarks on code availability)

I was only able to review the code relating to downstream analysis as reviewing all code would require exporting large amounts of biobank-level data. However, the code for data extraction seems appropriate. Cod for downstream analysis was

usable with sufficient instruction to run.

Referee #2

(Remarks to the Author)

The authors analyze whole genome sequence (WGS) data for EBV DNA sequence from the UKB and AoU large databases. They find elevated EBV DNA in subset of donor samples which can then be further correlated with donor molecular traits and phenotypes. The authors find some very interesting correlations, including elevated EBV DNA in individuals with biomarkers to respiratory, autoimmune and cardiovascular disease. They find correlations of EBV DNA levels with antigen processing pathways and antigen presenting cells. They use predictive viral epitopes to implicate MHC class II as a major determinant of EBV persistence.

Overall, this is an impressive and important study that attempts to correlate EBV DNA levels with various molecular signature pathways and phenotypes from the large data bases of UKB and AoU. This is a very important effort to build accurate correlations between EBV DNA levels in available blood samples with any disease mechanisms, especially those not previously recognized. Some very intriguing new connections are drawn, particularly with strong correlations between EBV and COPD (Fig. 2) or Crohn's Disease (Fig. 3). However, there remain some concerns as to the interpretation of the data and whether the computational methods are sufficient, by themselves, to draw real-world conclusions without further experimental validation.

Specific Comments

1. Primary concern is the interpretation of high levels of EBV in whole blood samples. It is not obvious that the methods used can clearly distinguish whether viral copies detected are from a few lytic cells or a larger population of latently infected cells. Only 9.7% of individuals had EBV DNA detected, but surely these individuals are mostly EBV+ and were only captured as positive during periods of reactivation?

Methods: lines 454-463. The explanation for why the EBV reads must be derived from latent EBV infection is not clear. Most individuals will have some cells that are undergoing spontaneous lytic reactivation. The methods described, as far as I can tell, do not distinguish between a latent or lytic simply by examining DNA content from short read (paired end illumina) sequencing. Knowing if the DNA was from lytic reactivation or expanded latency pool would be valuable.

EBV copy number and activity is known to fluctuate over time for most healthy carriers. It is not clear whether some or most of these detectable loads are due to transient increase in EBV due to sporadic reactivation. How might this affect the interpretation of the data and the correlations with any particular disease or genotypes?

Further related to this concern, Line 537 indicates that top EBV load is 1 copy per 1.4 cells. This must clearly indicate a very active lytic infection in a large number of cells. Such high carriers should be considered separately from merely detectable latent infections, and stratifications should be re-evaluated for disease correlations and genotypes (e.g. HLA types).

2 Drug treatments were not accounted for but could have significant effects on EBV load and reactivation.

Immunosuppressive drugs may explain some increases in EBV copy number confounding some of the correlations. How was drug treatment accounted for in this study?

3. Fig 3D is remarkable, particularly the negative PC correlation of Crohn's Disease and Ulcerative Colitis with EBV negative status. This needs to be further commented and clarified as to whether this implies that EBV activity is negatively correlated with these autoimmune diseases. If that is correct interpretation, that is different than previous findings (e.g. Ebert et al., Nature Communications volume 15, Article number: 8383 (2024)).

4. Figure 5. The HLA bound peptides from EBV are suggested by a computational prediction algorithm which has limited predictive value. There needs to be some experimental validation for this analysis prior to reaching any strong conclusions.

(Remarks on code availability)

I am not able to evaluate the computational code.

Referee #3

(Remarks to the Author)

Epstein-Barr virus (EBV) is a ubiquitous human herpes virus infecting upwards of 90% of the population. This work by Nyeo et al. seek to understand the relationship between EBV latency and a range of phenotypes using a phenome-wide association (PheWAS) approach, as well as the genetic architecture of latent EBV and examine pleiotropism with complex disease. Additionally, the authors consider HLA variation in the population studied and implications from predicted binding of EBV epitopes to HLA. To identify EBV latency in the study population, the authors use a bioinformatics approach whereby they identify reads mapped to the EBV genome from existing whole genome sequencing (WGS) data. This approach allows for extraordinary sample size by accessing the UK Biobank (discovery) and All of Us (replication cohorts), improving on sample size for previous studies of EBV in health and disease by many orders of magnitude. In doing so, they show associations between the EBV phenotype thus derived, and a number of complex traits, and show evidence for shared genetic architecture.

Overall, I find the approach very compelling, and an extremely clever use of an existing large dataset. The authors should be commended on their identification of a highly significant knowledge gap. Understanding the role that this very common virus

plays in human health is of great interest, particularly in light of studies that have shown its critical role in MS pathogenesis, for example. Beyond that, it only goes to reason that latent infection may carry unrecognized health consequences, and novel approaches to uncovering these effects are much welcome. Further, the ways in which genetic background, both human and EBV, impact the manifestation of disease pathogenesis are even more poorly understood. This work attempts to answer truly important and novel questions. Additionally, a method for doing so which can be broadly applied to already collected datasets would be very valuable. However, there are several aspects of this work that undermine the analyses and conclusions presented in the paper.

The manuscript is well-written and easy for the reader to follow. I find that the majority of data provided is high quality. The author's bioinformatic analysis of sequencing data and experimental choices are reasonable and account for methodological concerns of working with NGS data, such as handling of reads that map to repetitive regions. The quality of figures and presentation in this manuscript is very high. It is clear that the authors prioritized reducing complexity while increasing visual impact and reader accessibility and should be commended for doing so.

Major:

1. Most problematic here is the underlying assumption that the authors are detecting a molecular signature of EBV latency. To the best of my knowledge, infection with EBV always results in latent infection¹. Given this, the question arises, what, if anything biological (as opposed to sequencing/bioinformatic artifact), are the authors measuring? They repeatedly refer to the molecular trait they identify as a measure of EBV latency/EBV DNA positivity ("DNAemia") (eg lines 24, 107, 115)—however, we expect that 90% (the value given by the authors for EBV seropositivity in the cohort) will bear B lymphocytes with incorporated EBV DNA, whereas the authors find this trait in ~10% of their cohorts. They come to that figure by examining the number of reads mapping to the EBV genome after masking repeat regions that likely cross-map with human reads, and using EBV seropositivity as a "ground truth" they empirically land on a number that most strongly associates with that phenotype. While this approach feels a bit like reverse-engineering these results, I'm open to the notion that this is detecting something interesting—but not EBV latency, per se. One wonders, for example—are the authors detecting an inflammatory response, whereby B cells latently infected with EBV are dividing? This could explain, for example, the association with RA. The authors state that some associations, such as emphysema and tachycardia, as possibly attributable to the known association of smoking with EBV reactivation—however, this undermines their claim that they have detected latent, rather than active, lytic virus. In summary, it is clear that the DNAemia metric is capable of finding EBV DNA from peripheral blood samples. However, I do not have confidence from what the authors provided that this can be used to infer its latent state. I caution that unless the authors can satisfactorily provide additional evidence to support their assumption that the patterns they are capturing represent EBV latency, this paper will require thematic revision and re-evaluation of its major biological conclusions.

2. While the above approach is interesting, and the authors show concordance of their measures of EBV-mapped reads with orthogonal data including seropositivity and scRNAseq, it absolutely needs further support in even a small cohort where all measure could be evaluated concomitantly with the WGS data, to provide some concrete validation. It is impossible from the data presented to feel confident about what specific trait is being identified here.

3. The authors use demographic data to support the notion that this molecular trait is representative for latent EBV infection, including sex-bias, age-associated increase, and an increase with northern latitudes. On the latter point, the authors claim that this is consistent with findings linking EBV infection with vitamin D levels. This is problematic because the paper they cite in support of that relationship² shows no such thing, but does state "The timing of EBV infection may have significant latitudinal variation with the majority of EBV infections in more tropical areas occurring early in life.." which in fact suggests that the variation in latitude that the authors observe may be due to infection recency in the cohort. Likewise, it is clear from Extended Data Figure 1 that there a dip in frequency of this trait after middle age, with frequency again increasing after age 75. Again, this looks more like a measure that is measuring infection recency, and very possibly lytic virus, with increase later in life coinciding with known decline in immune function.

4. In sections "EBV is a biomarker of complex traits" (lines 121-159) and "Polygenic variation underlies EBV DNAemia" (lines 162-220), the authors describe the association results of the two cohorts (UKB and AofU) separately. They do not explicitly describe which results replicated in both studies. The purpose of a replication cohort (AofU) is to identify signals that reproduce in both cohorts independently, thus increasing confidence in the signal. As the results are currently written, the reader cannot meaningfully compare between the two cohorts. I would recommend re-writing these two sections by comparing the association signals of the two cohorts first, identifying the signals that are reproducible in both, and then re-writing the results to describe only those signals that reproduced. Otherwise, there is no reason to include a replication cohort.

5. I also had some concerns regarding the genetic analysis in the section: "Polygenic variation underlies DNAemia"

a. Why have the authors exclusively focused on exome variants in the human genome? Certainly, they are important, but increasingly, genetics literature emphasizes the importance of intergenic, non-coding and regulatory elements in genetic architecture of complex traits. Particularly in the context of a virus known for extensive restructuring of host cell chromatin architecture. Presumably some of the 21 independent associations from the genotyping array GWAS were of this category? Please include a description of those results as well.

b. It is unclear how the results from the array-based genome-wide association study and the exome association study were meaningfully compared. It would be helpful to add a discussion of how much overlap there was, if any, in terms of associated loci identified via these two methods. Especially when discussing the "consistent" results at the MHC (lines 183-186). Without the results from the gwas, there is no way for the reader to evaluate if they are in fact consistent.

c. How were NGS reads assembled for HLA genes in the ExWAS analysis? Due to the high levels of allelic diversity at HLA genes, simply aligning reads to a reference sequence will not sufficiently capture the correct allelic composition. There are several tools specifically designed to call HLA genotypes, why was one of these not used? The authors even describe using T1K in a downstream analysis, to infer HLA genotypes in the AoU cohort. This tool should also be applied here as well, to ensure better accuracy of ExWAS-derived HLA association signals. Especially since so many of their strongest signals seem to come from HLA genes.

d. When the authors describe their association findings across the MHC (lines 183-196, Figure 3b), they do not

acknowledge or address the well-established presence of strong linkage disequilibrium (LD) in the MHC, nor how that impacts interpretation of association signals in the area. Though the authors did calculate LD scores for three regions described in ExData Fig 3b-d, it seems they did not apply this analysis to the MHC. This is an extremely well-documented phenomenon and known technological challenge to studying the MHC and it should be acknowledged in order to meaningfully interpret association signal in the MHC.

e. Of the 690 missense ExWAS signals, 634 (92%) occur in the MHC (Extended data Table 3). That's an overwhelming proportion localizing to a single region of the genome. And many of those MHC signals occur in non-HLA genes. Both these results merit acknowledgement and discussion but get neither. Please include. It may also be related to the previously referenced strong LD in the MHC, which can inflate signal due to LD with the causal variant(s). This further emphasizes the importance of considering LD at the MHC, as it might lead to a reduction of independent associated loci.

6. In performing the GWAS on their EBV molecular trait, the authors identify the strongest signal in the HLA region. Beyond that, much of the subsequently identified genetic pleiotropy and pathway enrichment appears to be driven largely by HLA. Given that, it is inexplicable that no analysis of specific HLA allelic variation with this trait is presented. In fact, the authors have these data (HLA imputed genotypes in the study cohort), and they later use it to consider patterns of binding of EBV peptides to the HLA present in the cohorts. This latter seems an unusual and roundabout approach—why not look at binding rather to specific HLA alleles associated with the measured trait? Was this analysis performed at all? Additionally, rather than generating all possible peptides from the generic EBV reference sequence, why not focus on the peptides derived from significant EBV variants discovered in the following section “Genetic diversity in EBV sequences” to test the HLA peptide binding performance? I think this would be more meaningful than just generating all possible peptides and showing that they all have different performance. This will add functional significance to the peptide-HLA allele relationship, as well as providing additional evidence to support functional relevance of the variants described in this paper.

Minor:

1. It is well-established that socioeconomic status (SES) plays a role in timing and prevalence of EBV infection—was this accounted for in these analyses?
2. A basic demographic table, eg median age, etc, with crosstabs for EBV seropositivity would be helpful
3. Was analysis done in any non-European ancestry individuals? If not, why?
4. The authors often refer to “strong” associations, eg, EBV with “malaise/fatigue.” While I agree that the results appear to be strongly significant, I don't know that I would categorize an OR=1.27 as a strong association.
5. There is no clear definition of the metrics “EBV DNA positivity” or “binarized EBV serology” (Lines 168-169). This makes it difficult to evaluate the conclusion in the following statement (lines 169-171).
6. Line 171- Unfinished sentence
7. I was unable to find the meta data describing the results in Figure 3a. Please include a table/figure (even if supplemental) containing the results for at least the 21 independent genome-wide loci obtained from the genotyping chip.
8. Line 572: It is not entirely clear what “EBV DNA detection” refers to, in the logistic model term. Is this the DNAemia measurement? If so, please keep terminology consistent throughout the paper. If not, please define how you measured “EBV DNA detection” specifically.
9. Line 589: Leftover revision comment “(LD reference?)”
10. Line 650-651: “we removed appropriate genes from both...” Please define how these “appropriate” genes were chosen for removal.
11. Line 661-662: Please provide more detail on how you generated this “synthesis” of reads from a variety of sources as input for T1K. This is not a standard approach.
12. Line 661-662: Please label the T1K software properly, it should be all caps. Find elsewhere in manuscript and correct.
13. Line 673: Grammar.
14. Please consistently refer to the UKB and AoU cohorts in the same way throughout the paper.

REFERENCES

1. Münz, C. Latency and lytic replication in Epstein–Barr virus-associated oncogenesis. *Nature Reviews Microbiology* 17, 691-700 (2019).
2. Sabel, C.E., et al. The latitude gradient for multiple sclerosis prevalence is established in the early life course. *Brain* 144, 2038-2046 (2021).

(Remarks on code availability)

Review of code and bioinformatic workflow

README.md:

- The information regarding the use of GATK to access the bam files on AoU hosted-data is satisfactory
- Please include the “samtools view” parameters and values used to access locally stored UKB bam files, such that a user could apply the same command to their data and generate the same output you did.
 - o It is unclear what is the desired output of the bam files. Presumably fastq files?
 - o And if the desired output is fastq files, there are more steps required to generate fastq from bam than just samtools view. Please describe these commands and the parameters used.
- While I understand that due to the data sharing limitations, it is not possible to provide the actual source data for many of the analysis stages, it is still essential to describe the conceptual workflow used to go from .bam files to the input for each of the R scripts. As it stands, it is impossible for the user to make that jump. Please add a conceptual workflow to the README, that describes each stage of the analysis and the data generated at each step, up to the point of input for the R scripts.
 - o Ideally, the authors would create an example dataset from simulated short reads using both the human and EBV genome, which would allow them to share example data at each step. ArtSim (Huang et al., 2011) is a simple and convenient way of

doing this for NGS reads: <https://www.niehs.nih.gov/research/resources/software/biostatistics/art>

- Since one of the claims of the manuscript is that the authors “demonstrate how existing WGS data can derive novel molecular phenotypes, which may generalize to hundreds of viruses comprising the blood virome”, it is essential that they provide a clear bioinformatic workflow that others can reproduce from start to finish. While the R scripts are overall very clean well annotated, there is currently no way for the reader/user to know how to generate the required input files for the analysis from the bam files described in the README.

celltype-pathway-mapping/code/00_ref_project.R

- *Could not run*

- Line 4: This is not available to install via the standard R CRAN repository. Please add installation commands to script.

o Library(BuenColors)

- Line 13: Missing file- ref_path <-

"~/Dropbox/main_papers/pearson/pearson_large_data_files/input/pbmc/pbmc_multimodal.h5seurat"

o This is presumably a private directory belonging to the authors that a public user cannot access.

o There is no description of this file provided for a user to be able to provide one themselves

- Line 19: Missing file - ebv_hits <- (fread("../data/ebv_hits.txt", header = FALSE)[[1]])

celltype-pathway-mapping/code/01_genesetanalysis.R

- *Could not run*

- Line 1-7: These are not available to install via the standard R CRAN repository. Please add installation commands to script.

o library(BuenColors)

o library(clusterProfiler)

o library(ReactomePA)

o library(clusterProfiler) (This is a repeat)

o library(org.Hs.eg.db)

o library(annotables)

- Line 30: Missing file: all_hits <- readLines("../data/ebv_hits.txt")

epitope-scoring/code/01_test_sliding_peptides.R

- Successful run

- library(Biostrings): This is not available to install via the standard R CRAN repository. Please add installation commands to script.

epitope-scoring/code/02_plot_epitopes.R

- *Could not run*

- Line 1-2:

o library(BuenColors)

o library(rtracklayer)

These are not available to install via the standard R CRAN repository. Please add installation commands to script.

- Line 10: Missing input file: "../output/full_EBV_annotation.rds"

epitope-scoring/code/03_enrich_epitopes.R

- Successful run + Warning message

- Line 1-2: These are not available to install via the standard R CRAN repository. Please add installation commands to script.

o library(BuenColors)

o library(Biostrings)

Output warning messages:

```
> class1_summary$pvalue <- sapply(1:4, function(i){
+ prop.test(class1_summary$total_br[i], sum(class1_summary$total_br), p = class1_summary$prop_total[i])$p.value
+ })
```

Warning message:

In prop.test(class1_summary\$total_br[i], sum(class1_summary\$total_br), :

Chi-squared approximation may be incorrect

```
> class2_summary$pvalue <- sapply(1:4, function(i){
+ prop.test(class2_summary$total_br[i], sum(class2_summary$total_br), p = class2_summary$prop_total[i])$p.value
+ })
```

Warning message:

In prop.test(class2_summary\$total_br[i], sum(class2_summary\$total_br), :

Chi-squared approximation may be incorrect

viral-sequences/code/01_type1_type2.R

- Successful run

- Line 4: This is not available to install via the standard R CRAN repository. Please add installation commands to script.

o Library(BuenColors)

viral-sequences/code/02_overall_mafs.R

- Successful run

- Line 1, 4: These are not available to install via the standard R CRAN repository. Please add installation commands to script.

o library(Biostrings)
o library(BuenColors)

viral-sequences/code/03_top_VUS.R

- Successful run

- Line 2: This is not available to install via the standard R CRAN repository. Please add installation commands to script.

o library(BuenColors)

viral-sequences/code/11_munge_VEP.R

- Successful run

viral-sequences/code/13_epitopes.R

- Successful run

Referee #4

(Remarks to the Author)

I co-reviewed this manuscript with one of the reviewers who provided the listed reports.

(Remarks on code availability)

Referee #5

(Remarks to the Author)

Nyeo et al present a piece of technical work undertaking large-scale analyses of Epstein-Barr virus (EBV) signatures in cohorts of large biomedical databases, such as UK Biobank and All of US. In general, the work is well done in terms of analyses and the methods are valid, while the sample sizes are impressive. However, one major limitation of the work is that the results are not novel, nor particularly impactful. Most of the results described are well-known associations of EBV with diseases, and if there are additional novel results, the authors have not presented these clearly enough. The second major issue is that there is hardly a discussion section that could help the reader put the findings into context and interpret them. Not providing adequate context is a recurring problem in the manuscript, frequently having no relevant information nor rationale provided. As just one example, the epitopes section is not well explained (also Figure 5), and therefore following the results is not straightforward. As a second example, how and why dendritic cells (line 244, figure 4) are involved in EBV latency; the authors could provide more context and discuss more on why the findings are interesting for EBV biology and disease. For more examples, see specific points below. Sometimes it is also unclear whether the authors are using results from alternative datasets or indeed reanalysing these data themselves.

In general, the paper does not seem written for a general audience, and I think the fact that there is no actual discussion, while the methods are better written than the results presentation further highlights this.

More specifically, I have the following major points:

1) Page 3, Lines 36-68: Describe the meaning/importance of EBV latent infection and provide information on relevant EBV biology.

2) Lines 106-107: Given that 90% of the population are EBV-seropositive, but EBV DNA is detected in only 9.7% of the UK biobank's (and 11.9% in AoU) participants, what is the importance of this detection? In the methods (Line 462) the authors state that "We interpret our measure to reflect the tail of latent viral retention among individuals for whom this is highest", however they do not offer a view of what this highest latent EBV retention really means, and this should be contextualised in the results and discussion.

3) Relevant to the point above, the authors should comment on whether the ratio of EBV-infected B lymphocytes versus non-infected B lymphocytes is what is driving the EBV DNA detection and its associations with disease and comment on its importance.

4) By using DNA-seq data, the authors cannot differentiate between EBV latent or lytic infection, so they interpret everything as EBV latent infection. However, arguably a proportion of this EBV DNA signature will originate from lytic infections in individuals with various illnesses, that cannot be separated from latent cases from the data. Lytic infections could be present due to primary infection (ie active infectious mononucleosis) and also reactivation which can happen in individuals under stress, illness, or immunosuppression. For example, while the authors acknowledge the presence of mononucleosis cases (lines 153-159), they fail to address how the distinction between lytic and latent infections could impact their conclusions. This presence of lytic infection is further demonstrated by the authors' own results, see Line 132: "Significant quantitative associations (n = 156) included detection of two EBV antigens". One of these two antigens is ZEBRA (Extended Figure 2a), which is a marker of lytic acute infection.

The authors should therefore acknowledge and discuss this, namely that the EBV DNAemia signature is not only due to latent infection but rather in some instances due to lytic infection. Given this and the known literature around EBV

reactivation in several settings (under stress, several health conditions, immunosuppression), they should reflect on why they detect only 1 transcript in their single cell RNA-seq data from the 1000 healthy blood donors cohort. I think the most likely explanation is that this was a cohort of healthy blood donors who already had their primary EBV infection at younger age (cohort individuals older than 19 years old) and who did not have EBV reactivation at the time of sampling as they were donating blood while healthy – in contrast the UK Biobank and All of US have large patient groups where reactivation could indeed occur. Additionally for the single-cell RNA-seq study of the 1000 blood donors, was the library prep/sequencing protocol suitable to retain non-human RNA transcripts, or could there be a bias introduced there as well?

5) Line 101-102: Relevant to the point above, the authors should clarify whether they undertook complementary analyses for detecting EBV transcripts? It seems that way from methods but in the main text it is ambiguous whether this is novel analyses or published in the study referenced.

6) Lines 120-159: The clinical findings of associations with various illnesses are not novel, as already stated the technique is interesting and the sample sizes are noteworthy, however if there are novel findings, they are not demonstrated properly.

7) The Mustafa et al. reference in the abstract does not support the statement for hundreds of viruses comprising the blood virome. In fact, in the Mustafa et al study cited in the abstract, the authors state explicitly that out of 94 viruses, only 19 are not contaminants. From Mustafa et al abstract: "... we mapped sequences to 94 different viruses, including sequences from 19 human DNA viruses, proviruses and RNA viruses... The remaining 75 viruses mostly reflect extensive contamination of commercial reagents and from the environment". In any case, the work undertaken in the Nyeo et al study cannot be replicated for the majority of viruses that do not establish latency - the authors should discuss the instances where this would be feasible e.g herpesviruses.

8) Line 49-56: "Beyond its role in human disease...caused by infection". If there were important findings relevant to EBV from these large genetic cohorts what were these – the authors do not make it clear. Instead, the authors should use this space to clarify further and emphasise EBV contributions to human mortality and morbidity.

9) Line 93: What is the novel molecular phenotype?

10) Line 125-126: How did the authors select these phenotypes? Were these all the available phenotypes in the UK BioBank or did some selection take place?

11) Line 126 onwards: the authors should provide more context for the significant associations they detect and replicate in the two cohorts. Relevant to that, in Figure 2a: how did they choose which associations to label, including both relevant and not-relevant EBV conditions. Furthermore, the associations in the respiratory group seem more pronounced compared to other categories, what is the authors' view on this?

12) Line 161, Section Polygenic Variation: As it stands, the rationale for and the findings from this work are unclear. Do the authors claim they are showing that genetic variation is the reason for individuals to respond differently to EBV infection? Relevant to that, there is really no discussion section to contextualise any of the results.

13) Line 222-237 – Pleiotropy with complex diseases. This section is difficult to follow, and the results do not make sense as presented. As an example, Figure 3b is unclear, what has been done and what exactly does it show? What do the p-values refer to? Also is Crohn's and ulcerative colitis "seronegative" because there were no associations found with EBV DNA viraemia? Finally, what is "latent EBV" in this figure – is it a category in UKB or something the authors have defined?

14) Line 245 – what are EDs?

15) Line 256: This is not a new conclusion

16) Line 321: What do these EBV genes do? What is their function, does it make sense that these seem to drive NPC oncogenesis. I think these EBV genes are known to be associated with nasopharyngeal carcinomas.

17) Page 24: The threshold the authors used to define "EBV-positive" individuals is 1 pair of reads (300bp). This could potentially arise from sequencing run contamination and is borderline positive. How does the EBV prevalence and all significant associations change with a slightly higher threshold, for example 2 or 3 pairs of EBV reads (4-6 reads)? The

authors should include information on such an analysis, as if the positivity rate remains similar and all the associations still hold this will show the threshold the authors used is robust enough - however if findings change a lot, the authors should reflect on the reason.

18) Relevant to the above point, do the authors find any EBV positive individuals by sequencing, who are at the same time EBV seronegative? This would be a good "sanity check" for the validity of their approach and interpretation of their findings.

19) Figures 1e and 1g are not intuitive. Why in 1g, the odds ratio for EBV genome abundance and EBV seropositivity is lower for greater genome abundance?

20) Line 102: "A cutoff of 1.2 viral genomes per 10^4 human cells yielded the strongest concordance with seropositivity (OR = 82.2, $P = 2.2 \times 10^{-16}$; Fig. 1g)." - this does not make a lot of sense, why do 2 EBV genome per 10K human cells have lower concordance with seropositivity? Also does seropositivity have a quantitative unit and is not binary? Please provide more information.

21) Line 481: What EBV antibody is used in the UK biobank? Is it the one that shows prior infection (IgG)? Are there further antibodies included such as EA-IgG which indicates EBV reactivation - please provide more information.

22) Line 278 "Further, we observed that predicted immuno-dominant peptides were depleted in latency-associated EBV genes specifically for MHC class I peptides, reflecting potential viral evolution to evade host immunity during latency – please provide a reference for this statement.

Minor

1) Line 171, "Namely" is not followed by a sentence.

2) Figure 1 d) clarify how the 18 infectious agents were chosen (only this information was available in UK Biobank or something else?)

3) Figure 5 line386/ There is an extra dot at the end of the sentence for point e).

(Remarks on code availability)

I reviewed the github repository but did not attempt to install and run the code.

The code seems well organised, with different directories for the different sets of analyses, with documentation and a top-level README file. By design, the individual-level data cannot be shared publicly in the github repo so not all analyses can be replicated, however this is understandable and expected due to data sharing agreements. The authors provide contact details for researchers who wish to perform similar analyses.

Referee #6

(Remarks to the Author)

I co-reviewed this manuscript with one of the reviewers who provided the listed reports.

(Remarks on code availability)

Version 2:

Reviewer comments:

Referee #1

(Remarks to the Author)

I appreciate the authors careful consideration of my comments. I have no further comments and support publication of the revised manuscript.

(Remarks on code availability)

Referee #2

(Remarks to the Author)

The revised manuscript has added significant new analyses and remains an important study. However, the revision may have compounded some of the concerning issues by making stronger, yet not fully substantiated claims.

Major concern include:

1. While the correlations of the many diseases with EBVemia is curious, the notion that EBV is potentially causative for 242 significant disease indications seems very unlikely and inconsistent with most other epidemiological studies. The authors should think of other, non-mechanistic reasons that this correlation exists, and may reflect a general state of immune insufficiency.
2. The authors appear to have made even stronger claims that the DNAemia reflects "latent" infection only. For EBV, the terminology of latency is complicated, as latent reservoirs likely reflect on-going lytic activity and repopulation of latently infected B-cells. It is not yet known, and not shown by any of the data, whether increased EBV DNAemia may be due to increase lytic activity in some reservoir (e.g. oropharynx). This could reflect important differences in underlying biological mechanism for any disease correlation. For this reason, I think the manuscript is better served with a more conservative and cautious use of the term "latent".
3. Are individuals that have high DNAemia consistently high over time in longitudinal studies. While some longitudinal data was provided for serology (reviewer figure 7), it was not clear whether a similar longitudinal study was available for EBV DNAemia. This should be clarified as individuals with high EBV DNA loads may reflect recent reactivations in some anatomical locations that resolve over time, explaining why RNAseq rarely detects a tissue-resident reactivations. Similarly, the state of EBV serology may also not reflect a chronic reactivation status.
4. The revised Introduction is more problematic since it more emphatically concludes that the EBV state is "Latent" and also that the reservoir is in the PBMC, which may not be correct, as there is reservoir in lymphoid tissue in oropharynx and elsewhere.
5. Regarding Reviewer Fig. 5. While the computational simulation supports that possibility that the detection levels are consistent with latent EBV at 1 in 104 per cell, the counter-argument is that most healthy individuals (~90%) should be detected, while that was not the case. The variation in detection is potentially interesting, but the authors do not consider the longitudinal variation of EBV load, whether from increase latently infected cells or lytic activation in oropharynx and repopulation of the latent reservoir. This is likely a normal cycle that may be perturbed in some individuals and disease states. While the overall analyses and correlations may hold, the underlying mechanism and interpretation of the data is open to question. The authors should have a better appreciation for the variation in EBV load over the normal course of individual life-time, including age and environment related changes.
6. The longitudinal data shown in Reviewer Fig. 7 may further support the sporadic nature of the EBV DNA copy number and transcript detection. While this may not nullify the overall value of the study, it may change the interpretation of the data.
7. Given the new findings shown in Reviewer Fig. 10 that immunosuppressive drugs may contribute to EBV DNAemia, how might that affect the overall conclusions. Rather than the diseases associated with the EBV DNAemia, are the drug treatments for those diseases contributing EBV copy number control?
8. The authors conflate latency with the lack of viral transcripts, but this is not correct. Viral latency in LCLs is a type III latency with significant RNA, including very high levels of some non-coding RNAs, such as EBERS. Many of these RNAs are not readily detected in RNAseq due to technical issues, that may include lack of non-polyadenylation or other reasons not yet clear. The authors may want to suggest that the majority of latency is type 0 latency, where no viral RNA is expressed. This is possible in memory B-cells in peripheral blood. Either way, the authors should have a better understanding of EBV latency types.
9. Extended Figure 3 and the longitudinal study (4H). How many individuals could be followed for how many time points? While the overall study is fine, the authors over-interpret the EBV latency issue. EBV virus is frequently detected in saliva of most individual at regular intervals, indicating virus is chronically and perpetually reactivating in the oropharynx and replenishing the latent reservoir. It remains unclear to me whether some individuals have long-term chronic high loads of EBV in PBMC, or if these loads vary over time with those having very high loads correlating with poor immune control and a vast number of other disease indications?

Other comments

Line 22. "persist for a lifetime in peripheral blood..." is misleading

Line 26. "biomarker for latent EBV infection." Is also misleading, as it is a biomarker for EBV viral load, which is a complex sum of both latent and sporadic lytic activity.

Line 169. "0 individuals with EBV DNAemia had seroconversion or seroreversion over longitudinal samples." Understood that serology does not change over time, but it is surprising that the same individuals remain stable DNAemia over longitudinal samples. This should be stated more clearly- that the DNAemia individual remain at the same DNA copy number over time. If this DNAemia is not stable over time for the same individuals, this should be stated more clearly.

(Remarks on code availability)

I was not able to review the code and not my area of expertise.

Referee #3

(Remarks to the Author)

The authors provided an extremely thorough revision and response to the review, which has substantially improved the paper.

On the question of whether the authors measure "EBV DNAemia" is measuring latent EBV infection, they provide quite a lot of additional analysis, which is encouraging. I agree that the data support the notion that what is being measured is not lytic virus. However, I still think that it is confusing, and inaccurate, to describe it simply as a measure of latency, which implies

NO latency in the remaining ~90% of subjects. The authors do state that they believe that they are measuring the tail end of the distribution of latent virus DNA, but at other times again imply that only those that are detected are experiencing actual latency. I think it's important to clarify that while nearly all of the subjects who are EBV seropositive will have some latent virus, those with measurable EBV DNA are experiencing, for whatever reasons, a higher latent load.

It would be helpful for the authors to double check the databases they downloaded the RNA data from (GTEx, OneK1K) and ensure that the publicly accessible transcript data have not been pre-filtered to exclude non-human sequences.

Similarly, they should double check that the version of the hg38 reference sequence used to make transcript alignment bam files does include the chrEBV. If they have already done this, it would be easy to add a statement to the manuscript. Line 757-759 could be clarified to explicitly address this in their methods: "In GTEx, we obtained the mapped alignment files (.bam) from both WGS and RNA-seq of PBMCs for 681 donors, in which both were profiled for each individual. We downloaded .bam files from Anvil after obtaining permissions through dbGaP and subsequently extracted chrEBV for all sequencing datasets."

The manuscript is not completely clear in the use of EBV DNA+, EBV DNAemia and EBV UMIs (which has now been introduced). Perhaps minor, but they seem to be used interchangeably? Some clarification here would be helpful. "Using this threshold, we classified 47,452 (9.7%) individuals as EBV DNA+ for subsequent analyses (Extended Data Fig. 1c). Though the proportion of individuals with EBV DNAemia (9.7%) is lower than the rate of seropositivity, we interpret our measure to reflect the tail of individuals for whom persistent EBV DNA is highest"

I appreciate the reviewers examining the question of whether increased numbers of B cells could be responsible for the increased EBV DNA load. From their additional analysis, it seems, yes. The fact that this doesn't appear to be due to active infection seems to me beside the point-increased numbers of B cells may simply be indicative of an inflammatory response. While I appreciate the authors' reluctance to include these data with no replication set, I think that it provides important context and could be mentioned and included in supplemental data.

I appreciate that the authors performed a full regression on the imputed HLA data. However, it's unfortunate that these data are not really discussed or highlighted here, given the clear importance of these variants on the trait in question. It's very difficult to even discern what the strongest associations are here; the extended data table where they are presented is not even sorted in any obvious way, not by p-value or even by allele name. Also, while not major, the allele names are not shown correctly anywhere that they are given, where we should be seeing two colon-delimited fields at this resolution.

Last, it seems the authors do not further interpret the ~600 exWAS loci that mapped to the MHC, many of which occur in non-HLA genes. It looks like they chose to just leave it open ended with their statement: "Specifically, as the MHC locus is a hyper-polymorphic region with strong linkage disequilibrium, ascertaining causal variants at this locus is particularly challenging⁵³. Hence, we first focused our efforts on interpreting associated loci outside the MHC locus." This is a fair compromise, but it is a bit disappointing.

(Remarks on code availability)

The authors addressed all computational/reproducibility concerns. The inclusion of a Terra workflow that generates EBV DNA counts for 1000GenomesProject individuals was an excellent addition and significantly increases usability.

Referee #4

(Remarks to the Author)

I co-reviewed this manuscript with one of the reviewers who provided the listed reports.

(Remarks on code availability)

Referee #5

(Remarks to the Author)

Nyeo et al have revised their manuscript to include additional analyses and rewrite sections that lacked explanations and context. I am satisfied with the revised paper, as the authors have done great work in addressing my (and other reviewers') comments, resulting in a much stronger manuscript.

Some minor typos:

Line 155: "same" presumably missing from "the same sequencing workflow"

Line 709: "with a 10% forced 0 rate for 500,00" : should be 50K ie mistake in placement of comma.

(Remarks on code availability)

Referee #6

(Remarks to the Author)

I co-reviewed this manuscript with one of the reviewers who provided the listed reports.

(Remarks on code availability)

Version 3:

Reviewer comments:

Referee #2

(Remarks to the Author)

The authors have provided an excellent revision and comprehensive rebuttal. The revised manuscript is substantially improved and provides a compelling correlative analyses of high EBV viral load in PBMC with various inflammatory disease and gene loci. The study is of great interest to the EBV community, and others studying inflammatory disease and GWAS studies of large databases.

One minor comment:

Lines 143-145.

In memory B cells, EBV predominantly remains in a transcriptional state (termed latency 0) with a limited viral gene expression program¹¹. In contrast, viruses in lytic reactivation express a wide range of viral proteins that enable immune evasion¹¹.

While EBV type 0 latency may be one common state for EBV in some resting memory B-cells, other more sensitive methods find some viral transcripts in memory B-cell subpopulations in healthy donors. The failure to detect EBV transcripts in RNAseq datasets may be due to some technical issues relating to the failure to capture low level EBV transcripts in RNAseq libraries. The point being that the lack of EBV reads in RNAseq datasets does not necessarily indicate a type 0 latency. The authors should be cautious to not over-interpret the EBV latency transcriptional state, and concluding type 0 latency is not fully justified based on the existing data.

(Remarks on code availability)

Referee #3

(Remarks to the Author)

The authors have thoroughly responded to all critiques, and the manuscript is substantially improved. I have no further comments and commend the authors for their responsiveness to all reviews and for this interesting work.

(Remarks on code availability)

Code was previously reviewed and found to be adequate

Referee #4

(Remarks to the Author)

I co-reviewed this manuscript with one of the reviewers who provided the listed reports.

(Remarks on code availability)

Dear ,

Thank you for the very prompt handling of our manuscript, including the constructive reviewer feedback. We were encouraged by the positive assessment of our work from all reviewers, and we greatly appreciate the feedback that has meaningfully improved the presentation quality and interpretation of our manuscript. While the core findings are consistent between versions, in the revised document, we expand on the contextualization and interpretation of our EBV DNAemia biomarker. We emphasize the following points that reflect our substantially revised manuscript:

- At the suggestions of all reviewers, we have included additional analyses and text to explain the nature of the EBV DNAemia biomarker, including an improved definition of the latent nature of the detected viral nucleic acids. Our analyses provide compelling evidence that the EBV DNAemia that we observe is **not associated with viral RNA** (revised **Extended Data Fig. 3**), which would be present during reactivation. Further, we provide novel simulations and longitudinal serology analyses supporting that serostatus remains stable for EBV DNAemia, and that there are no significant differences in serology levels for latent EBV DNA carriers over time. In sum, we present a multitude of new analyses that corroborate our interpretation of viral DNA that can be explained by **latent** rather than lytic EBV infection.
- We expand and emphasize points of novelty for our phenotypic-association studies (revised **Fig. 2 + Extended Data Fig. 4**), including highlighting traits with known but limited evidence of associations with EBV (e.g., depression; chronic fatigue), as well as showcase the utility of our approach to power associations between rare/understudied diseases (e.g., neuromyelitis optica).
- At the suggestions of Reviewers 1 and 3, we have included genetic analyses of non-European populations (revised **Fig. 3 + Extended Data Fig. 5,6**). In doing so, we recover three additional genome-wide significant hits at biologically plausible loci (*BIM*, *TERT*, *GSDMB*) as well as an additional 20 loci by combining diverse genetic ancestries between UKB and AoU (45 total underlying loci). These results motivate the continued study of the blood virome in diverse populations worldwide using our approach of retrospective quantification of viruses from WGS data.
- We have greatly revised our analyses and conclusions regarding the MHC associations (revised **Fig. 5 + Extended Data Fig. 8,9**), including the characterization of individual alleles associated with EBV DNAemia. We observed that our associations with the EBV DNAemia biomarker are highly reproducible between UKB and AoU, collectively reflecting a more comprehensive and consistent characterization of the MHC locus in these large biobanks.
- We have substantially revised and documented all analysis code to reproduce our findings, including the addition of a public resource from the 1000 Genomes Project (1000G). The WGS data from 1000G allowed public access, such that we could demonstrate the extraction and quantification of EBV DNA for downstream analyses. These resources have been made fully available on GitHub.

Overall, our revised version now contains **5 main text figures and 9 Extended Data Figures**. The changes to the main manuscript text, as well as our responses to the reviewer comments, are noted in blue. We appreciate the opportunity to strengthen the manuscript and look forward to any additional feedback.

Sincerely,
Slavé, Ryan, and Caleb
on behalf of all co-authors

Overall changes impacting reviewer feedback

One recurrent question from all reviewers concerned the nature of the EBV DNA measure from WGS data. In our revised version, we have added a substantial section characterizing the evidence behind why we conclude that the viral DNA is predominantly from a latent viral infection (as opposed to a lytic infection) and how this interpretation impacts our work. As our sample size exceeds 735,000 individuals, there will undoubtedly be cases that were experiencing lytic EBV infection during the time of sample collection (either from a primary infection or from viral reactivation). We have made a concerted effort to both characterize the EBV DNA systematically. However, we still qualify that exceptions will exist using these large sample sizes. We summarize these analyses and interpretations below:

New analyses supporting interpretation of latent EBV DNA

- 1. Analyses of 1,663 blood-derived transcriptomes show a total of 4 EBV-derived reads.** We outline detailed analyses of the GTEx cohort (matched WGS and RNA-seq) and OneK1K cohort (large scRNA-seq cohort with additional control analyses) to show that EBV transcriptional activity is rare.
- 2. Additional analyses from the UKB cohort, including longitudinal serology and simulations of read copy number, support a stable biomarker.**

Ultimately, due to the strength and conclusivity of these analyses, we have elected to keep the word “latent” in our title and abstract, as we believe this is scientifically accurate. Nevertheless, we have rewritten sections throughout the document to qualify the nature of the viral DNA that we are detecting.

Substantial text changes

- 1. We have added a new section entitled *EBV DNAemia likely reflects latent viral infection* that shows the results of these new analyses and provides context. [revised manuscript Lines 140-174]**
- 2. We have substantially revised the ending of the paper, including a *Discussion* paragraph focused on the interpretation of the EBV DNAemia measurement. [revised manuscript Lines 450-495]**

Interpretation of the EBV DNAemia biomarker and its impact on our results

Our genetic association results likely represent conservative estimates. There is almost certainly some degree of EBV DNA that results from recent acute infections, DNA contamination, or other confounders that would obscure the EBV DNAemia phenotype in our study. However, we contend that such edge cases in the context of a large population sample would weaken the underlying signal rather than be a systematic confounder, resulting in spurious associations near loci that are consistent with immune and anti-viral responses. We provide extensive new analyses to estimate contamination rates (which appear low; **Extended Data Fig. 1b**) and characterize potential acute infections (which also seem to be minimal; **Extended Data Fig. 3**). Even in the presence of such rare edge cases, the results that we report in this manuscript, which include a total of 45 distinct genome-wide significant loci across our cohorts and 87 replicated PheWAS associations, are likely the strongest associations. We anticipate that future work that refines our methodology may further identify loci and traits with greater statistical power.

We appreciate the careful consideration of these key results and discuss our specific data points and individual responses in the pages that follow.

Referee #1 (Remarks to the Author)

Summary: The authors describe an association study between detection of EBV viral DNA and genetic variants, disease phenotypes and transcriptional profiles using whole genome sequencing and phenotype data from two large biobanks, the UK biobank (discovery) and the All of Us cohort (replication). They first describe a novel method for extracting off target, EBV reads from WGS data and demonstrate that it can be consistently recovered from biobank scale data sets. Next, they perform PheWAS and GWAS analyses that elucidate a potential role for EBV persistence in cancer, autoimmune disorders and respiratory and conditions and show genetic variation in genes involved in peptide presentation (predominantly but not exclusively driven by HLA variation) are a major driver of EBV persistence. This role is further substantiated by investigating gene expression across a large scRNA-seq data set which confirmed a major role for antigen processing and peptide presentation. Through HLA/peptide prediction analyses, they further demonstrate that peptide presentation through HLA class II alleles are key to determining EBV persistence. Overall this is a well performed and well-presented study that has high potential for translatability. I have only minor comments/suggestions.

We sincerely appreciate the positive assessment of our study execution and presentation, as well as the potential translational impact of the work.

An obvious limitation of this study is that it restricts to only non-Finnish European individuals. Given the geographic heterogeneity in EBV outcomes, in particular some types of cancers, it would be important to understand how genetic ancestry may influence EBV persistence. Although perhaps beyond the scope of the current work, the authors should comment on whether they believe host or viral factors are likely to impact this heterogeneity. Indeed, there is opportunity in the AoU cohort to investigate this, it would be interesting to know how these persistence levels vary across various ancestry groups.

We agree with the thoughtful suggestion to include analyses of the different genetic ancestries available. We have repeated a multitude of downstream analyses to expand our associations beyond only individuals of European ancestry.

- In UKB, these include: Ashkenazi Jewish (ASJ; $n = 2,629$), East Asian (EAS; $n = 2,107$), South Asian (SAS; $n = 9,119$), Hispanic or Latin American (AMR; $n = 616$), and African (AFR; $n = 8,145$) ancestries.
- In AoU, these include: African (AFR; $n = 56,911$), Admixed American (AMR; $n = 45,034$), East Asian (EAS; $n = 5,706$), Middle Eastern (MID; $n = 942$), and South Asian (SAS; $n = 3,217$).

Reviewer Figure 1 summarizes the percent EBV DNAemia abundance in these populations, which is now reported in the respective Extended Data Figures associated with each biobank.

Reviewer Figure 1 (manuscript Extended Data Fig. 1j,2e). Rates of EBV DNAemia per biobank stratified by genetic ancestry. Error bars show standard error of the mean (SEM).

Though the low sample sizes of non-European populations may hinder the power to analyze these populations in isolation, we determined that a meta-analysis framework combining the summary statistics across different ancestries was a valuable approach. Using donors from these additional five genetic ancestries in UKB, we conducted additional multi-ancestry genetic association analyses. **Reviewer Figure 2** shows a Manhattan plot from the meta analysis, revealing three additional loci that surpass genome-wide significance.

Reviewer Figure 2 (manuscript Extended Data Fig. 6a). Meta-analysis of six total ancestries in UKB. A Manhattan plot of all associated variants is shown, highlighting three regions where loci did not reach genome-wide significance when analyzing only UKB NFE individuals.

Reassuringly, the additional loci that surpass genome-wide significance were association loci near *BIM*, *TERT*, and *GSDMB*, all genes with plausible biological involvement in immune regulation and viral latency, as we discuss in the text:

[Page 9, Lines 302-306] We identified loci that were genome-wide significant in the meta-analysis but not the NFE-only GWAS, including variants near *BIM*⁷⁵ and *GSDMB*⁷⁶ (Extended Data Fig. 6a). These genes regulate apoptosis during both innate and adaptive immune system development and function. We also observed variants overlapping the *TERT* locus, supporting evidence for an interaction between EBV and telomere regulation to preserve latency⁷⁷.

Motivated by the additional hits that our meta-analysis framework could uncover, we have also completed a meta-analysis spanning a total of 9 populations, including the 6 noted above for UKB and an addition 3 from

AoU, including individuals of EUR ($n = 131,938$; AoU-EUR), AFR ($n = 56,911$; AoU-AFR), or AMR ($n = 45,034$; AoU-AMR) ancestry. **Reviewer Figure 3** summarizes our final additional GWAS included in the revised version, which recovers 20 new hits that reach genome-wide significance.

Reviewer Figure 3 (manuscript Extended Data Fig. 6b). Meta-analysis of nine total populations combining UKB and AoU. A Manhattan plot of all associated variants is shown, highlighting 7 of 20 regions where loci did not reach genome-wide significance when analyzing only the UKB-NFE cohort.

We note that a full summary of implicated loci across all three GWASes, including relative annotations of ExWAS values, replication between UKB (as discovery) and AoU, and relative nearby genes, is contained in a revised **Extended Data Table 4**. A summary of the additional loci uncovered by this last additional GWAS is added in the text:

[Page 9, Lines 310-312] *These loci included associations near immune-associated transcription factors (BCL11A, MYC, ETS1) and surface receptors (CD160, KLRC1, TRAF3), underscoring the increase in statistical power by combining EBV DNAemia GWAS across heterogeneous populations and geographical origins.*

In summary, we sincerely appreciate this excellent suggestion to examine EBV DNAemia across these distinct populations. As the results convey, there is demonstrable signal-detection utility of extending genetic associations to more diverse populations.

I'm also interested in the authors perspective on why the association with MS wasn't more obvious in the data. Indeed, the effect sizes per cohort are actually in opposing directions (although non-significant). Do the authors believe this lack of detection is due to power considerations or something about the underlying biology of the DNAemia phenotype?

Upon a more extensive literature review, our lack of association seems consistent with other studies. Among patients with MS, prior work has shown that EBV DNA is detectable in only 26.4–81.3% of patients' peripheral blood mononuclear cells (PBMCs) or whole blood samples². This stands in stark contrast to the ~100% seropositivity among MS patients – suggesting that EBV DNAemia likely is a distinct biomarker.

Due to the large sample size and relatively high occurrence of MS in these biobanks ($n > 1,000$), we do not necessarily lack the power to detect an association between MS and EBV DNAemia if a strong relationship exists. However, we note that other groups have failed to identify an association between prior EBV infection and MS in UKB, despite the association occurring in other biobanks³. In our case, the documented absence of EBV DNA in MS cases (via both PCR and sequencing), combined with the overall high sample size, suggests that we are powered to detect an association if it exists and is quantifiable in UKB.

Ultimately, our interpretation is that there is a component of shared heritability between MS and EBV DNAemia (which is reflected in our formal analysis in **Fig. 3d**). However, our results indicate that there is another axis of variation that is distinct. While this is purely speculative, we have considered (guided by the noted opposing direction) that EBV DNAemia is the consequence of a poor immunological response against the virus. MS, on the other hand, may be the result of a hyper-effective immune response to EBV. Further basic science work will likely help unpack this enigmatic relationship.

I was similarly surprised that HLA class II didn't come up in the serology only GWAS given the overall strong effect size and previous identifications (PMID: 26456283). Again, I suppose this could be a function of the seronegative sample size but I'm curious if the authors believe that DNAemia and seropositivity have similar underpinnings. Related to this, the statement on line 337 "Notably, EBV DNAemia, but not serostatus, is a polygenic trait" suggest the authors believe they have different architectures. I'm not convinced this is supported by the data presented.

We appreciate the critical thought on this statement. In the mentioned study (Hammer *et al.* 2015⁴), the authors reported results from a GWAS on **quantitative** IgG levels for EBV antigen EBNA-1, finding significant associations with the HLA class II locus⁴ ($n = 2,363$ immunocompetent individuals). In contrast, our GWAS utilized a composite **binarized** serostatus based on the presence of antibodies to four EBV antigens (VCA p18, EBNA-1, ZEBRA, and EA-D).

For completeness, we reproduce the quantitative IgG GWAS for the VCAp18 EBV antigen, shown in **Reviewer Fig. 4**. Notably, the only region surpassing genome-wide significance was at the MHC locus, which is consistent with the study noted by the reviewer⁴.

Reviewer Figure 4. Quantitative VCAp18 GWAS in UKB. A Manhattan plot of all associated variants with anti-VCAp18 titers is shown, highlighting the region surpassing genome-wide significance in the MHC locus that was not present in the binarized serostatus GWAS (Manuscript Extended Data Figure 6c).

Ultimately, the difference between binarized and quantitative outcomes in UKB is consistent with other prior studies, including by Scepanovic *et al.* 2018⁵, which examined both binary serostatus against EBNA-1 (presence of EBNA-1 antibodies, regardless of levels) and quantitative IgG levels against EBNA-1 in 1,000 healthy individuals. Namely, Scepanovic *et al.* similarly found significant associations within the HLA class II locus only for the quantitative IgG levels, but not for binary EBNA-1 serostatus⁵. This study suggested that larger sample sizes may be necessary to detect associated variants with binarized serostatus. However,

even with our sample size of 8,669 individuals, we did not observe any variants reaching genome-wide association significance for EBV serostatus. Further, from our inferred four-digit HLA calls, no individual HLA allele was HLA-C*08:02 ($P = 5.7 \times 10^{-5}$). Hammer *et al.*⁴ also noted this discrepancy by noting that a low but significant correlation was observed between the odds ratios for EBV serostatus and effect sizes (betas) of IgG levels ($r = 0.39$, $p < 2.2 \times 10^{-16}$).

For the purposes of our manuscript, we elected to show the binarized GWAS result, as it aligns more closely with our method for binarizing EBV DNAemia. However, noting that quantitative serology levels recovered genome-wide significant associations, we have modified the quoted sentence from the text for overall consistency and to minimize potential confusion:

[Page 10, Lines 317-319] Namely, serostatus reflects a history of past infection, which is largely independent of genetic variation, whereas our EBV DNAemia trait identifies a subset of individuals with persistent viral DNA.

And included in the new discussion section:

[Page 13, Lines 461-463] Further, we characterize EBV DNAemia as a polygenic trait regulated by genetic loci impacting antigen presentation and both adaptive and innate immune signalling.

We appreciate the careful thought that has allowed our reanalysis and characterization to be more clearly described and consistent with prior literature.

Minor:

Line 171 seems incomplete. Was there meant to be an elaboration of the previous sentence?

We have addressed this accordingly.

I caught a couple of other typos; parentheses usage on line 231 and 235, line 233 "...shared genetic architecture *between* EBV DNAemia and ...", line 236 "rick loci", line 245 "EDs" should be DCs?

That is correct – we did intend to refer to DCs (dendritic cells). We have addressed this accordingly.

Referee #1 (Remarks on code availability):

I was only able to review the code relating to downstream analysis as reviewing all code would require exporting large amounts of biobank-level data. However, the code for data extraction seems appropriate. Cod for downstream analysis was usable with sufficient instruction to run.

We appreciate the additional time and effort spent reviewing our custom analysis code. We have enhanced these resources with greater detail and instructions in our revised version of the manuscript.

Referee #2 (Remarks to the Author):

The authors analyze whole genome sequence (WGS) data for EBV DNA sequence from the UKB and AoU large databases. They find elevated EBV DNA in subset of donor samples which can then be further correlated with donor molecular traits and phenotypes. The authors find some very interesting correlations, including elevated EBV DNA in individuals with biomarkers to respiratory, autoimmune and cardiovascular disease. They find correlations of EBV DNA levels with antigen processing pathways and antigen presenting cells. They use predictive viral epitopes to implicate MHC class II as a major determinant of EBV persistence.

Overall, this is an impressive and important study that attempts to correlate EBV DNA levels with various molecular signature pathways and phenotypes from the large data bases of UKB and AoU. This is a very important effort to build accurate correlations between EBV DNA levels in available blood samples with any disease mechanisms, especially those not previously recognized. Some very intriguing new connections are drawn, particularly with strong correlations between EBV and COPD (Fig. 2) or Crohn's Disease (Fig. 3). However, there remain some concerns as to the interpretation of the data and whether the computational methods are sufficient, by themselves, to draw real-world conclusions without further experimental validation.

We sincerely appreciate the positive assessment of our study on the importance of our work. In our revised version of the manuscript, we have provided extensive new analyses of various datasets, as well as reproducible computational methods for the full impact of our study to be fully realized.

Specific Comments

1. Primary concern is the interpretation of high levels of EBV in whole blood samples. It is not obvious that the methods used can clearly distinguish whether viral copies detected are from a few lytic cells or a larger population of latently infected cells. Only 9.7% of individuals had EBV DNA detected, but surely these individuals are mostly EBV+ and were only captured as positive during periods of reactivation?

We understand the primary concern and have taken several steps to characterize this phenomenon in our revised version of the manuscript. In particular, we present a new suite of analyses, summarized in **Reviewer Figures 5-7**, that provides substantial additional evidence to corroborate our characterization that the EBV DNA detected in our framework is indeed primarily from latent rather than lytic infection.

First, to characterize the 9.7% of individuals with EBV DNA detected, we hypothesized that this set of individuals likely represent a tail of a spectrum of EBV DNA latency across the population. To demonstrate this, we performed a simulation study to ask if a simple normal distribution of log EBV DNA counts with a left-censor (representing a limit of detection) could approximate the empirical distribution of EBV abundance. The result of this simulation follows in **Reviewer Fig. 5**.

In brief, we assumed a mixture of 10% true zeros (EBV negative individuals) and 90% EBV exposure. For the 90% EBV-exposed, we then simulated a normal distribution of log-scaled EBV DNA copies per 10,000 cells (mean = 0.2 viral genomes / 10,000 cells; approximated from maximum likelihood estimation) and censored the distribution near the limit of detection of 1 viral genome per 10,000 cells (**Reviewer Fig. 5d**). Once censoring values were lower than this threshold, the simulated data and empirical observations were very similar, including at high values of EBV DNA latency (**Reviewer Fig. 5e-g**). Hence, we conclude that a relatively simple model of latent viral DNA abundance can explain the underlying distribution of the data.

We suggest that individuals with high levels of reactivation would likely reflect a distinct process that would result in a fraction of the distribution that couldn't be explained by a single underlying distribution.

Reviewer Figure 5 (manuscript Extended Data Fig. 1d-g). Simulations of EBV DNAemia under a censored log-normal distribution. (d) Simulated data of a mixture of 10% 0 EBV and 90% log-normal EBV. The dotted line indicates the threshold for data censoring. **(e)** Result of data censoring on simulated data. **(f)** Empirical distribution of observed EBV levels. **(g)** Comparison of donor positivity from simulated and observed EBV levels. The threshold of 1.2 EBV genomes per 10^4 human cells was chosen in the manuscript for downstream analyses, and the results are consistent with the simulation even at high levels of latent EBV DNA detection.

Ultimately, we include this simulation in the revised version of the text and the reviewer response for illustrative purposes. We do not claim that our simulation perfectly recapitulates the biology of our observed EBV DNA levels. However, we believe that this simulation conveys that even the top individuals with high EBV DNA levels (e.g., exceeding 1 copy per 1,000 cells) could be explained by the same generating function as created by the overall observed data. Our responses below further characterize why we interpret EBV reactivation within the UKB population to be relatively rare at the time of sampling.

Methods: lines 454-463. The explanation for why the EBV reads must be derived from latent EBV infection is not clear. Most individuals will have some cells that are undergoing spontaneous lytic reactivation. The methods described, as far as I can tell, do not distinguish between a latent or lytic simply by examining DNA content from short read (paired end illumina) sequencing. Knowing if the DNA was from lytic reactivation or expanded latency pool would be valuable.

We agree that resolving lytic versus latent infection is of great value for interpreting our study. Consequently, a major focus of the new analyses in our revised manuscript has been characterizing whether the EBV DNA detected from blood-derived WGS could be attributed to latent infection or lytic reactivation.

As viral RNA is a necessary intermediate for lytic infection or reactivation⁶, we reasoned that more systematic analyses of peripheral blood (sc)RNA-seq datasets would characterize the degree of lytic infection from healthy individuals during sampling. As neither UKB nor AoU have RNA-seq profiles available from peripheral blood, we relied on two additional large-scale cohorts, GTEx ($n = 682$ donors with WGS + RNA-seq) and OneK1K ($n = 982$ donors with scRNA-seq). We reanalyzed these cohorts using the same bias-aware chrEBV read processing (for GTEx) and with a streamlined kallisto workflow for single-cell data (for OneK1K) that was inspired by our recent work characterizing HHV-6 reactivation⁷. The result of these comprehensive transcriptomics analyses is shown in **Reviewer Figure 6**. GTEx has 681 donors who had both WGS and RNA-seq. Among them 178 had EBV DNAemia and 1 (~0.6%) of those 178 with EBV DNAemia had non-zero RNA (note: this donor had only 1 EBV RNA read).

Reviewer Figure 6 (manuscript Extended Data Fig. 3a-g). Characterization of potential EBV reactivation in large genomics datasets. (a) Schematic for GTEx consortium, highlighting donors where matching WGS and RNA-seq data are available from peripheral blood. **(b)** Sum of per-base read coverage of high-confidence EBV-mapping reads. Two repetitive regions with inflated coverage are noted in purple and red. **(c)** Characterization of EBV detection from 681 GTEx donors with paired WGS and RNA-seq from peripheral blood. **(d)** Schematic of data generation from OneK1K cohort. **(e)** Summary of EBV quantification from OneK1K cohort, showing total read number from the consortium (top) and number of EBV-assigned UMIs. **(f)** Summary of scRNA-seq results from a lymphoblastic cell line (LCL), a positive control for EBV transformation. **(g)** Summary of scRNA-seq results from a kidney transplant, a positive control for EBV reactivation.

Next, we formalized our OneK1K analysis, where PBMCs from 982 donors were profiled with scRNA-seq, spanning 53.9B reads and 1.449M cells (**Reviewer Fig. 6**). Here, we detected 1 total EBV UMI from this entire cohort, reflecting that viral reactivation is minimal in these samples. To control for the possibility that 10x 3' scRNA-seq technologies are biased against EBV transcripts, we reprocessed two libraries with known EBV expression, an immortalized LCL sample as well as a primary kidney transplant dataset (**Reviewer Fig. 6f,g**). In both cases, we were able to detect hundreds to thousands of EBV UMIs despite a sequencing depth per library that was ~500-1,000x lower than the sum of the OneK1K cohort.

These analyses strongly suggest that across these ~1,600 donors that lytic infection is a rare event. From these results, we have additional confidence in our interpretation that the EBV DNAemia phenotype reflects latent viral infection. Nevertheless, we do note that even in the setting of reactivation, our results likely still hold and are interpretable. Namely, individuals experiencing higher sporadic reactivation

EBV copy number and activity is known to fluctuate over time for most healthy carriers. It is not clear whether some or most of these detectable loads are due to transient increase in EBV due to sporadic reactivation. How might this affect the interpretation of the data and the correlations with any particular disease or genotypes?

We appreciate the question and acknowledge that EBV DNA levels will fluctuate over time in some capacity. To assess this, we considered the longitudinal data available in UKB relevant to our study, consisting of a limited set of longitudinal serology measurements from a subset of individuals. Here, these 262 individuals, including 17 individuals with EBV DNAemia, had baseline serology profiles and WGS with additional serology measurements taken 2–6 years following the initial sample collection⁸. We also emphasize that no longitudinal WGS was taken, as each individual was sequenced only once, and for

these 262 individuals (including all 17 individuals with EBV DNAemia), the WGS blood sample was taken at the first timepoint. **Reviewer Fig. 7** shows analyses of these data, including an exemplar individual for context (**Reviewer Fig. 7h**) and the overall analysis of longitudinal serology changes for these 262 individuals (**Reviewer Fig. 7i**).

Reviewer Figure 7 (manuscript Extended Data Fig. 3h,i). Longitudinal serology analyses from UKB. (h) Depiction of longitudinal serology measures of four EBV antigens for an exemplar EBV DNA+ donor. **(i)** Characterization of EBV longitudinal antigen titers across UKB. Statistical test: two-sided Student's *t*-test.

We hypothesized that if individuals in our dataset annotated with EBV DNAemia were the consequence of recent (sporadic) reactivation, these individuals could have increased viral titers longitudinally, including for antigens that track with recent infection or reactivation (e.g., ZEBRA and EA-D track with lytic infection). Analyses in **Reviewer Fig. 7** show no significant changes in longitudinal serology of individuals with EBV DNAemia. Ultimately, this supports our interpretation that the EBV DNA we detect is from latent infection.

Beyond these new analyses, we suggest that the possibility of fluctuations (i.e., variation in measurement) only indicates that our results are a **lower bound** for heritability and associated traits/genetic loci. In other words, our associations, including phenotypic outcomes and genetic linkage, reflect that the snapshot measurement from WGS enables sufficient signal to drive these significant associations, including dozens of reproducible phenotypic associations (**Fig. 2**), dozens of reproducible genetic variant associations (**Fig. 3**), and interpretable downstream analyses (**Fig. 4/5**). If the result was made with less error, then we would likely have even more associated loci / heritability signal. Hence, we would suggest that no major re-interpretation of our results are needed.

To further contextualize the extent of fluctuations in peripheral blood from healthy individuals, we examined the literature for longitudinal EBV DNA PCR analyses. Notably, a longitudinal study conducting EBV DNA PCRs in blood over a period up to 261 months found that, in healthy individuals, once EBV DNA has been detected, it remained detectable at **all** subsequent timepoints, suggesting that EBV DNA is a relatively stable biomarker in peripheral blood⁹. Hence, while there will certainly be fluctuations in large cohorts like UKB and AoU, we suggest these effects are averaged and there is no meaningful change in the interpretation of the significant associations as we do recover.

In future studies, if longitudinal DNA measurements are accessible to derive a more stable viral exposure outcome, we would expect that additional loci with more modest effect sizes could be identified, but our loci should nearly all be replicated. Hence, if anything, our interpretation of the data and its results is strengthened by the possibility of fluctuations, as our results likely represent a lower-bound of what is possible. Noting this, we have added an additional sentence in the discussion in particular to acknowledge how fluctuating levels may impact our interpretation:

[Page 15, Lines 485-488] As our per-donor WGS measurements represent a snapshot of fluctuating viral DNA levels over a lifetime, longitudinal profiles would likely refine our EBV DNAemia phenotyping and power further discovery of genetic loci that underlie population variation in viral latency and persistence.

Further related to this concern, Line 537 indicates that top EBV load is 1 copy per 1.4 cells. This must clearly indicate a very active lytic infection in a large number of cells. Such high carriers should be considered separately from merely detectable latent infections, and stratifications should be re-evaluated for disease correlations and genotypes (e.g. HLA types).

We are grateful for the attention to detail on this point, as this brought up an interesting nuance in our dataset.

In UKB, the top individual had a copy number of 1 EBV genome per 92 cells (highlighted in **Fig. 1f**). From the results of the simulation in **Reviewer Fig. 5**, the top simulated data point was 1 EBV genome per 50 cells, reflecting that at least for UKB, we could explain even the highest values under the assumption of a log-normal distribution of latent EBV DNA. Therefore, we suggest that whatever underlying source of variation in EBV DNA copy number (assuming a log-normal distribution) could capture the full range of what we observe. Hence, there is limited evidence of another component that would be required to capture the tail of the distribution in this cohort, and we interpret the detected EBV DNA as predominately latent EBV DNA.

As an additional data point to verify this, we observed the top individual from the GTEx peripheral blood WGS to be 1 EBV copy per ~20 cells. In this paired sample with both RNA-seq and WGS profiled from the same populations of peripheral blood cells, we detected 0 EBV reads out of 50M+ from the library. The lack of RNA was consistent for other EBV DNA high donors in GTEx, as no appreciable levels of EBV RNA were detected in any of the EBV DNA+ individuals, save for a single read in one donor that was DNA+ but not among the highest 170 donors.

To the reviewer's point of considering individuals with very high levels, we re-ran our ExWAS at various thresholds considering thresholds up to 20 EBV genomes per cell. To summarize these results, these 20+ EBV genome donors represented the top 283 individuals, and association analyses showed Q variants were significant at genome-wide significance ($P < 5 \times 10^{-8}$), including for the HLA. At other thresholds (e.g., 8 EBV genomes / 10,000 cells, representing the top 2,115 individuals), we observed no significant associations outside the HLA region (**Reviewer Fig. 8**). Hence, if the interpretation is correct that these very high samples reflect active infections, we suggest that their incidence at the time of blood sample collection would be stochastic and largely independent of host genetics. Alternatively, these individuals may also be the extreme tail of an underlying distribution (as suggested in **Reviewer Fig. 5**), but the very high threshold results in a masking of individuals that leads to underpowered association results. In either case, these additional analyses provide strong support for the threshold that is used in the manuscript.

In longer form, to test variable thresholds of EBV DNAemia, we repeated our ExWAS workflow for **five** additional thresholds in addition to the one used in the current and revised manuscript. A summary of the cutoffs and number of positive donors is noted here:

- 1.2 EBV genomes / 10,000 cells: 47,452 positive individuals (baseline / in manuscript)
- 2.0 EBV genomes / 10,000 cells: 22,692 positive individuals
- 2.67 EBV genomes / 10,000 cells: 13,426 positive individuals
- 4.67 EBV genomes / 10,000 cells: 5,866 positive individuals

- 8.0 EBV genomes / 10,000 cells: 2,115 positive individuals
- 20.0 EBV genomes / 10,000 cells: 283 positive individuals

The results of our ExWASes at different thresholds are summarized in **Reviewer Figure 8**. For simplicity, we group all chromosome 6 genes together for ease of comparison at other loci. In brief, we observed that no genes were discovered at higher thresholds, and genetic signals were only lost as the threshold increased, ultimately resulting in 0 non-MHC genes being associated at the 8 EBV genomes threshold and 0 variants anywhere (MHC or otherwise) that were genome-wide significant at the 20 EBV genomes threshold. We attribute the drop in associations at higher thresholds as a result of changing the threshold by a modest amount greatly impacting statistical power, but also that the detected signals are not driven exclusively by high-threshold samples.

Related to this point, by drawing a threshold at higher levels, individuals who were previously “positive” become “negative,” likely increasing the error rate and further limiting statistical power.

Reviewer Fig. 8. Summary of ExWAS results at non-chromosome 6 loci at different EBV DNA thresholds. The top shows the number of DNA+ individuals at variable thresholds. The bottom shows whether the protein-coding gene had a significant ($P < 5 \times 10^{-8}$) association for one or more protein-altering variants associated with the binarized EBV DNA individuals. 1.2 was selected for main analyses in the manuscript.

Ultimately, our interpretation of these high EBV carriers is that most of these individuals (e.g., 10 EBV copies per 10,000 cells) can be explained by an underlying variable abundance of latent viral DNA in the population. However, there are almost certainly instances across these cohorts in which very high EBV levels are reflecting a recent or current lytic infection or reactivation event, particularly given the very large sample sizes in our cohorts. Though we cannot exclude the possibility of lytic infection in some donors, our RNA analyses, the simulation study, and the longitudinal serology results collectively suggest that the vast majority of the observed EBV DNA+ donors likely reflect latent EBV DNA. Lastly, when considering only these very high EBV donors, we detect no significantly associated loci, reflecting that there is no statistically significant genetic risk driven exclusively by very high carriers (which could be interpreted as including individuals with a possible recent infection).

2 Drug treatments were not accounted for but could have significant effects on EBV load and reactivation. Immunosuppressive drugs may explain some increases in EBV copy number confounding some of the correlations. How was drug treatment accounted for in this study?

We identified a list of 169 immunosuppressive drugs (**Extended Data Table 1**) and 15,661 individuals with any of these treatments who also have EBV DNAemia quantified in UKB. A simple Fisher test for association between drug treatment and EBV DNA+ yields a positive and significant association, which we have now added to the manuscript to demonstrate this association (OR = 1.2, $P = 7.51 \times 10^{-14}$; **Reviewer Fig. 9; Extended Data Fig. 1k**).

To characterize this in AoU, we obtained unique RxNorm codes for 53 of the 169 drugs (**Extended Data Table 1**) and queried for individuals with any of these drug exposures, along with the exposure start and end dates. Each individual was annotated as immunosuppressed only when the date of the WGS biosample collection fell between the drug exposure start/end dates (or after start dates, if no end date was recorded). We observed a non-significant trend between immunosuppressive drug exposure at the time of WGS collection and EBV DNA+ (OR = 1.03; $P = 0.54$; **Reviewer Fig. 9; Extended Data Fig. 2f**).

Reviewer Figure 9 (manuscript Extended Data Fig. 1k,2f). Characterization of immunosuppressive drug treatments on EBV DNA positivity. Left: UK Biobank; Right: All of Us. All bar plots show the proportion of individuals within the group with the corresponding standard error of the mean. Statistical test: Fisher’s exact test.

In our prior analyses, we did not explicitly account for drug exposure, and thus we have now performed additional analyses to assess how this would impact our conclusions. We reperformed the PheWAS with drug treatment as a covariate and compared it to our initial PheWAS results. We observed a near perfect correlation between this drug-corrected PheWAS and the original PheWAS, both with respect to the P values and effect sizes of the associations (**Reviewer Fig. 10**).

Reviewer Figure 10. Robustness of PheWAS analyses to drug treatments. Per-phenotype summary statistics from the manuscript (x-axis) and with adding immunosuppressive drugs as a covariate (y-axis). The left panel shows the $-\log_{10} P$ values; the right panel shows the beta coefficient from the PheWAS regression. Statistical test: logistic regression with relevant covariates documented in the PheWAS methods.

In summary, we observe a modest effect size between immunosuppression status and EBV, and near-perfect concordance of the PheWAS results with or without accounting for these drug annotations. We have now added the panels shown in **Reviewer Fig. 9** as well as accompanying text to the manuscript to describe this sub-analysis for the readership.

3. Fig 3D is remarkable, particularly the negative PC correlation of Crohn's Disease and Ulcerative Colitis with EBV negative status. This needs to be further commented and clarified as to whether this implies that EBV activity is negatively correlated with these autoimmune diseases. If that is correct interpretation, that is different than previous findings (e.g. Ebert et al., Nature Communications volume 15, Article number: 8383 (2024)).

We have further clarified this observation in the revised manuscript and also performed a replication analysis to strengthen these observations.

In the cupcake method, the variant by immune-mediated diseases (IMD) effect matrix is factored using principal component analysis. The cupcake scores are derived from principal component (PC) loadings, or the weights that each SNP has on defining a principal component. Principal component 1 (PC1; as defined in the original publication¹⁰) captures the main component of genetic risk, which we previously determined segregates autoantibody-positive ("seropositive") and autoantibody-negative ("seronegative"). Therefore, the negative value in **Fig. 3D** does not imply a negative correlation in of itself. Hence, the interpretation of the cupcake result enables characterizing which genetic architectures are more or less similar to any other IMD, reporting a measure of the shared heritability between traits.

In our specific case, the "negative" loading of CD and UC on PC1 reflects their established "seronegative" biology (i.e., a relative absence of highly disease-specific pathogenic autoantibodies). Our finding that EBV DNAemia clusters with "positive" PC1 scores (alongside RA, SLE, and T1D) indicates a shared genetic component between EBV DNAemia and the "seropositive" IMD spectrum. This does not imply that EBV DNAemia is genetically negatively correlated with CD/UC occurrence. Instead, the negative PC1 score for

CD/UC signifies that **these diseases possess a different component of genetic architecture from EBV DNAemia and antibody-associated IMDs**, not necessarily an inverse epidemiological or mechanistic relationship with EBV DNAemia. Therefore, we emphasize that our findings are not necessarily discordant with previous epidemiologic data.

To further confirm these results, we have now projected the summary statistics from AoU that provide an independent replication of these results. Here, we observe a very similar result with a positive PC1 value, as well as a statistically significant enrichment (**Reviewer Fig. 11**).

Reviewer Figure 11 (manuscript Fig. 3d). Revised analyses of genetic architecture overlap. The independent AoU architecture was added to replicate the direction and effect of the EBV result clustering with seropositive IMDs.

In our revised manuscript, we add additional clarification and interpretation of the cupcake results in the “Pleiotropy with complex disease” section, which discusses the genetic architecture of EBV DNAemia:

[Page 10, Lines 329-334] Notably, cupcake PC1 segregates an IMD genetic axis characterized by autoantibody seropositivity⁷⁸. Specifically, “seropositive” traits (like RA³⁵, SLE³⁷, and T1D⁷⁹) have similarly signed PC1 loadings, whereas “seronegative” traits (like Crohn’s disease and ulcerative colitis) are distinct (**Fig. 3d**). Consistent with our PheWAS and prior reports of EBV pathogenesis, we observed a cupcake PC1 score that reflects a shared component of genetic architecture between EBV DNAemia and autoimmune diseases such as RA, SLE, and T1D from both UKB (P = 1.3 × 10⁻⁵) and AoU (P = 4.8 × 10⁻⁷).

4. Figure 5. The HLA bound peptides from EBV are suggested by a computational prediction algorithm which has limited predictive value. There needs to be some experimental validation for this analysis prior to reaching any strong conclusions.

We agree in principle that the predictive value of presented peptides from HLA computational model can be limited, particularly for any one peptide. However, we stress that the analysis presented here **averages** over **hundreds** of HLA alleles by analyzing the full EBV proteome. We suggest that the predictive value of

the algorithm in this setting with large sample sizes enables discovering trends as we show here, even from imperfect predictions. Further, all the association results that we show are highly concordant between the two biobanks, meaning we are highlighting only results that are independently replicating rather than those that may be impacted by limitations from the computational algorithm.

Next, we stress that there is experimental validation of our predictions, including what is shown **Reviewer Fig. 12 / manuscript Fig. 5b**. Specifically, 16 distinct peptides predicted to be the top epitope presented by one or more HLA alleles were detected in the IEDB database, which provides high-quality annotations of validated presented peptides. We have clarified the figure legend in this panel to reflect this, as the IEDB “known” peptides provide a measure of experimental validation:

Reviewer Figure 12 (manuscript Fig. 5b). Summary of top antigens bound per MHC class, colored by whether the peptide is experimentally confirmed in IEDB¹⁴.

We emphasize that the abundance of red peptides (a statistically significant result against a null of random peptide presentation) and overall diversity (16 predicted distinct peptides that have experimental validation) exceeds what could be done from any one lab. Hence, our results do complement computational analyses with experimental validation. We reiterate this point below in revised text:

[Page 12, Lines 391-396] The top predicted epitopes prioritized by NetMHC were corroborated by previously identified EBV antigens in the Immune Epitope Database (IEDB)⁹², including 9 of 83 (10.8%) class I peptides and 7 of 110 (6.4%) class II peptides (Fig. 5b; Extended Data Fig. 8d; Extended Data Table 7). These overlaps were significantly enriched over a random set of peptides for both class I ($P = 3.5 \times 10^{-23}$; binomial test) and class II ($P = 0.047$; binomial test), verifying the capacity for NetMHC to predict viral peptide processing and presentation across HLA alleles.

Nevertheless, we agree that NetMHC has its limitations, and we should be cautious in communicating our results. To this end, in our revised manuscript, we have modified the analyses and conclusions from this section in several ways. First, we provide further support on the accuracy of the HLA allele genotyping (also suggested by Reviewer #3, Point 5; see **Reviewer Fig. 18 below**), which reflects that we have a high confidence in assembling four-digit HLA alleles per person. Using these corroborated genotypes, we report the association between the individual HLA haplotypes and EBV DNAemia (also suggested by Reviewer #3, Point 6; see **Reviewer Fig. 19**), which provides reproducible associations between HLA alleles and EBV DNAemia that do not utilize any binding strength prediction values. Instead, this analysis uses only the genetic data of HLA alleles and inferred EBV DNAemia to identify alleles associated with increased or decreased risk of EBV DNAemia. Finally, we note that we have rewritten our concluding remarks in this section specifically to avoid any strong takeaways, acknowledging that the computational prediction of these epitopes is imperfect:

[Page 12, Lines 414-417] Together, these results support a model where individual genetic variation, predominantly in MHC class II, is a determinant in the latency and persistence of EBV infection (Fig. 5f), and that the computational modeling between host alleles and the viral proteome is predictive of the incidence of EBV DNAemia.

Referee #2 (Remarks on code availability):

I am not able to evaluate the computational code.

Referee #3 (Remarks to the Author):

Epstein-Barr virus (EBV) is a ubiquitous human herpes virus infecting upwards of 90% of the population. This work by Nyeo et al. seek to understand the relationship between EBV latency and a range of phenotypes using a phenome-wide association (PheWAS) approach, as well as the genetic architecture of latent EBV and examine pleiotropism with complex disease. Additionally, the authors consider HLA variation in the population studied and implications from predicted binding of EBV epitopes to HLA. To identify EBV latency in the study population, the authors use a bioinformatics approach whereby they identify reads mapped to the EBV genome from existing whole genome sequencing (WGS) data. This approach allows for extraordinary sample size by accessing the UK Biobank (discovery) and All of Us (replication cohorts), improving on sample size for previous studies of EBV in health and disease by many orders of magnitude. In doing so, they show associations between the EBV phenotype thus derived, and a number of complex traits, and show evidence for shared genetic architecture.

Overall, I find the approach very compelling, and an extremely clever use of an existing large dataset. The authors should be commended on their identification of a highly significant knowledge gap. Understanding the role that this very common virus plays in human health is of great interest, particularly in light of studies that have shown its critical role in MS pathogenesis, for example. Beyond that, it only goes to reason that latent infection may carry unrecognized health consequences, and novel approaches to uncovering these effects are much welcome. Further, the ways in which genetic background, both human and EBV, impact the manifestation of disease pathogenesis are even more poorly understood. This work attempts to answer truly important and novel questions. Additionally, a method for doing so which can be broadly applied to already collected datasets would be very valuable. However, there are several aspects of this work that undermine the analyses and conclusions presented in the paper.

We sincerely appreciate the positive assessment of our study, both in terms of the methodology and the potential impact of this work.

The manuscript is well-written and easy for the reader to follow. I find that the majority of data provided is high quality. The author's bioinformatic analysis of sequencing data and experimental choices are reasonable and account for methodological concerns of working with NGS data, such as handling of reads that map to repetitive regions. The quality of figures and presentation in this manuscript is very high. It is clear that the authors prioritized reducing complexity while increasing visual impact and reader accessibility and should be commended for doing so.

We are grateful for the positive feedback on the strengths of our work.

Major:

1. Most problematic here is the underlying assumption that the authors are detecting a molecular signature of EBV latency. To the best of my knowledge, infection with EBV always results in latent infection¹. Given this, the question arises, what, if anything biological (as opposed to sequencing/bioinformatic artifact), are the authors measuring? They repeatedly refer to the molecular trait they identify as a measure of EBV latency/EBV DNA positivity ("DNAemia") (eg lines 24, 107, 115)—however, we expect that 90% (the value given by the authors for EBV seropositivity in the cohort) will bear B lymphocytes with incorporated EBV DNA, whereas the authors find this trait in ~10% of their cohorts. They come to that figure by examining the number of reads mapping to the EBV genome after masking repeat regions that likely cross-map with human reads, and using EBV seropositivity as a "ground truth" they empirically land on a number that most strongly associates with that phenotype. While this approach feels a bit like reverse-engineering these

results, I'm open to the notion that this is detecting something interesting—but not EBV latency, per se. One wonders, for example—are the authors detecting an inflammatory response, whereby B cells latently infected with EBV are dividing? This could explain, for example, the association with RA. The authors state that some associations, such as emphysema and tachycardia, as possibly attributable to the known association of smoking with EBV reactivation—however, this undermines their claim that they have detected latent, rather than active, lytic virus. In summary, it is clear that the DNAemia metric is capable of finding EBV DNA from peripheral blood samples. However, I do not have confidence from what the authors provided that this can be used to infer its latent state. I caution that unless the authors can satisfactorily provide additional evidence to support their assumption that the patterns they are capturing represent EBV latency, this paper will require thematic revision and re-evaluation of its major biological conclusions.

We appreciate the reviewer's attention to detail and agree that further interpretation of our EBV DNAemia biomarker is required to support our major biological conclusions.

First, regarding the discrepancy of the ~10% EBV DNA positivity in our data versus ~90% EBV seropositivity in the population, we provide results from a simulation analysis that provides clearer contextualization (**Reviewer Figure 5**). Namely, our simulation supports the idea that the EBV DNAemia population is a tail of the distribution of exposed individuals that have the highest latent EBV DNA levels. Others in the population are likely carriers of EBV DNA (reflecting past infection), but below the limit of detection from the WGS data.

Reviewer Figure 5 (reproduced from above). Simulations of EBV DNAemia under a censored log-normal distribution. (d) Simulated data of a mixture of 10% 0 EBV and 90% log-normal EBV. The dotted line indicates the threshold for data censoring. (e) Result of data censoring on simulated data. (f) Empirical distribution of observed EBV levels. (g) Comparison of donor positivity from simulated and observed EBV levels. The threshold of 1.2 EBV genomes per 10^4 human cells was chosen in the manuscript for downstream analyses, and the results are consistent with the simulation even at high levels of latent EBV DNA detection.

To further characterize the potential B cell responses associated with EBV DNAemia as suggested, we evaluated the GTEx data with our EBV DNAemia annotations. We first tested whether there is a difference in B cell numbers associated with EBV DNAemia, and indeed, we observed a modest but significant increase in B cell fraction (via bulk RNA-seq deconvolution). Next, we hypothesized that if the detected EBV DNA originated from active lytic infection, we would observe an elevated B cell activation score, since B cell receptor activation and/or plasma cell differentiation can trigger EBV reactivation in infected cells¹⁵. Interestingly, we instead observed a significant decrease in B cell activation score in the EBV DNA+ group (**Reviewer Fig. 13**).

Reviewer Figure 13. RNA-seq analyses of GTEx peripheral blood. Left: estimated B cell fraction using RNA-seq deconvolution and cell type abundance estimation. Right: B cell activation (Panther Pathway) module score per donor. *P* values are from Mann Whitney U Tests with two-sided hypothesis testing for $n = 681$ GTEx donors.

Due to the limitations of these analyses (i.e., bulk-level RNA-seq) and lack of any quality replication cohort, we have elected to report these results here in the Response to Reviewer file. We note that for both the single-cell RNA (OneK1K cohort) and bulk RNA (GTEx) cohorts, we could not identify sufficient samples with evidence of EBV RNA to enable comparative analyses of lytic/reactivated EBV. The lack of EBV RNA in these large genomics datasets further supports our original framing that the EBV DNA we detect is a marker of latent rather than active lytic infection.

In sum, we appreciate the careful consideration and hope that our extensive new analyses documented in detail in responses to previous reviewers further establish our EBV DNAemia as a latent viral trait. Within the manuscript, we have added a section that details all additional analyses conducted to characterize our viral DNA measurement as a latent viral molecular phenotype.

2. While the above approach is interesting, and the authors show concordance of their measures of EBV-mapped reads with orthogonal data including seropositivity and scRNAseq, it absolutely needs further support in even a small cohort where all measure could be evaluated concomitantly with the WGS data, to provide some concrete validation. It is impossible from the data presented to feel confident about what specific trait is being identified here.

We agree that additional molecular characterization of this phenomenon is required to properly characterize our EBV DNAemia trait. Hence, we refer the reviewer to the significant additional analyses presented in **Reviewer Figures 5-7**. We now provide a total of four orthogonal analyses to characterize our DNAemia phenotype:

1. Paired WGS and RNA-seq dataset from whole blood of 681 individuals from the GTEx consortium (**Reviewer Fig. 6a-c**; below) shows a total of 3 reads even among 178 DNA+ individuals.
2. An scRNA-seq dataset of 982 donors from OneK1K, with additional public scRNA-seq datasets that serve as positive controls, similarly shows only 1 EBV-assigned UMI (**Reviewer Fig. 6d-g**; below).
3. Paired serology over a longitudinal setting from 266 individuals in UKB shows no significant difference in four EBV antigen levels stratified by donor EBV DNAemia (**Reviewer Fig. 7**; below).
4. Simulations that contextualize our underlying data could be interpreted as a censored value from an underlying normal distribution of log EBV counts / cell (**Reviewer Fig. 5**; above).

Reviewer Figure 6 (reproduced from above). Characterization of potential EBV reactivation in large genomics datasets. (a) Schematic for GTEx consortium, highlighting donors where matching WGS and RNA-seq data are available from peripheral blood. (b) Sum of per-base read coverage of high-confidence EBV-mapping reads. Two repetitive regions with inflated coverage are noted in purple and red. (c) Characterization of EBV detection from 681 GTEx donors with paired WGS and RNA-seq from peripheral blood. (d) Schematic of data generation from OneK1K cohort. (e) Summary of EBV quantification from OneK1K cohort, showing total read number from the consortium (top) and number of EBV-assigned UMIs. (f) Summary of scRNA-seq results from a lymphoblastic cell line (LCL), a positive control for EBV transformation. (g) Summary of scRNA-seq results from a kidney transplant, a positive control for EBV reactivation.

Reviewer Figure 7 (reproduced from above). Longitudinal serology analyses from UKB. (h) Depiction of longitudinal serology measures of four EBV antigens for an exemplar EBV DNA+ donor. (i) Characterization of EBV longitudinal antigen titers across UKB. Statistical test: two-sided Student's *t*-test.

In sum, these additional lines of evidence bolster our characterization of the EBV DNAemia trait. Regardless of the precise mechanism of EBV DNAemia, we emphasize that we are observing reproducible associations between our viral trait, human phenotypes, and human genetic variants. As we acknowledge that we cannot fully rule out other modes of EBV DNA exposure across the full ~735,000 individuals that we analyze in this paper, to complement these analyses, we have added additional discussion in several parts of the manuscript to qualifying our analyses and contextualize our results.

3. The authors use demographic data to support the notion that this molecular trait is representative for latent EBV infection, including sex-bias, age-associated increase, and an increase with northern latitudes.

On the latter point, the authors claim that this is consistent with findings linking EBV infection with vitamin D levels. This is problematic because the paper they cite in support of that relationship shows no such thing, but does state “The timing of EBV infection may have significant latitudinal variation with the majority of EBV infections in more tropical areas occurring early in life..” which in fact suggests that the variation in latitude that the authors observe may be due to infection recency in the cohort.

We have revised the manuscript to reflect on the associations. We emphasize that these analyses are in large part meant to provide context to the reader as we characterize our phenotype, but we do not claim for these relationships to be causal nor any particular covariate association to be essential for the conclusions of our work. Hence, we agree that the previous statement about with higher latitudes should be revised. The sentence now reads:

[Page 5, Lines 126-128] *Annotating each individual by birth location, we observed an increasing proportion of EBV DNA+ individuals at more northern latitudes in the UK, consistent with prior reports of increased EBV infection further from the equator²⁹ (Extended Data Fig. 1h).*

Likewise, it is clear from Extended Data Figure 1 that there a dip in frequency of this trait after middle age, with frequency again increasing after age 75. Again, this looks more like a measure that is measuring infection recency, and very possibly lytic virus, with increase later in life coinciding with known decline in immune function.

For ease of interpretation, we have reproduced our age-associated frequency plots for both biobanks as **Reviewer Figure 14** below:

Reviewer Figure 14 (manuscript Extended Data Fig. 1i,2d). Characterization of the association between age and sex on EBV DNAemia.

While we cannot definitively explain what drives the variation in this result, we respectfully disagree that this dip is necessarily reflecting infection recency. First, we note the magnitude of this dip ranges from 17.0% ($\pm 0.5\%$) in males aged 55-60 to 11.4% ($\pm 0.4\%$) in males aged 70-75. Importantly, the observed dip is only

evident for males in the AoU cohort but does not reproduce for males in UKB or females in either cohort. Hence, we suggest whatever source of variation underlies this effect may not be fully generalized.

Regarding the possibility of infection recency, data from the US indicates that 83% seropositivity for individuals aged 18-19¹⁶. Similarly, that retrospective data from the UK reports >90% seropositivity among individuals aged 22-24¹⁷. Hence, most individuals in this analysis were likely initially infected ~40 years prior at the peak and ~55 years prior at the lowest point. We suggest that this 15-year difference in the context of 4+ decades of prior infection certainly has some effect, but it does not seem obvious to us that this would explain the drop in the rate of EBV DNAemia.

Regardless of the reason, our association results include sex, age, age², and age*sex as covariates that can capture this variation to minimize confounding for the GWAS or PheWAS results. Ultimately, if there is a source of unmeasured confounding in our associations, we emphasize that we primarily report results from both biobanks, in which no such dip occurs in UKB, hopefully limiting the impact of these unobserved variables.

As part of the revised text, we further note the possibility of age-associated fluctuations in EBV DNA copy number and that longitudinal measurements could resolve these potential challenges. We again emphasize that, if anything, our results are conservative and better donor phenotyping should improve power for discovery (rather than invalidate any of our statistically rigorous associations):

[Page 14, Lines 485-488] As our per-donor WGS measurements represent a snapshot of fluctuating viral DNA levels over a lifetime, longitudinal profiles would likely refine our EBV DNAemia phenotyping and power further discovery of genetic loci that underlie population variation in viral latency and persistence.

4. In sections “EBV is a biomarker of complex traits” (lines 121-159) and “Polygenic variation underlies EBV DNAemia” (lines 162-220), the authors describe the association results of the two cohorts (UKB and AofU) separately. They do not explicitly describe which results replicated in both studies. The purpose of a replication cohort (AofU) is to identify signals that reproduce in both cohorts independently, thus increasing confidence in the signal. As the results are currently written, the reader cannot meaningfully compare between the two cohorts. I would recommend re-writing these two sections by comparing the association signals of the two cohorts first, identifying the signals that are reproducible in both, and then re-writing the results to describe only those signals that reproduced. Otherwise, there is no reason to include a replication cohort.

We appreciate the suggestion to formalize the replication associations from these analyses. We have now revised both sections of the text accordingly, and we have also provided greater detail in the **Extended Data Tables** with explicit labels for whether an association was replicated or not across the two cohorts.

For phenotypic associations replicated in both UKB and AoU, we have included a more detailed Extended Data Table 3 to complement the interpretation of the original scatterplot that showed the replication of the results (**Fig. 2b**). Namely, we have included the complete list of tested phenotypic traits in UKB (**Extended Data Table 3a**) and the full UKB PheWAS results (**Extended Data Table 3b**), as well as the 141 UKB ICD codes from **3b** that reached phenome-wide significance ($P < 3.3 \times 10^{-6}$, Bonferroni correction) and had matched codes in AoU (**Extended Data Table 3c**). In our revised manuscript, we have included this section to further clarify the comparison:

[Page 7, Lines 200-204] We sought to replicate these associations using the AoU cohort (Fig. 2b). As the underlying EHR data between cohorts vary, we focused on 141 of the significantly associated matching ICD10 codes in UKB with sufficient representation in the AoU cohort for replication analyses (minimum $n = 24$ cases). 87 (62%) of these matching codes replicated in AoU ($P < 0.05$; OR concordant with UKB statistics), resulting in a set of traits that we examined more closely (Methods; Extended Data Table 3).

For genetic associations that were replicated across the two biobanks, we have annotated each of the sentinel GWAS loci from our UKB NFE analysis with whether they were replicated in the AoU EUR GWAS (Extended Data Table 4a). In our revised manuscript, we have also further elaborated on the 40,675 variants that were genome-wide significant ($P < 5 \times 10^{-8}$) in UKB and passed quality control filters in AoU:

[Page 9, Lines 292-297] Of these, 91.4% of variants were replicated in the AoU GWAS (nominal $P < 0.05$; OR concordant; Fig. 3c). As the genome-wide significant hits occurred in the HLA region, we computed that 12 of the 19 (63%) assayed sentinel GWAS loci replicated in the AoU GWAS (nominal $P < 0.05$; OR concordant; Extended Data Table 4). These concordant results indicate that persistence of latent EBV DNA is a polygenic trait, and loci underlying EBV DNAemia are reproducible across continents and quantifiable using repurposed WGS reads.

5. I also had some concerns regarding the genetic analysis in the section: “Polygenic variation underlies DNAemia”

a. Why have the authors exclusively focused on exome variants in the human genome? Certainly, they are important, but increasingly, genetics literature emphasizes the importance of intergenic, non-coding and regulatory elements in genetic architecture of complex traits. Particularly in the context of a virus known for extensive restructuring of host cell chromatin architecture. Presumably some of the 21 independent associations from the genotyping array GWAS were of this category? Please include a description of those results as well.

We apologize for the confusion in these sections. Our initial focus on exome variants was strictly for interpretability of the effect variant, as protein-altering genes have the most straightforward interpretation rather than the non-coding genome, which can be challenging to ascertain function. Regardless, we agree that our presentation of the results (i.e., focusing on protein changes) did not provide a sufficient representation of the underlying GWAS that was performed.

To address this, we have provided additional analyses that highlight the genetic architecture, including the analyses in Reviewer Figures 15 and 16.

In brief, we used stratified LD Score regression to assess heritability enrichment in different annotations of the genome (Reviewer Fig. 15). The result showed an enrichment in regions of high conservation in the genome as well as non-coding regions, including super enhancers and transcription start sites (TSS). Additional analyses using ATAC-seq data from 18 FACS-purified populations showed an enrichment of signal in the lymphoid lineage, particularly B cells, and terminally differentiated myeloid cells, which is consistent with our prior protein-coding analyses (Fig. 4). These analyses highlight the importance of the non-coding genome in mediating the effects of associated loci from the genome-wide significant data.

Reviewer Figure 15 (manuscript Extended Data Figure 5a,7a). Analyses of non-coding associations of EBV DNAemia genetic associations. (Left) Partitioned heritability results from stratified LD score regression using baseline annotations¹⁸. Shown are the 9 genomic features with positive heritability enrichment at a nominal $P < 0.05$ from the LD Score calculation. **(Right)** Enrichment of non-coding variation in accessible chromatin from ATAC-seq profiles of sorted immune populations. P values were derived from an empirical distribution using a permutation test (chromVAR¹⁹).

To complement the zoomed-out analyses, we also include a vignette of the *EOMES* locus that we previously highlighted (**Reviewer Fig. 15**). Using AlphaGenome²⁰, an *in silico* model that predicts variant effects in genomics assays (including chromatin profiles via ChIP-seq), we provide additional mechanistic insight showing that this variant near the *EOMES* promoter is linked to increased repressive chromatin marks and a corresponding decrease in H3K27ac and other active marks. We note that we have previously highlighted the rs3806624 variant in our original manuscript, and showing this detailed effect hopefully mitigates any sense that variants associated with EBV DNAemia are not linked to non-coding genetic variation.

Reviewer Fig. 16 (manuscript Extended Data Fig. 5f). Characterization of a *EOMES* promoter variant using AlphaGenome. The figure illustrates a non-coding variant near the *EOMES* promoter that is associated with increased repressive chromatin marks and reduced levels of active marks.

b. It is unclear how the results from the array-based genome-wide association study and the exome association study were meaningfully compared. It would be helpful to add a discussion of how much overlap there was, if any, in terms of associated loci identified via these two methods. Especially when discussing the “consistent” results at the MHC (lines 183-186). Without the results from the gwas, there is no way for the reader to evaluate if they are in fact consistent.

We agree that the comparison between the ExWAS and GWAS was previously insufficient. To this end, we have additional analyses and tabular results to demonstrate that the results are indeed consistent.

First, we compared the set of 43 variants in non-MHC and 801 variants in the MHC locus that were associated at genome-wide suggestive levels ($P < 5 \times 10^{-8}$). **Reviewer Fig. 17** shows the results of this comparison, which have also been added to the manuscript for the non-MHC variants. In short, there is a very strong association between the effect sizes ($r = 1$) whether the variant was measured via the array or sequencing. We include this result to emphasize the overall consistency.

Reviewer Figure 17 (manuscript Extended Data Fig. 5b). Comparison of genetic loci detected in both ExWAS and GWAS.

Further, to compare what GWAS loci have signals implicated in ExWAS (i.e., protein-coding gene alterations), we have provided an annotation in **Extended Data Table 4a** that reflects whether an associated locus also had a protein-coding alteration, to refine the interpretation of the GWAS. 9 out of the 22 genome-wide significant GWAS sentinels we highlight had a corresponding protein-altering variant (within 100 KB of the sentinel) that also reached genome-wide significance ($P < 5 \times 10^{-8}$). We note that these were performed for the UKB NFE analyses, as this was the same population powered for the ExWAS associations. We now better compare the ExWAS and GWAS results in the revised manuscript.

c. How were NGS reads assembled for HLA genes in the ExWAS analysis? Due to the high levels of allelic diversity at HLA genes, simply aligning reads to a reference sequence will not sufficiently capture the correct allelic composition. There are several tools specifically designed to call HLA genotypes, why was one of these not used? The authors even describe using T1K in a downstream analysis, to infer HLA genotypes in the AoU cohort. This tool should also be applied here as well, to ensure better accuracy of ExWAS-derived HLA association signals. Especially since so many of their strongest signals seem to come from HLA genes.

For the ExWAS analyses in **Fig. 3**, we previously provided gene annotations to reflect which coding gene was overlapping the protein-altering variant. In the revised version of the manuscript, we have removed these gene annotations for the sake of clarity, as they reflect variants from the ExWAS.

For more involved downstream analyses (e.g., **Fig. 5**), we indeed utilize proper tools for calling HLA genotypes (instead of simply aligning reads to a reference). We clarify the tools used for inferring individual allelic haplotypes from these cohorts here:

1. For UKB, the 4 digit HLA calls are available as part of the Research Analysis Platform, and we used these calls throughout our analyses. These were derived from the HLA*IMP:02²¹ software, now further clarified in the methods.
2. For AoU: no pre-existing HLA haplotypes are available as part of the provided data. Hence, we assembled these genotypes using T1K²².

To assess the accuracy of the imputed HLA alleles in UKB, we calculated concordance rates between imputed HLA genotype calls and corresponding calls derived from whole exome sequencing (WES, DRAGEN v4.3.6) for a subset of individuals. We chose the 9,687 donors who had serology measurements as an example (data field 23053). For each gene, we tabulated the total number of samples compared (total_samples), the numbers of concordant and discordant calls (concordant_samples and discordant_samples), and calculated the concordance rate. **Reviewer Figure 18** summarizes these concordance rates, which ranged from 85.6%-97.2%, with the exception of HLA DQA1.

Reviewer Figure 18. Confirmation of HLA allele inferences from arrays using sequencing data. (a) Concordance rate between HLA*IMP:02 calls and WES calls per HLA allele. **(b)** Estimated error rates for the DQA1 gene for either modality using trio data in the UKB.

Mendelian error rates (MER) were calculated separately for WES-derived HLA calls and SNP array-imputed HLA calls using 1,030 parent-offspring trios from UKB for DQA1 to assess the low concordance. For each gene, a Mendelian error was defined as any trio genotype configuration inconsistent with expected Mendelian inheritance patterns, considering only informative sites where all three individuals had non-missing genotype calls. The per-gene MER was computed as the number of trios with Mendelian inconsistency divided by the number of trios with complete genotype calls. This result shows a negligible error rate from the SNP array data (input into HLA*IMP:02) but a much higher error rate for the sequencing data.

Hence, for the revised version of our manuscript, we have elected to continue using the HLA*IMP:02 calls based on these additional analyses, as the UKB research analysis platform already hosts the HLA*IMP:02 calls such that other users can easily access these results. Explicitly, any instance of an HLA allele now refers to the full assembled haplotype rather than an individual variant.

d. When the authors describe their association findings across the MHC (lines 183-196, Figure 3b), they do not acknowledge or address the well-established presence of strong linkage disequilibrium (LD) in the MHC, nor how that impacts interpretation of association signals in the area. Though the authors did

calculate LD scores for three regions described in ExData Fig 3b-d, it seems they did not apply this analysis to the MHC. This is an extremely well-documented phenomenon and known technological challenge to studying the MHC and it should be acknowledged in order to meaningfully interpret association signal in the MHC.

e. Of the missense ExWAS signals, 634 (92%) occur in the MHC (Extended data Table 3). That's an overwhelming proportion localizing to a single region of the genome. And many of those MHC signals occur in non-HLA genes. Both these results merit acknowledgement and discussion but get neither. Please include. It may also be related to the previously referenced strong LD in the MHC, which can inflate signal due to LD with the causal variant(s). This further emphasizes the importance of considering LD at the MHC, as it might lead to a reduction of independent associated loci.

We agree with clarifying the signals at the MHC locus in Reviewer 3 points 5d and 5e. We have revised the manuscript in several places to account for the complication of the MHC region as well as its impact on the HLA locus, which we summarize below:

First, we have added text acknowledging the complex polymorphisms and strong LD in this region:

[Page 8, Lines 258-261] Specifically, as the MHC locus is a hyper-polymorphic region with strong linkage disequilibrium, ascertaining causal variants at this locus is particularly challenging⁵³. Hence, we first focused our efforts on interpreting associated loci outside the MHC locus.

Second, the reviewer's anticipation was correct that the abundance of replicated loci may be reduced outside of the MHC. As we note in Reviewer 3 Point 4, we observed that the replication rate was lower outside the MHC but still reflective of concordant results (63% of testable sentinel loci reached genome-wide significance in the AoU independent replication cohort):

*[Page 9, Lines 293-295] we computed that 12 of the 19 (63%) assayed sentinel GWAS loci replicated in the AoU GWAS (nominal $P < 0.05$; OR concordant; **Extended Data Table 4**).*

Finally, as suggested in Reviewer 3, Point 6 below, we perform a multivariate regression analysis in order to study the impact of functional HLA alleles (four digit genotypes).

[Page 11, Lines 367-373] Hence, we reasoned that explicit modeling of the HLA variation could refine our understanding of genetic variation underlying viral persistence.

*To assess this, we first assembled donor four-digit HLA alleles across all donors in UKB and AoU with NFE or EUR ancestry (**Extended Data Fig. 8a; Methods**). Using these per-donor genotypes and similar covariates to our GWAS, we performed a multivariate regression to assess whether each HLA allele was associated with increased or decreased rates of EBV DNAemia (**Methods**).*

6. In performing the GWAS on their EBV molecular trait, the authors identify the strongest signal in the HLA region. Beyond that, much of the subsequently identified genetic pleiotropy and pathway enrichment appears to be driven largely by HLA. Given that, it is inexplicable that no analysis of specific HLA allelic variation with this trait is presented. In fact, the authors have these data (HLA imputed genotypes in the study cohort), and they later use it to consider patterns of binding of EBV peptides to the HLA present in the cohorts. This latter seems an unusual and roundabout approach—why not look at binding rather to specific HLA alleles associated with the measured trait? Was this analysis performed at all?

We appreciate the excellent suggestion and have now included these analyses, showing that the overall effects are concordant. In brief, we conducted specific HLA associations using the per-person four-digit

HLA alleles as predictors in a logistic regression with EBV DNAemia as an outcome, along with standard covariates used throughout the paper (i.e., age, sex, genetic principal components, etc.). **Reviewer Fig. 19** shows the results of this association for both class I and class II alleles. In the plot, we highlight specific HLA alleles, including HLA-A*0301, HLA-B*5501, HLA-B*3501, and HLA-DRB1*1201 with known relevance for either EBV or EBV-related immune traits.

We have added a paragraph in our revised manuscript to help refine the MHC association signal in our data by providing an association signal per allele, including a comprehensive set of per-allele summary statistics for both biobanks in **Extended Data Table 7**. We hope that the results provide a more straightforward interpretation of the EBV DNAemia-MHC association in our genetic association analyses and a useful resource for identifying potential risk alleles of EBV DNAemia.

Reviewer Figure 19 (manuscript Extended Data Fig. 8b,c). Association of HLA alleles with EBV DNAemia. Scatterplot of associations for UKB-NFE and AoU-EUR (**b**) HLA class I alleles and (**c**) HLA class II alleles. Annotated alleles were called out in the main text with known relevance to EBV or EBV-related diseases.

For convenience, we reproduce the additional paragraph in the main text below that was motivated by these analyses. We appreciate the excellent suggestion to more fully characterize these HLA associations:

[Page 11, Lines 370-384] *To assess this, we first assembled donor four-digit HLA alleles across all donors in UKB and AoU with NFE or EUR ancestry (Extended Data Fig. 8a; Methods). Using these per-donor genotypes and similar covariates to our GWAS, we performed a multivariate regression to assess whether each HLA allele was associated with increased or decreased rates of EBV DNAemia (Methods). In total, we identified a total of 42 HLA alleles associated with EBV DNAemia, including 18 class I and 24 class II alleles (nominal $P < 0.05$ in both cohorts; Extended Data Fig. 8b,c; Extended Data Table 6; Methods). The result of this regression framework allowed for the explicit characterization of individual risk alleles for EBV DNAemia. For instance, one of the strongest risk alleles for EBV DNAemia was HLA-A*0301 (UKB: $Z = 2.6$, $P = 0.0060$; AoU: $Z = 10.3$; $P = 9.63 \times 10^{-25}$), an allele linked to increased risk of multiple sclerosis⁸⁵. Conversely, a protective allele against EBV DNAemia, HLA-DRB1*1201 (UKB: $Z = -8.7$, $P = 4.6 \times 10^{-18}$; AoU: $Z = -3.6$; $P = 3.9 \times 10^{-4}$), has been associated with less severe multiple sclerosis⁸⁶. We also note that two other negatively associated HLA alleles, such as HLA-B*3501 (UKB: $Z = -11.1$, $P = 1.3 \times 10^{-28}$; AoU: $Z = -8.7$; $P = 2.6 \times 10^{-18}$) and HLA-B*5501 (UKB: $Z = -11.2$, $P = 7.3 \times 10^{-29}$; AoU: $Z = -7.61$; $P = 2.8 \times 10^{-14}$), present known immunodominant epitopes from the EBV proteome^{87,88}, reflecting that strong peptide presentation may underlie decreased EBV DNAemia.*

Motivated by these anecdotes of immunodominant peptide binding from associated alleles, we hypothesized that systematic predictions of EBV peptide display and processing could further characterize variation in population-level EBV DNAemia.

Additionally, rather than generating all possible peptides from the generic EBV reference sequence, why not focus on the peptides derived from significant EBV variants discovered in the following section “Genetic diversity in EBV sequences” to test the HLA peptide binding performance? I think this would be more meaningful than just generating all possible peptides and showing that they all have different performance. This will add functional significance to the peptide-HLA allele relationship, as well as providing additional evidence to support functional relevance of the variants described in this paper.

We appreciate the reviewer’s thoughtful suggestion to combine two pieces of our analyses, leveraging our HLA-peptide binding approach to evaluate whether EBV variants generate better or worse binding peptides that could explain variability in immune escape and, in turn, detection in certain cohorts.

We first created all possible 8-, 9-, 10-, 11-, and 15-mers that covered each variant amino acid in the four variants of unknown significance (VUS) that were lowly or not detected in the UKB/AoU cohorts. We then scored the pairs of wildtype and mutated peptides using NetMHCpan (8 to 11-mers for class I HLA allele binding) and NetMHCIIpan (15-mers for class II HLA allele binding). The summary of these results for the 4 VUS are shown in **Reviewer Figure 18**.

Reviewer Figure 20 (manuscript Extended Data Fig. 9d,e). Characterization of altered epitope binding by nasopharyngeal carcinoma (NPC) variants of unknown significance (VUS). (d) NetMHCpan+NetMHCIIpan predicted presentation rank for class I and (e) class II peptides, each containing a mutated residue from one of four VUS. Blue dotted lines are the weakest validated IEDB peptide for class I/II (Extended Data Table 8). Dotted blue lines reflect the weakest epitope nominated by NetMHC and confirmed to be bound by IEDB for class I and class II.

We observed no obvious change in the binding score (computed as the change in rank), indicating that peptide variant presentation likely does not explain the potential mechanism of effect for the VUS (Reviewer Fig. 20). Instead, if these VUS are indeed functional in NPC, we hypothesize that there is a viral intrinsic function, which we summarize in our revised manuscript text:

[Page 13, Lines 444-447] In addition, our viral proteome NetMHC workflow suggested that these VUS are unlikely to alter peptide presentation (or thereby enable immune evasion), indicating that these variants, if indeed functional, may modulate viral-intrinsic functions (Extended Data Fig. 9d,e; Methods).

Minor:

1. It is well-established that socioeconomic status (SES) plays a role in timing and prevalence of EBV infection—was this accounted for in these analyses?

We appreciate the suggestion to consider SES. For UKB, we utilized the available Townsend deprivation index, a census-based index of material deprivation (positive values indicating high deprivation and

negative values indicating relative affluence). For AoU, we used the deprivation index available within the workbench, an externally sourced index from the U.S. Census American Community Survey via a three-digit zip code linkage (ranging from 0 to 1, with higher values indicating more deprivation). Annotating each individual in these biobanks with these measures along with our measure of EBV DNAemia, we assessed the impact of SES on EBV infection, the results of which are shown in **Reviewer Fig. 19**.

Reviewer Figure 21. Association between EBV DNAemia and deprivation indices across both biobanks. The rates of EBV DNAemia are plotted and stratified by **(a)** within UKB; **(b)** within UKB, stratified by genetic ancestry; **(c)** within AoU; **(d)** within AoU, stratified by genetic ancestry.

In brief, we observed a noticeable increase in %EBV DNA+ occurring at the higher deprivation bin when specifically stratifying by the bottom 80% and upper 20%. Fisher association tests confirmed a positive and statistically significant association between high deprivation and EBV DNA positivity (OR = 1.25, $P < 2.2 \times 10^{-16}$) (**Reviewer Fig. 21a**). We find a similar relationship in the US AoU cohort using the deprivation index (OR = 1.35, $P < 2.2 \times 10^{-16}$) (**Reviewer Fig. 21b**).

These results confirm the noted point about SES and EBV DNAemia, further providing a validation of our measure. However, we elected *not* to explicitly account for SES in our downstream analyses upon further evaluation of SES with other covariates. Namely, we observed that accounting for genetic ancestry largely attenuated the univariate associations in both UKB and AoU (**Reviewer Fig. 21c,d**), including in our meta-analysis framework. Given the strong association between SES and genetic ancestry, we felt it prudent not to draw attention to this association in our main analyses.

2. A basic demographic table, eg median age, etc, with crosstabs for EBV seropositivity would be helpful

We have now added such a table in the revised **Extended Data Table 1**.

3. Was analysis done in any non-European ancestry individuals? If not, why?

In our revised analyses, we have now included two additional genome-wide analyses that meta-analyze non-European individuals in UKB as well as a meta-analysis over six genetic ancestries in UKB and three in AoU. These additional multi-ancestry analyses identified a total of 23 additional genome-wide significant loci across a series of new meta analyses in **Extended Data Fig. 6**. This demonstrates the value of considering diverse populations in biobank-based virology studies and motivates more diverse biobanks to extend this work.

4. The authors often refer to “strong” associations, eg, EBV with “malaise/fatigue.” While I agree that the results appear to be strongly significant, I don’t know that I would categorize an OR=1.27 as a strong association.

We agree with this suggestion and throughout the manuscript have revised the text throughout the manuscript accordingly.

5. There is no clear definition of the metrics “EBV DNA positivity” or “binarized EBV serology” (Lines 168-169). This makes it difficult to evaluate the conclusion in the following statement (lines 169-171).

We have revised our manuscript such that bias-corrected EBV DNA load is now consistently referred to as the EBV DNAemia biomarker. Here, EBV DNA+ refers to the individuals with EBV DNAemia higher than the threshold of 1.2 EBV genomes per 10^4 cells. Additionally, binarized EBV serology refers to seropositivity, as defined by the UKB measurement (“EBV seropositivity with Epstein-Barr virus,” data field 23053):

[EBV positive / DNAemia definition: Page 4, Lines 113-117] Using serostatus as a ground truth and accounting for standard GWAS covariates, a cutoff of 1.2 viral genomes per 10^4 human cells yielded the strongest concordance with seropositivity (OR = 82.2, $P = 2.2 \times 10^{-16}$), with all DNA+ donors having at least one positive seroresponse against the four EBV antigens measured (**Fig. 1g; Extended Data Fig. 1b; Methods**). Using this threshold, we classified 47,452 (9.7%) individuals as EBV DNA+ for subsequent analyses (**Extended Data Fig. 1c**).

[Serology positive definition: Page 4, Lines 91-96] To assess this, we utilized complementary EBV serostatus of four EBV antigens (VCA p18, EBNA-1, ZEBRA, and EA-D) as an orthogonal measure of prior infection, evaluated for a subset of 9,687 individuals. Noting that EBV seropositivity requires sufficient antibody titers for at least 2 of 4 EBV antigens, we observed a nominal association between detectable EBV DNA when including these two repetitive regions and seropositivity (Fisher’s Exact test odds ratio [OR] = 1.2, $P = 0.03$; **Fig. 1c; Methods**).

6. Line 171- Unfinished sentence

We have revised this text accordingly.

7. I was unable to find the meta data describing the results in Figure 3a. Please include a table/figure (even if supplemental) containing the results for at least the 21 independent genome-wide loci obtained from the genotyping chip.

We apologize for this oversight and have now added this table in **Extended Data Table 4a**.

8. Line 572: It is not entirely clear what “EBV DNA detection” refers to, in the logistic model term. Is this the DNAemia measurement? If so, please keep terminology consistent throughout the paper. If not, please define how you measured “EBV DNA detection” specifically.

We apologize for the confusion. The term was referring to the EBV DNAemia measurement. We have revised our manuscript such that bias-corrected EBV DNA load is now consistently referred to as the EBV DNAemia biomarker.

9. Line 589: Leftover revision comment “(LD reference?)”

We appreciate the reviewer catching this and have removed the comment.

10. Line 650-651: “we removed appropriate genes from both...” Please define how these “appropriate” genes were chosen for removal.

The “appropriate genes” were referring to the HLA or chromosome 6 genes at the beginning of the sentence. The revised sentence now reads:

[Page 37, Lines 920-922] For analyses with HLA (Fig. 4e) and chromosome 6 excluded (Fig. 4f), we removed either HLA or chromosome 6 genes both from the foreground (i.e., test set) and background set for statistical analyses.

11. Line 661-662: Please provide more detail on how you generated this “synthesis” of reads from a variety of sources as input for T1K. This is not a standard approach.

We have revised the methods to reflect the precise execution of the T1K workflow, which combines reads aligning to the canonical HLA region as well as all alternate contigs.

[Page 37, 941-948] For the AoU cohort, predetermined HLA genotypes were not available in the workbench. Hence, we reconstructed the HLA calls for individuals of EUR ancestry, using the T1K toolkit¹¹⁷ (v1.0.8-r237) by extracting reads aligning to canonical chr6 HLA region (chr6: 25,500,000 to 34,000,000) and all alternative HLA contigs in the hg38 reference. These alignments were streamed with the GATK PrintReads commands, using a .bed file of these coordinates, and streamed into the T1K genotyper with default parameters. Following T1K toolkit recommendations, the donor haplotypes were assigned for alleles called with a quality score >0. Homozygotes were determined by donors with only a single allele called and with a quality score >30.

Additionally, as part of our online code resources, we provide a reproducible Dockerfile and Docker image for the T1K genotyping workflow that we developed for calling HLA alleles in AoU.

12. Line 661-662: Please label the T1K software properly, it should be all caps. Find elsewhere in manuscript and correct.

We have revised the text accordingly.

13. Line 673: Grammar.

We have revised the text accordingly.

14. Please consistently refer to the UKB and AoU cohorts in the same way throughout the paper.

We appreciate the reviewer catching our inconsistent use and have corrected the manuscript such that all mentions of the UK Biobank cohort are referred to as “UKB” and the All of Us cohort as “AoU.”

REFERENCES

1. Münz, C. Latency and lytic replication in Epstein–Barr virus-associated oncogenesis. *Nature Reviews Microbiology* 17, 691-700 (2019).
2. Sabel, C.E., et al. The latitude gradient for multiple sclerosis prevalence is established in the early life course. *Brain* 144, 2038-2046 (2021).

Referee #3 (Remarks on code availability):

Review of code and bioinformatic workflow

README.md:

- The information regarding the use of GATK to access the bam files on AoU hosted-data is satisfactory
- Please include the “samtools view” parameters and values used to access locally stored UKB bam files, such that a user could apply the same command to their data and generate the same output you did.
 - It is unclear what is the desired output of the bam files. Presumably fastq files?
 - And if the desired output is fastq files, there are more steps required to generate fastq from bam than just samtools view. Please describe these commands and the parameters used.

We apologize for the confusion here. The output of `samtools view` is simply another .bam file with only the chrEBV reads. We have now clarified this within the main README in our reproducibility GitHub (https://github.com/clareaulab/ebv_biobank_gwas):

```
samtools view Donor_*****.wgs.bam chrEBV -b -o Donor_*****.ebv.bam
```

As we document in the README, no other steps on the command line are required, and the downstream processing to quantify EBV DNA reads from these bam files are provided in Jupyter notebooks (e.g., all-of-us-notebooks/EBV_DNA_Quantification/01_Quantify_EBV_DNA.ipynb). Related to the next point from the reviewer, we now provide additional reproducibility and working examples for the 1000 Genomes cohort that we anticipate will allow for users to apply to their own data.

- While I understand that due to the data sharing limitations, it is not possible to provide the actual source data for many of the analysis stages, it is still essential to describe the conceptual workflow used to go from .bam files to the input for each of the R scripts. As it stands, it is impossible for the user to make that jump. Please add a conceptual workflow to the README, that describes each stage of the analysis and the data generated at each step, up to the point of input for the R scripts.
 - Ideally, the authors would create an example dataset from simulated short reads using both the human and EBV genome, which would allow them to share example data at each step. ArtSim (Huang et al., 2011) is a simple and convenient way of doing this for NGS reads: <https://www.niehs.nih.gov/research/resources/software/biostatistics/art>

We appreciate the concrete suggestion to improve reproducibility given the constraints of data sharing. In preparing the simulated dataset, we found that hosting/sharing simulated data at ~30x coverage would be prohibitive. As an alternative solution, we identified a WGS dataset from the 1000 Genomes Project that is available within a public Terra workspace, enabling users to readily quantify EBV DNA retrospectively (<https://app.terra.bio/#workspaces/anvil-datastorage/1000G-high-coverage-2019>). Specifically, any user can access the input .cram files irrespective of data permissions to reproduce the core computational innovation presented in our analysis. We note that the new notebook that produces EBV DNA counts from the 1000 Genomes Project (also hosted on Terra/GCP) would then directly serve as input for the downstream analyses (i.e., GWAS and PheWAS) detailed in the AoU notebooks folder.

We have documented the additional 1000G analyses and workflow in the README of our manuscript GitHub (https://github.com/clareaulab/ebv_biobank_gwas) and provide a full .ipynb for streaming the .cram data: https://github.com/clareaulab/ebv_biobank_gwas/blob/main/mwe-1000G/01_1000G_EBV.ipynb. We appreciate

the emphasis on ensuring a reproducible workflow even with data sharing limitations, and we anticipate that our solution will be useful for individuals seeking to reproduce and/or extend our findings.

- Since one of the claims of the manuscript is that the authors “demonstrate how existing WGS data can derive novel molecular phenotypes, which may generalize to hundreds of viruses comprising the blood virome”, it is essential that they provide a clear bioinformatic workflow that others can reproduce from start to finish. While the R scripts are overall very clean well annotated, there is currently no way for the reader/user to know how to generate the required input files for the analysis from the bam files described in the README.

We again appreciate the detailed and specific request to ensure that our workflow is fully reproducible. By offering a streamlined example via the publicly-available 1000 Genomes Project, we anticipate this will ensure that the bioinformatic workflow is reproducible for new users given the constraints on data access for other biobanks.

celltype-pathway-mapping/code/00_ref_project.R

- *Could not run*
- Line 4: This is not available to install via the standard R CRAN repository. Please add installation commands to script.
 - Library(BuenColors)
- Line 13: Missing file-
 - `ref_path <- "~/Dropbox/main_papers/pearson/pearson_large_data_files/input/pbmc/pbmc_multimodal.h5seurat"`
 - This is presumably a private directory belonging to the authors that a public user cannot access.
 - There is no description of this file provided for a user to be able to provide one themselves
- Line 19: Missing file -
`ebv_hits <- (fread("../data/ebv_hits.txt", header = FALSE)[[1]])`

celltype-pathway-mapping/code/ 01_genesetanalysis.R

- *Could not run*
- Line 1-7: These are not available to install via the standard R CRAN repository. Please add installation commands to script.
 - `library(BuenColors)`
 - `library(clusterProfiler)`
 - `library(ReactomePA)`
 - `library(clusterProfiler)` (This is a repeat)
 - `library(org.Hs.eg.db)`
 - `library(annotables)`
- Line 30: Missing file: `all_hits <- readLines("../data/ebv_hits.txt")`

We appreciate the reviewer’s attention to detail. For all packages that were noted as unavailable via the standard R CRAN repository, we have added the installation code within the README (e.g., `devtools::install_github("caleblareau/BuenColors")`).

We have also added the information needed to access the `pbmc_multimodal.h5seurat` file, which is available at a URL provided by the original authors (https://atlas.fredhutch.org/data/nygc/multimodal/pbmc_multimodal.h5seurat).

Finally, we have added the list of EBV gene hits (`ebv_hits.txt`) into the `data` folder.

Together, these changes should mitigate any issues with reproducing these scripts.

epitope-scoring/code/01_test_sliding_peptides.R

- Successful run
- `library(Biostrings)`: This is not available to install via the standard R CRAN repository. Please add installation commands to script.

epitope-scoring/code/02_plot_epitopes.R

- *Could not run*
- Line 1-2:
 - `library(BuenColors)`
 - `library(rtracklayer)`

These are not available to install via the standard R CRAN repository. Please add installation commands to script.

- Line 10: Missing input file: `"../output/full_EBV_annotation.rds"`

epitope-scoring/code/03_enrich_epitopes.R

- Successful run + Warning message
- Line 1-2: These are not available to install via the standard R CRAN repository. Please add installation commands to script.
- `library(BuenColors)`
- `library(Biostrings)`
- Output warning messages:

```
> class1_summary$pvalue <- sapply(1:4, function(i) {
  prop.test(class1_summary$total_br[i],
            sum(class1_summary$total_br),
            p = class1_summary$prop_total[i])$p.value
})
```

Warning message:

In `prop.test(class1_summary$total_br[i], sum(class1_summary$total_br), :`
Chi-squared approximation may be incorrect

```
> class2_summary$pvalue <- sapply(1:4, function(i) {
  prop.test(class2_summary$total_br[i],
            sum(class2_summary$total_br),
            p = class2_summary$prop_total[i])$p.value
})
```

Warning message:

In `prop.test(class2_summary$total_br[i], sum(class2_summary$total_br), :`
Chi-squared approximation may be incorrect

For all packages that were noted as unavailable via the standard R CRAN repository, we have added the installation code within the README (e.g., `BiocManager::install("Biostrings")`). We have also deleted the reference to the `full_EBV_annotation.rds` input file, as it is not used within the script. For the warning messages, these do not impact the code execution or results.

viral-sequences/code/01_type1_type2.R

- Successful run
- Line 4: This is not available to install via the standard R CRAN repository. Please add installation commands to script.
 - `library(BuenColors)`

viral-sequences/code/02_overall_mafs.R

- Successful run
- Line 1, 4: These are not available to install via the standard R CRAN repository. Please add installation commands to script.
 - `library(Biostrings)`
 - `library(BuenColors)`

viral-sequences/code/03_top_VUS.R

- Successful run
- Line 2: This is not available to install via the standard R CRAN repository. Please add installation commands to script.
 - `library(BuenColors)`

viral-sequences/code/11_munge_VEP.R

- Successful run

viral-sequences/code/13_epitopes.R

- Successful run

For all packages that were noted as unavailable via the standard R CRAN repository, we have added the installation code as comments in the updated scripts (e.g., `BiocManager::install("Biostrings")`).

Referee #4 (Remarks to the Author):

I co-reviewed this manuscript with one of the reviewers who provided the listed reports.

We sincerely appreciate the careful attention to our code and reproducibility, with the detailed and specific feedback. We expect all code will run with the specified requirements now documented.

Referee #5 (Remarks to the Author):

Nyeo et al present a piece of technical work undertaking large-scale analyses of Epstein-Barr virus (EBV) signatures in cohorts of large biomedical databases, such as UK Biobank and All of US. In general, the work is well done in terms of analyses and the methods are valid, while the sample sizes are impressive. However, one major limitation of the work is that the results are not novel, nor particularly impactful. Most of the results described are well-known associations of EBV with diseases, and if there are additional novel results, the authors have not presented these clearly enough.

We appreciate the recognition of the quality of our analyses and the overall insights from our work. We respectively disagree about the novelty and impact of our results. In particular, we highlight key aspects of findings that, to the best of our knowledge, have not been demonstrated in other contexts:

1. We demonstrate that *existing* WGS data can be engineered as a molecular biomarker for endemic viral infections. This is a key conceptual and methodological advance presented in our paper – one that can be readily adopted to cost-effectively study many other viruses (now clarified in the **Discussion**) or applied in other biobanks. To that end, and incorporating suggestions from additional reviewers, we have enhanced the accessibility of our codebase, enabling future studies that investigate correlates and determinants of viral infection using population-scale genome sequence data.
2. Using our reconstruction of EBV DNAemia from existing WGS data, we demonstrate that the abundance of EBV DNA at the population level is a polygenic trait that is regulated by 20+ distinct genetic loci. To the best of our knowledge, this is a novel finding that we anticipate will enable many future studies at the intersection of systems virology and human genetics. Notably, we estimate the h_g^2 (SNP-based heritability) to be 2.2%. For context, this estimate is similar to the h_g^2 of susceptibility to various infections (derived from associated ICD10 codes) reported in population-based registers (e.g., Denmark²⁸) and in severity of COVID-19 to SARS-CoV-2 infection²⁹. Therefore, our characterization of viral DNA as a polygenic trait and characterization of associated loci represent a conceptually new direction for the field and is in line with other adjacent complex trait phenotypes.
3. In our original submission we were purposefully conservative by focusing the reporting on **reproducible** traits between two large independent biobanks. As we now report in **Extended Data Table 2** and **Reviewer Fig. 22** below, many additional statistically significant associations were identified in the UKB cohort that we could not readily map to the more limited AoU phenotype definitions, either due to challenges with how human phenotypes are organized in AoU or its limited sample sizes across rarer indications. Hence, this collection of associations identified from our discovery cohort can be further investigated in future studies.

Related to these additional specific points, we have revised our manuscript to expand on the novelty of our work, including featuring more salient results. We appreciate the detailed feedback in the points below and have revised our manuscript with these elements accordingly.

The second major issue is that there is hardly a discussion section that could help the reader put the findings into context and interpret them. Not providing adequate context is a recurring problem in the manuscript, frequently having no relevant information nor rationale provided. As just one example, the epitopes section is not well explained (also Figure 5), and therefore following the results is not straightforward.

We have now greatly revised the manuscript, particularly in the results and discussion sections of the revised text, and we provide concrete revisions of these points below.

As a second example, how and why dendritic cells (line 244, figure 4) are involved in EBV latency; the authors could provide more context and discuss more on why the findings are interesting for EBV biology and disease. For more examples, see specific points below. Sometimes it is also unclear whether the authors are using results from alternative datasets or indeed reanalysing these data themselves. In general, the paper does not seem written for a general audience, and I think the fact that there is no actual discussion, while the methods are better written than the results presentation further highlights this.

We appreciate the feedback on how to improve sections that were unclear. In our revised version, we now provide additional context and explanations.

More specifically, I have the following major points:

1) Page 3, Lines 36-68: Describe the meaning/importance of EBV latent infection and provide information on relevant EBV biology.

We appreciate the excellent suggestion to define and comment on the importance of EBV latency. We have now substantially re-written the introduction of the manuscript to better reflect the meaning and importance of EBV latent infection:

[Page 3, Lines 46-53] The underlying causes behind the clinical heterogeneity of EBV infection – ranging from asymptomatic infection to severe disease – remain incompletely understood. The most severe manifestation of infections are EBV-triggered cancers that collectively account for 130,000-200,000 deaths worldwide each year⁹. Conversely, immunocompetent individuals may harbor EBV DNA that persists in latency within peripheral memory B cells, in which the virus expresses a minimal gene program^{4,10-12}. Characteristic of latent viral infections, EBV can reactivate sporadically or in response to acute stressors or host immunosuppression, resulting in expanded viral reservoirs and potentially lethal clinical complications^{10,13}.

2) Lines 106-107: Given that 90% of the population are EBV-seropositive, but EBV DNA is detected in only 9.7% of the UK biobank's (and 11.9% in AoU) participants, what is the importance of this detection? In the methods (Line 462) the authors state that "We interpret our measure to reflect the tail of latent viral retention among individuals for whom this is highest", however they do not offer a view of what this highest latent EBV retention really means, and this should be contextualised in the results and discussion.

To better convey this idea, we have now added an explicit simulation analysis (the results shown above in **Reviewer Figure 5**) that demonstrates an underlying basis for what we are detecting. In brief, our simulation showcases that our empirical distribution of EBV DNAemia can likely be captured by an underlying log-normal distribution of EBV copy number, such that the DNA+ population represents the right tail of the distribution, reflecting the individuals with the highest latent DNA levels. Therefore, our study is effectively attempting to characterize what genetic and phenotypic associations are linked to this right tail of the distribution relative to others in the population. The fraction of the population that is positive primarily reflects individuals with EBV DNA loads that are above the limit of detection when exploiting existing WGS data to study persistent EBV infection.

Reviewer Figure 5 (reproduced from above). Simulations of EBV DNAemia under a censored log-normal distribution. **(d)** Simulated data of a mixture of 10% 0 EBV and 90% log-normal EBV. The dotted line indicates the threshold for data censoring. **(e)** Result of data censoring on simulated data. **(f)** Empirical distribution of observed EBV levels. **(g)** Comparison of donor positivity from simulated and observed EBV levels. The threshold of 1.2 EBV genomes per 10^4 human cells was chosen in the manuscript for downstream analyses, and the results are consistent with the simulation even at high levels of latent EBV DNA detection.

Interpreting this simulation further (panel **e**), if one were to sequence $\sim 100x$ deeper per whole genome (requiring an infeasible coverage of $\sim 3,000x$ per donor), we would likely identify $\sim 85-90\%$ of the UKB cohort as DNA+, as our limit of detection would be improved by two orders of magnitude. We now clarify this point further in the revised text:

[Page 4-5, Lines 117-124] Though the proportion of individuals with EBV DNAemia (9.7%) is lower than the rate of seropositivity, we interpret our measure to reflect the tail of individuals for whom persistent EBV DNA is highest. Indeed, simulated data from a censored log-normal distribution of per-person EBV DNA levels closely approximated the empirical distribution, a framework motivated by prior viral copy number statistical analyses^{27,28} (**Extended Data Fig. 1d-g; Methods**). While our simulation model does not test a quantitative hypothesis, the result illustrates how a relatively simple model of EBV DNA abundance can explain the underlying distribution of our observed EBV copy numbers, even at high levels.

[Page 14, Lines 470-477] Despite $\sim 90\%$ EBV seropositivity among adults in the UK and US, we identify a distinct population of 9.7-11.9% of individuals with detectable EBV DNA in peripheral blood, suggesting that past infection is necessary but not sufficient for DNAemia. Instead, our simulations imply that the EBV DNAemia population reflects a tail of exposed individuals that have the highest latent EBV DNA levels (**Fig. 1e,f; Extended Data Fig. 1d-g**). We hypothesize that others in these populations are carriers of EBV DNA from past infections, but at levels below our limit of detection, and we estimate that sequencing $\sim 100x$ deeper per whole genome would detect EBV DNA in $\sim 85-90\%$ of individuals, which could further enable additional studies of latent EBV DNA as a quantitative biomarker.

[Page 31, Lines 704-714] To better characterize the discrepancy where $\sim 90\%$ of individuals in UKB are seropositive, yet we detect non-zero DNA levels in only 14.3%, we conducted a simulation study. Using maximum likelihood estimation, we estimated values for the mean and standard deviation of a log-normal distribution to initialize the simulation and subsequently modified these values to account for 1) the mixture with 10% 0s (representing the uninfected individuals) and 2) adjusted the mean for a round, interpretable number. The final values used in the simulation shown in **Extended Data Fig. 1d**, with a 10% forced 0 rate for 500,00 individuals, and the remaining 450,000 individuals were simulated via a log-normal distribution, with a mean of 0.2 EBV genome copies per 10,000 cells, a standard deviation of 0.62, and a censored value of 0.71. We emphasize that this simulation does not test an explicit statistical question but is designed primarily for illustrative purposes, to show that a single underlying component can explain many features of the empirical data (rather than requiring a second component).

3) Relevant to the point above, the authors should comment on whether the ratio of EBV-infected B lymphocytes versus non-infected B lymphocytes is what is driving the EBV DNA detection and its associations with disease and comment on its importance.

We appreciate the suggestion to explain this phenomenon. First, we note that we have no specific insight on whether this ratio is increasing from the data analyzed in this work, as bulk WGS does not allow for deconvoluting the relative cell type proportions. Further, B lymphocyte count is not a variable measured in UKB or AoJ. Additionally, despite our best abilities to quantify EBV in single cells (e.g., OneK1K), we did not detect sufficient infected cell numbers to make a meaningful comment on this possibility from the data. Hence, we refer to the literature, which indeed suggests that an increased number of latency-infected cells (among immunosuppressed individuals) would explain EBV DNAemia³³. We have added this commentary in our new **Discussion** section:

[Page 14, Lines 478-487] *Prior studies in immunosuppressed populations indicate that EBV DNAemia can be attributed to the abundance of latently infected B cells in a sample¹³, which we anticipate drives our detection of EBV DNA across our cohorts.*

4) By using DNA-seq data, the authors cannot differentiate between EBV latent or lytic infection, so they interpret everything as EBV latent infection. However, arguably a proportion of this EBV DNA signature will originate from lytic infections in individuals with various illnesses, that cannot be separated from latent cases from the data. Lytic infections could be present due to primary infection (ie active infectious mononucleosis) and also reactivation which can happen in individuals under stress, illness, or immunosuppression. For example, while the authors acknowledge the presence of mononucleosis cases (lines 153-159), they fail to address how the distinction between lytic and latent infections could impact their conclusions.

Based on the additional analyses we have conducted (as detailed in **Reviewer Figures 5** above; **Reviewer Figures 6 and 7** below), we suggest that the overwhelming majority of EBV DNAemia cases indeed reflect viral latency and not lytic infection. For example, and as outlined in earlier responses, when studying samples from the GTEx resource that have both WGS and paired RNA-seq data, we found that only 1.7% of individuals with any detectable EBV DNAemia had RNA-seq evidence of being lytic (all of which had only 1 sequencing read; **Reviewer Fig. 6c**).

Reviewer Figure 6 (reproduced from above). Characterization of potential EBV reactivation in large genomics datasets. (a) Schematic for GTEx consortium, highlighting donors where matching WGS and RNA-seq data are available from peripheral blood. (b) Sum of per-base read coverage of high-confidence EBV-mapping reads. Two repetitive regions with inflated coverage are noted in purple and red. (c) Characterization of EBV detection from 681 GTEx donors with paired WGS and RNA-seq from peripheral blood. (d) Schematic of data generation from OneK1K cohort. (e) Summary of EBV quantification from OneK1K cohort, showing total read number from the consortium (top) and number of EBV-assigned UMIs. (f) Summary of scRNA-seq results from a lymphoblastic cell line (LCL), a positive control for EBV transformation. (g) Summary of scRNA-seq results from a kidney transplant, a positive control for EBV reactivation.

We also characterize the longitudinal serology responses against EBV antigens stratified by EBV DNAemia, and we observe no evidence of variable antibody responses that would result from recent reactivation or infection from individuals with EBV DNAemia (**Reviewer Fig. 7**).

Reviewer Figure 7 (reproduced from above). Longitudinal serology analyses from UKB. (h) Depiction of longitudinal serology measures of four EBV antigens for an exemplar EBV DNA+ donor. (i) Characterization of EBV longitudinal antigen titers across UKB. Statistical test: two-sided Student's *t*-test.

Nevertheless, we acknowledge that we cannot exclude lytic infection in every individual. Hence, we summarize our new section, “EBV DNAemia likely reflects latent viral infection,” with the following statement:

[Page 6, Lines 171-174] *While a proportion of the 735,954 donors in UKB and AoU were likely experiencing lytic infection or reactivation during blood sampling, these analyses suggest that EBV DNA captured by WGS of population biobanks largely reflects latent infection, even for instances with high viral DNA measures.*

This presence of lytic infection is further demonstrated by the authors’ own results, see Line 132: “Significant quantitative associations (n = 156) included detection of two EBV antigens”. One of these two antigens is ZEBRA (Extended Figure 2a), which is a marker of lytic acute infection. The authors should therefore acknowledge and discuss this, namely that the EBV DNAemia signature is not only due to latent infection but rather in some instances due to lytic infection. Given this and the known literature around EBV reactivation in several settings (under stress, several health conditions, immunosuppression), they should reflect on why they detect only 1 transcript in their single cell RNA-seq data from the 1000 healthy blood donors cohort. I think the most likely explanation is that this was a cohort of healthy blood donors who already had their primary EBV infection at younger age (cohort individuals older than 19 years old) and who did not have EBV reactivation at the time of sampling as they were donating blood while healthy – in contrast the UK Biobank and All of US have large patient groups where reactivation could indeed occur.

We appreciate the consideration regarding the latent and lytic-derived viral DNA. We now better qualify that in some cases we may be detecting lytic DNA:

[Page 14, Lines 478-487] *As EBV is an endemic pathogen, a subset of the >735,000 individuals in UKB and AoU were likely experiencing lytic EBV infection or viral reactivation at the time of blood sampling. However, extensive blood transcriptomic analyses of 1,663 donors from GTEx and OneK1K detected minimal evidence of EBV RNA transcripts, suggesting that the vast majority of blood-derived EBV DNA from our cohorts reflects latent rather than lytic infection. Prior studies in immunosuppressed populations indicate that EBV DNAemia can be attributed to the abundance of latently infected B cells in a sample¹³, which we anticipate drives our detection of EBV DNA across our cohorts. As our per-donor WGS measurements represent a snapshot of fluctuating viral DNA levels over a lifetime, longitudinal profiles would likely refine our EBV DNAemia phenotyping and power further discovery of genetic loci that underlie population variation in viral latency and persistence.*

Regarding the reviewer's point of the OneK1K cohort composition, the blood samples were collected from both female (58%) and male donors (42%), whose ages ranged from 19 to 97 years old, with 73% of participants being older than 60 years old (see Supplementary Materials from the original manuscript³⁶). Notably, this OneK1K cohort skews older than either AoU or UKB, and as EBV reactivation increases in frequency at older ages³⁷, we suggest that this cohort is a fair, if not conservative, comparison to assess frequency of reactivation at sampling time.

Ultimately, with our further characterization of the GTEx cohort (**Reviewer Fig. 6**), we conclude that individuals with elevated levels of EBV DNA can be present in the population without any clear molecular indication of lytic infection or reactivation. The OneK1K cohort provides a large sample size to substantiate our conclusion.

Additionally for the single-cell RNA-seq study of the 1000 blood donors, was the library prep/sequencing protocol suitable to retain non-human RNA transcripts, or could there be a bias introduced there as well?

In our revised manuscript we have added several analyses in response to this point. First, we note that most EBV (and other herpesvirus) transcripts are polyadenylated³⁸, which allows for detection via standard 10x Genomics scRNA-seq library preparation, as was performed in the OneK1K cohort. Indeed, our previous work characterizing HHV-6 reactivation in T cells used exclusively 10x scRNA-seq detection with poly-A capture chemistry, which yielded sensitive and specific capture of herpesvirus transcripts^{7,38}. We further note that 10.3% of cells profiled in this cohort were B cells, reflecting ample power to detect EBV reactivation if it occurred at an appreciable frequency.

To showcase this point, we have now added additional analyses in **Reviewer Figure 6e,f** that demonstrate sensitive EBV transcript detection using the same technology in LCLs and kidney transplants, where cells are transformed with EBV or possibly undergoing lytic reactivation, to verify the true positive detection of EBV via these technologies. Hence, by providing these additional analyses as "positive controls," we can more confidently interpret the lack of EBV transcripts in the OneK1K cohort as a true lack of lytic reactivation. Finally, we also note that our new analyses of the independent GTEx data, which directly pairs DNA and RNA sequencing from the same blood draws for 681 donors, show minimal EBV RNA (i.e, 3 total paired-end reads from RNA from 50+ billion reads among all donors).

5) Line 101-102: Relevant to the point above, the authors should clarify whether they undertook complementary analyses for detecting EBV transcripts? It seems that way from methods but in the main text it is ambiguous whether this is novel analyses or published in the study referenced.

We indeed performed the complementary analyses to detect EBV transcripts from the OneK1K cohort (single-cell RNA-seq) and GTEx consortium (bulk RNA with matching WGS). Now that we have added a

new section to the manuscript, as well as extended figures to characterize our transcriptional analyses from these two large cohorts, we hope that our work quantifying viral RNA is clarified.

6) Lines 120-159: The clinical findings of associations with various illnesses are not novel, as already stated the technique is interesting and the sample sizes are noteworthy, however if there are novel findings, they are not demonstrated properly.

We appreciate the opportunity to clarify our rationale for reporting and highlighting various phenotypes from our association studies. In brief, we sought to be conservative in our interpretation of the PheWAS results, since we intended to use these phenotypic associations as additional corroboration of the EBV DNAemia biomarker derived from WGS. As the reviewer noted, the top associations were reassuringly enriched for very well-established indications (such as SLE, rheumatoid arthritis, COPD). However, several of the study-wide significant associations have been long suspected and anecdotally cited but not well-established, likely due to the limited sample sizes (and thus statistical power) of prior studies.

With the scale of our study, we were able to statistically validate some speculated associations and identify new ones. For example, to the best of our knowledge, the association with COPD has only been reported once before, based on a study of sputum samples from ~120 patients³⁹. Other associations have been anecdotally reported as being associated with acute EBV infection via small case studies, but never with chronic EBV infection, such as the association between EBV and increased incidence of chronic ischemic heart disease (OR = 1.19; $P = 2.8 \times 10^{-18}$), acute kidney failure (OR = 1.21; $P = 1.4 \times 10^{-16}$), stroke (OR = 1.20; $P = 6.1 \times 10^{-13}$).

To complement the study-wide significant associations, we also provide the full summary statistics from this first phenome-wide population-scale scan to provide the medical research community with a rich and systematic resource for future functional follow-up. In particular, we now add a new figure (**Reviewer Figure 20; Extended Data Fig. 4c**) that highlights various low-frequency clinical phenotypes that have been able to achieve nominal significance with EBV ($P < 0.05$). While some of these associations could be secondary correlates rather than causally associated, we note that our past efforts of utilizing biobanks with rare variants and/or rare diseases led to novel associations that, through functional follow-up and validation by the rest of the community, has led to discoveries for novel drug targets⁴⁰⁻⁴³. Hence, we agree with the reviewer that highlighting additional novel anecdotes are worthwhile, and we have done so in a careful manner within our revised version of the text:

*[Page 7, Lines 220-227] In addition to phenotypes that replicated between cohorts, we noted instances of neurological conditions that were nominally associated with EBV DNAemia in UKB but lacked case numbers to be assessed in AoU ($P < 0.05$; **Extended Data Fig. 4c**). These included all-cause dementia (OR = 1.16; $P = 6.0 \times 10^{-5}$), as well as rarer phenotypes like neuromyelitis optica⁴³, a rare autoimmune disease with similar clinical presentation as MS (OR = 6.31; $P = 2.7 \times 10^{-3}$), and acute disseminated demyelination (OR = 6.31; $P = 5.3 \times 10^{-3}$). While further work is required to establish a causative role for EBV in these phenotypes, our scalable approach enables systematic associations with a large set of conditions, including rare diseases that require large sample sizes as in UKB and AoU.*

Reviewer Figure 22 (manuscript Extended Data Fig. 4c). Characterization of PheWAS associations with the highest odds ratio in UKB. (c) Top neurological traits from UKB PheWAS, sorted by odds ratio.

7) The Mustafa et al. reference in the abstract does not support the statement for hundreds of viruses comprising the blood virome. In fact, in the Mustafa et al study cited in the abstract, the authors state explicitly that out of 94 viruses, only 19 are not contaminants. From Mustafa et al abstract: "... we mapped sequences to 94 different viruses, including sequences from 19 human DNA viruses, proviruses and RNA viruses... The remaining 75 viruses mostly reflect extensive contamination of commercial reagents and from the environment". In any case, the work undertaken in the Nyeo et al study cannot be replicated for the majority of viruses that do not establish latency - the authors should discuss the instances where this would be feasible e.g herpesviruses.

We have adjusted our language accordingly. Namely, we have changed our abstract to say *the breadth of viruses* rather than *hundreds of viruses*, as this better reflects the previous work on the blood virome. We note that *thousands* of anelloviruses have been detected in peripheral blood, yet the mapping heuristic in Mustafa et al. collapses these into a single entry in their results, resulting in only the 19 species that the reviewer pointed out. Therefore, rather than saying a specific number, we refined the text to reflect the "breadth" of possible viruses to study and have added a discussion of other species that may be detectable upon further analyses, as suggested:

[Page 14, Lines 490-495] *A notable limitation of blood-derived WGS data is that detection of viruses will require tropism in blood and the possibility of viral latency. Hence, we expect that viruses from the Herpesviridae, Polyomaviridae, Adenoviridae, Parvoviridae, and Anelloviridae families could be characterized with existing datasets, as these viral families are known to maintain persistence in peripheral blood over a lifetime¹⁰². Future cohorts that pair DNA and RNA-sequencing across diverse anatomic regions will be required to fully characterize the vast variation in host response to other endemic pathogens, including RNA viruses.*

8) Line 49-56: "Beyond its role in human disease...caused by infection". If there were important findings relevant to EBV from these large genetic cohorts what were these – the authors do not make it clear. Instead, the authors should use this space to clarify further and emphasise EBV contributions to human mortality and morbidity.

We apologize for the confusion in this statement. Our point was that EBV was used as an agent that could transform/immortalize primary human cells for long-term storage as lymphoblastoid cell lines (LCLs). The transformation via EBV enabled many critical studies of human genetic diversity, but the focus was not on genetic regulation of EBV. The critical concept is that since these studies used EBV for the immortalization, the EBV reference genome is included in the hg38 and subsequent reference genomes. We have now revised these lines for clarity:

[Page 3, Lines 59-63] EBV can transform primary B lymphocytes from healthy individuals into immortalized lymphoblastoid cell lines (LCLs)¹⁵, critical resources that enable long-term storage and large-scale genetic studies. Consequently, immortalized LCLs were the primary material used in the HapMap¹⁶, 1000 Genomes¹⁷, and Geuvadis¹⁸ Projects, which applied DNA and RNA sequencing to profile diverse global cohorts.

Related to Reviewer 5, Point 1 and the comment here about EBV contributions to mortality, we have added this detail to our newly revised introduction:

[Page 3, Lines 48-49] The most severe manifestation of infections are EBV-triggered cancers that collectively account for 130,000-200,000 deaths worldwide each year⁹.

We hope these details help with further contextualizing and motivating the key biological questions regarding variations in host responses to latent EBV infection.

9) Line 93: What is the novel molecular phenotype?

We have revised our text accordingly – the term was intended to refer to the EBV DNAemia phenotype.

10) Line 125-126: How did the authors select these phenotypes? Were these all the available phenotypes in the UK BioBank or did some selection take place?

We adopted the same approach as we described in our prior work characterizing rare variant associations with complex traits⁴⁹. In brief, we curated phecodes extensively to maximise phenotypic representation. Binary phenotypes included those derived from ICD 10 codes, self-reported survey data, and death or cancer registry data. We also constructed “union” phenotypes to unify similar phenotypes across these different data sources. Quantitative phenotypes included those derived from the baseline blood, urine, and biometric data. In the revised version of the manuscript, we now include additional details on the PheWAS in the results and include a summary reference appendix listing all tested phenotypes in our revised **Extended Data Table 2a**.

11) Line 126 onwards: the authors should provide more context for the significant associations they detect and replicate in the two cohorts. Relevant to that, in Figure 2a: how did they choose which associations to label, including both relevant and not-relevant EBV conditions. Furthermore, the associations in the respiratory group seem more pronounced compared to other categories, what is the authors' view on this?

We appreciate the suggestion to provide more context for reproduced and notable phenotypes. In **Fig. 2a**, we plotted the most significant association for each ICD 10 chapter to emphasize the wide-ranging associations of EBV DNAemia. We additionally intentionally annotated phenotypes that have been suggested to have EBV associations in prior literature (e.g., systemic lupus erythematosus, rheumatoid arthritis, etc.). To address these points, which were also suggested by Reviewer 3, we have rewritten these phenotypic and genetic association sections to explicitly characterize effects that have been replicated. We highlight relevant text excerpts here:

[Page 7, Lines 200-204] We sought to replicate these associations using the AoU cohort (**Fig. 2b**). As the underlying EHR data between cohorts vary, we focused on 141 of the significantly associated matching ICD10 codes in UKB with sufficient representation in the AoU cohort for replication analyses (minimum n = 24 cases). 87 (62%) of these matching codes replicated in AoU (P < 0.05; OR concordant with UKB statistics), resulting in a set of traits that we examined more closely (**Methods; Extended Data Table 3**).

Regarding the respiratory chapter associations, we re-evaluated this more carefully and observed that most of these associations were driven by COPD and emphysema, as well as related complications like pneumonia that simply have many more applicable phecodes documented in the UKB EHR. In our revised manuscript, we now provide all PheWAS associations (**Extended Data Table 2a**) with better annotations to aid in this interpretation, noting that the prominence of the respiratory chapter is partly attributed to correlated phecodes.

Nevertheless, we emphasize that our respiratory chapter results still have potential value in corroborating known associations. As one example, a prior study with a much smaller sample size ($n = 136$) found that EBV levels were significantly higher in sputum from smokers with COPD vs. unobstructed smokers³⁹. Our data not only replicate this signal, but provide further evidence of one potential way EBV DNAemia could be used as a predictive biomarker.

12) Line 161, Section Polygenic Variation: As it stands, the rationale for and the findings from this work are unclear. Do the authors claim they are showing that genetic variation is the reason for individuals to respond differently to EBV infection? Relevant to that, there is really no discussion section to contextualise any of the results.

In our manuscript, we estimate the h_g^2 (SNP-based heritability) to be 2.2%. This indicates that genetic variation is a component of the population-level variation in response to EBV infection, but host genetic factors are insufficient to fully explain the underlying variation in EBV DNAemia.

For context, this estimate is consistent with other complex traits, including responses to other infections:

- Our estimated h_g^2 of susceptibility to various infections reported in population-based registers (e.g., Denmark²⁸).
- The heritability of severity of COVID-19 following SARS-CoV-2 infection was a similar value (1.2-5.8%)²⁹.

We clarify our interpretation of the polygenic signal with additional discussion in the revised manuscript:

[Page 8, Lines 235-242] Overall, the SNP-based heritability (h^2) determined by LDscore Regression (LDSC) was 2.21% ($\pm 0.85\%$) with limited evidence of genomic inflation ($\lambda_{GC} 1.1$; LDSC intercept = 1.03 ± 0.008). For context, this SNP-based heritability estimate is similar in magnitude to prior reports of susceptibility to a broad class of viruses in population health registries⁴⁶, as well as the heritability of severe COVID-19 following infection from SARS-CoV-2⁴⁷. In addition, partitioned heritability analyses showed an enrichment at conserved and non-coding loci marked by enhancer and super-enhancer annotations (**Extended Data Fig. 5a**), consistent with other complex trait associations.^{48,49}

13) Line 222-237 – Pleiotropy with complex diseases. This section is difficult to follow, and the results do not make sense as presented. As an example, Figure 3b is unclear, what has been done and what exactly does it show? What do the p-values refer to? Also is Crohn's and ulcerative colitis “seronegative” because there were no associations found with EBV DNA viraemia? Finally, what is “latent EBV” in this figure – is it a category in UKB or something the authors have defined?

We have revised **Fig. 3d** for clarity.

First, we have renamed “latent EBV” to EBV DNAemia to be consistent with the terminology used throughout the rest of the paper. Next, “seropositive” and “seronegative” reflect whether a disease is known

to be associated with highly disease-specific pathogenic autoantibodies, which we now clarify in the revised main text and methods.

For additional statistical corroboration of our results, we include the All of Us GWAS summary statistics as an independent trait and observe a highly-consistent effect and significant result with the UKB statistics.

Reviewer Figure 10 (reproduced from above). Revised analyses of genetic architecture overlap. The independent AoU architecture was added to replicate the direction and effect of the EBV result clustering with seropositive IMDs.

In response to these questions and those of Reviewer #2, we have updated the main text and Methods to better explain these results, including an explanation of what the cupcake principal component scores represent and how statistical inference (i.e., P values) are derived:

[Page 36, Lines 900-907] The output from this projection is a score or delta (δ) for each principal component that quantifies the difference between the projected genetic risk for that trait on a particular basis axis and a synthetic control (which has zero effect sizes for all SNPs). This effectively measures how strongly the trait aligns with the risk architecture represented by that component. To account for uncertainty, the variance of δ is calculated using the propagation of error from the input GWAS summary statistics, adjusted for the same shrinkage weights and allele frequency variance as applied in basis construction. With δ and its variance, a z-statistic can be formed for each component, and standard statistical inference can be used to compute a P value⁷⁸.

14) Line 245 – what are EDs?

This was a typographical error intended to refer to DCs (dendritic cells). We have adjusted the text accordingly.

15) Line 256: This is not a new conclusion

In our revised manuscript, we have qualified this result to state that our conclusion is consistent with the known role of B cells and dendritic cells:

[Page 12, Lines 408-409] *consistent with the role of CD4-mediated immunity of viral infections via class II antigen presentation by B cells and DCs⁸⁴.*

16) Line 321: What do these EBV genes do? What is their function, does it make sense that these seem to drive NPC oncogenesis. I think these EBV genes are known to be associated with nasopharyngeal carcinomas.

We appreciate the clarification question regarding the four EBV variants that were detected at <5% in the UKB and AoU cohorts. In our revised manuscript, we have included additional context for each variant and the associated gene functions. Importantly, our analyses do not demonstrate causality in oncogenesis, and we instead highlight another dimension to our data where we can assess population-scale circulation of viral variation. Noting these limitations, we provide context for these genes:

BALF2 is an essential component of the lytic cycle, acting as a DNA-binding protein that facilitates EBV DNA replication during reactivation. Prior studies have implicated the two mentioned BALF2 variants, I613V and V317M, as high risk variants for nasopharyngeal cancer (NPC) that may contribute to more than 80% of the risk in regions with endemic NPC such as East Asia⁵².

RPMS is a nuclear protein that associates with a component of the Notch pathway to repress Notch activity; this was speculated to limit the ability of latently-infected memory B cells to respond to proliferative signals in order to maintain long-term latency⁵³. In particular, the mentioned variant, RPMS D51N, was found to lead to a longer half-life of the RPMS protein⁵⁴.

BNRF1 is a major EBV tegument protein that prevents the loading of repressive histones onto EBV genomes and promotes efficient viral replication by targeting cohesion complexes for degradation, which would otherwise restrict EBV replication compartments⁵⁵. BNRF1 P694H was also detected in NPC samples from endemic regions in East Asia⁵², but, to the best of our knowledge, a functional role for this variant has not been described.

We have summarized the prior functional literature in additional text in the **Methods** section (reproduced below):

[Page 39, Lines 1007-1017] *The four variants include BALF2 (I613V and V317M), a gene that is an essential component of the lytic cycle and acts as a DNA-binding protein that facilitates EBV DNA replication during reactivation. These two variants have been linked to high-risk NPC populations across multiple studies⁹⁹. RPMS is a nuclear protein that associates with a component of the Notch pathway to repress Notch activity¹²⁰, and the identified variant (D51N) is reported to result in a longer half-life of the protein¹²¹. Finally, BNRF1 is a tegument protein that prevents repressive histone loading onto EBV genomes and promotes efficient viral replication¹²². The P694H variant, to the best of our knowledge, has not been described with a functional role. We emphasize that our analyses do not necessarily validate the potential oncogenic role of these mutations. Instead, we highlight our approach to resolve population-level viral strain information that can be useful to mitigate geographical biases in interpreting variants of unknown significance in the EBV genome.*

17) Page 24: The threshold the authors used to define "EBV-positive" individuals is 1 pair of reads (300bp). This could potentially arise from sequencing run contamination and is borderline positive. How does the

EBV prevalence and all significant associations change with a slightly higher threshold, for example 2 or 3 pairs of EBV reads (4-6 reads)? The authors should include information on such an analysis, as if the positivity rate remains similar and all the associations still hold this will show the threshold the authors used is robust enough - however if findings change a lot, the authors should reflect on the reason.

We appreciate the emphasis on ensuring the robustness of our results. We have considered several lines of evidence for our association studies and have validated our choice of threshold. Conceptually, we emphasize that in the presence of contamination, the underlying effects would be obscured, having a conservative effect by decreasing statistical power for detecting associations. While we acknowledge that there is almost certainly measurement error and/or contamination in large population-based datasets, our results appear robust to them, and if anything, would reflect a conservative estimate of the true polygenic architecture of the DNAemia trait.

To explicitly test this expectation and evaluate higher thresholds, we repeated our ExWAS workflow for **five** additional thresholds in addition to the one used in the original and revised manuscript. A summary of the cutoffs and number of positive donors is noted here:

- 1.2 EBV genomes / 10,000 cells (~2 reads): 47,452 positive individuals (baseline / in manuscript)
- 2.0 EBV genomes / 10,000 cells (~3 reads): 22,692 positive individuals (-52%)
- 2.67 EBV genomes / 10,000 cells (~4 reads): 13,426 positive individuals (-72%)
- 4.67 EBV genomes / 10,000 cells (~8 reads): 5,866 positive individuals (-88%)
- 8.0 EBV genomes / 10,000 cells (~12 reads): 2,115 positive individuals (-96%)
- 20 EBV genomes / 10,000 cells (~30 reads); 283 positive individuals (-99.4%)

We note that due to differences in 30x WGS read coverage per individual, the integer value of the read does not necessarily map directly to the EBV genome threshold values, which were used in defining the set of individuals.

The results of our ExWASes at different thresholds are summarized in **Reviewer Figure 8**. For simplicity, we collapse the chromosome 6 genes to one bar and then consider the other loci with significant protein-altering genes. In brief, we observed that no genes were discovered at higher thresholds, and genetic signals were only lost as the threshold increased, ultimately resulting in 0 non-MHC genes being associated at the 8 EBV genomes threshold and 0 variants anywhere (MHC or otherwise) that were genome-wide significant at the 20 EBV genomes threshold. We attribute the drop in associations at higher thresholds to a result of changing impacting statistical power. Consistent with this, none of the genetic signals we detected were preferentially enriched among the higher thresholds. Related to this point, by drawing a threshold at higher levels, individuals who were previously “positive” become “negative,” likely increasing the error rate and further limiting statistical power.

Reviewer Figure 8 (reproduced from above). Summary of ExWAS results at non-chromosome 6 loci at different EBV DNA thresholds. The top plot shows the number of DNA+ individuals at variable thresholds. The bottom plot shows whether the protein coding gene had a significant ($P < 5 \times 10^{-8}$) association for one or more protein-altering variants associated with the binarized EBV individuals. 1.2 was selected for main analyses in the manuscript.

Ultimately, we suggest that even if a proportion of our EBV DNAemia positive donors reflect contamination rather than true infection, we would expect these individuals to be ~ random with respect to genetic variation. Hence, it is unlikely that genes shown in our GWAS/ExWAS are false positives from a systematic error, and if anything, our results would represent a **conservative** set of genes where more discovery could be enabled with ideal data. As an additional point, we note that a similar preprint⁵⁶, published after we received these reviewer comments, has characterized genetic associations with EBV DNAemia and arrive at similar conclusions, in which using a lower read threshold yields greater statistical power but highly concordant results (i.e., very similar β coefficients) with higher (5+ read) thresholds.

In sum, our additional analyses support our chosen threshold. While increasing the threshold does increase specificity, it results in a loss in statistical power. In our case, as we seem to have a low contamination rate, increasing the threshold leads to a loss of EBV+ individuals and resulting signal. Similar results are observed for our serology analyses (discussed in Reviewer 5, Points 19 and 20 below, with more intuition provided).

18) Relevant to the above point, do the authors find any EBV positive individuals by sequencing, who are at the same time EBV seronegative? This would be a good "sanity check" for the validity of their approach and interpretation of their findings.

We appreciate this excellent suggestion for a sanity check. For the 9,607 individuals with both EBV serology and EBV DNAemia status, there were 919 individuals that were EBV DNA+. Of these 919 DNA+ individuals, only 2 individuals (0.2%) were seronegative using the UKB definition of seropositivity. However, under a closer scrutiny of the data, 100% of EBV DNA+ individuals had evidence of EBV serology responses.

Specifically, one donor had an EBV DNA load of 1.36 EBV genomes / 10⁴ cells (just above our EBV DNA+ cutoff), with a high VCAp18 titer but low titers for the other three EBV antigens. The other donor had an EBV DNA load of 3.34 EBV genomes / 10⁴ cells, with a positive titer for EA-D but low titers for the other antigens. Hence, both donors had antibody response against at least one EBV antigen, suggesting that these donors may still have experienced past EBV infection, despite not meeting the UKB EBV seropositivity criteria.

In summary, this analysis provides a strong corroboration of our analysis, suggesting a limited false-positive rate of EBV DNAemia classification. We are grateful to the reviewer for the excellent suggestion and have added these details to a new **Extended Data Fig. 1B** (reproduced below as **Reviewer Fig. 23**), where we stratify donors by the number of seropositive EBV antigens, showcasing the 0% EBV DNA+ for individuals with 0 seropositive EBV antigens.

Reviewer Figure 23 (manuscript Extended Figure 1B). Characterization of EBV DNAemia rates by EBV serostatus levels.

19) Figures 1e and 1g are not intuitive. Why in 1g, the odds ratio for EBV genome abundance and EBV seropositivity is lower for greater genome abundance?

For 1e, this is meant to summarize the data, including the zero-inflated nature of the EBV detection. As part of the more extensive simulation analyses of EBV DNA, we provide alternative visualizations of the empirical distribution which is shown in **Extended Data Fig. 1f** (or **Reviewer Fig. 5f**, reproduced below). The plot is not intended to support a quantitative claim, but to illustrate the distribution of the data.

Reviewer Figure 5f (reproduced from above). Simulations of EBV DNAemia under a censored log-normal distribution. (f) Empirical distribution of observed EBV levels in the UKB.

For 1g, the individuals below the threshold are assigned “negative” even with detectable EBV DNA. Therefore, with higher EBV DNA thresholds, the number of EBV DNA- individuals increase, but these individuals are still seropositive, which diminishes the magnitude of the enrichment. Conceptually, the increasing threshold beyond a certain point “dilutes” the signal, since the proportion of (EBV DNA-, serology+) individuals increases relative to (EBV DNA+, serology+) individuals, while the two other combinations remain static. The result is a decrease in the magnitude of the odds ratio and the statistical significance.

20) Line 102: “A cutoff of 1.2 viral genomes per 10^4 human cells yielded the strongest concordance with seropositivity (OR = 82.2, P = 2.2×10^{-16} ; Fig. 1g).” - this does not make a lot of sense, why do 2 EBV genome per 10K human cells have lower concordance with seropositivity? Also does seropositivity have a quantitative unit and is not binary? Please provide more information.

We hope to clarify this result. For the first part of the question, we refer to Reviewer 5 / Point 19 above. At increasing thresholds, since the proportion of (EBV DNA-, serology+) individuals increases relative to (EBV DNA+, serology+) individuals, while the two other combinations remain static, this results in a less significant test statistic.

For the second part of the question, there are indeed quantitative antibody measurements (MFI) against 4 EBV antigens (VCAp18, EBNA1, ZEBRA, and EA-D) in UKB, with seropositivity thresholds defined using a reference panel⁵⁷. Explicitly, EBV seropositivity in UKB is defined as having two positive antibody serologies of four values [in units of MFI], with thresholds for each antigen at: VCA p18 > 250; EBNA-1 > 150; ZEBRA > 100; EA-D > 100.

In other words, the annotation of serostatus per person is provided by UKB and is static across all of our analyses. Our analyses in **Fig. 1** seek to define a threshold that is interpretable, as no pre-existing standard exists for quantifying EBV DNA positivity from WGS data.

21) Line 481: What EBV antibody is used in the UK biobank? Is it the one that shows prior infection (IgG)? Are there further antibodies included such as EA-IgG which indicates EBV reactivation - please provide more information.

We appreciate the reviewer’s excellent attention to detail and in the revised version of the manuscript, we now clarify these important points.

Notably, the isotype information of the antibody titers measured is not provided. Hence, inferring latent versus lytic status of EBV for each individual is not feasible using serology definitions of lytic and latent infection. For instance, the study on which the panel is based⁵⁷ reported that IgM, IgA, and IgG measurements on various antigens could be used to distinguish past versus reactivated EBV infection. However, these isotype values are not reported as part of the serology data in UKB. Further, the study suggests that presence of anti-ZEBRA or EA-D antibodies may indicate recent lytic reactivation. However, the high percentages of seropositivity and high false positives against ZEBRA and EA-D may indicate that the assay was overly sensitive, as the study also noted.

To address these points, we conducted additional analyses comparing seropositivity against each antigen separately. In line with the paper, we also observed high percentages of seropositivity against ZEBRA and EA-D in the serology dataset (86% and 91%, respectively). In comparison, a study specifically measuring

lytic reactivation titers using isotype-specific antibodies suggested 25% of the population at any given time undergoing lytic reactivation³⁵.

We then investigated whether antibody titers or seropositivity changed over time given EBV DNAemia status. Using the 262 individuals with a repeated serology measurement and baseline WGS, we compared titers and seropositivity across the two timepoints (at least 2 years apart; specific dates not available). Of the 17 EBV DNA+ individuals, none showed seroconversion or reversion against any of the four EBV antigens. There were also no significant differences in antibody titer changes for EBV DNA+/- (Reviewer Fig. 7i).

Reviewer Figure 7 (reproduced from above). Longitudinal serology analyses from UKB. (i) Characterization of EBV longitudinal antigen titers across UKB. Statistical test: two-sided Student's *t*-test.

Overall, the limited serology information limits our interpretation of reactivation from this data source alone. Hence, we have supplemented our analyses with substantial additional transcriptomic data from OneK1K and GTEx to further characterize rates of EBV reactivation in peripheral blood from a general population sampling like UKB, which we find minimal evidence.

22) Line 278 “Further, we observed that predicted immuno-dominant peptides were depleted in latency-associated EBV genes specifically for MHC class I peptides, reflecting potential viral evolution to evade host immunity during latency – please provide a reference for this statement.

We have added a reference supporting the idea that reduced presentation of viral protein fragments may serve as a strategy for the virus to evade host immune detection⁵⁸. This work reviews this overall phenomenon and provides a specific example of EBNA-1 downregulating translation to decrease MHC I presentation and thereby regulate latency.

Minor

1) Line 171, “Namely” is not followed by a sentence.

We have addressed this accordingly.

2) Figure 1 d) clarify how the 18 infectious agents were chosen (only this information was available in UK Biobank or something else?)

At the time of our first submission, only 18 infectious agents were available in our research analysis platform instance, which has since been updated. In our revised version, we tested *all* the infectious agents that were assessed with serology in UKB. Our revised Fig. 1D (reported as Reviewer Fig. 24 below) shows these results, again with EBV as the clearly most associated factor.

Reviewer Figure 24 (manuscript Fig. 1D). Statistical tests of EBV DNAemia with serology from 22 infectious agents. Serostatus for each infectious agent was tested for association with the EBV DNAemia annotation. Note that HHV6 is further separated into HHV6A and HHV6B. Statistical test: Fisher test for association.

These agents were selected by UKB because they are either established risk factors for disease outcomes or are of scientific interest, and the specific antigens were included based on known biological functions and/or existing assays. Our comprehensive testing and specificity to EBV provides a strong measure of specificity to our measurement.

3) Figure 5 line386/ There is an extra dot at the end of the sentence for point e).

We have addressed this accordingly.

Referee #5 (Remarks on code availability):

I reviewed the github repository but did not attempt to install and run the code. The code seems well organised, with different directories for the different sets of analyses, with documentation and a top-level README file. By design, the individual-level data cannot be shared publicly in the github repo so not all analyses can be replicated, however this is understandable and expected due to data sharing agreements. The authors provide contact details for researchers who wish to perform similar analyses.

We appreciate the time and effort spent reviewing the code. In our revised manuscript, we have expanded the details in the methods section. Further, we now provide a vignette using publicly available data from the publicly available 1000 Genomes Project. This enabled us to share the details behind how to access and process individual-level data for downstream analyses, ultimately leading into our workflow for the retrospective quantification of viral DNA from WGS data.

Referee #6 (Remarks to the Author):

I co-reviewed this manuscript with one of the reviewers who provided the listed reports.

We appreciate the additional time spent by Referees 5 and 6 in evaluating our code and manuscript. As we show for Referee #3, we have revised our code resources and enhanced the documentation related to our analyses. In particular, we have added another vignette analyzing the publicly available 1000G Project that provides a more direct means for other users to reproduce our workflows using publically available data. We anticipate that these additional measures will enable future users to similarly reanalyze large-scale WGS to study latent viral infections.

Supplemental Note

Soon after our bioRxiv preprint was posted online⁵⁹, a separate preprint was posted on medRxiv with striking similarities to quantify EBV DNAemia in UKB and AoU and reported many similar conclusions⁵⁶. Namely, the authors similarly show how bias-corrected quantification of EBV DNA from existing WGS enables a powerful novel biomarker for genetic and phenotypic association studies. Their work validates that the framework and results of our paper are rigorous across research groups. Notably, our paper includes several distinct components that were not captured in the other group's preprint, including novel analyses suggested as part of our *Nature* review process. Additionally, our comprehensive study of viral gene expression analyses from complementary cohorts (**Extended Data Fig 3**), peptide presentation correlations (**Fig. 5; Extended Data Fig. 8**), and viral genetic variation (**Extended Data Fig. 9**) are not captured by the other preprint. Our PheWAS is also broader in scope, spanning 3,289 binary and 1,931 quantitative phenotypes, compared to the 1,751 binary phenotypes studied in the Schmidt *et al.* preprint. Finally, following the advice of our helpful reviewer comments, our revised article expands the scope of our GWAS to include meta-analysis from non-European genetic ancestry individuals, which results in additional genome-wide significant loci missed when focusing solely on the European cohort (**Reviewer Fig. 25**). The Schmidt *et al.* preprint focuses on the European genetic ancestry of individuals. Below, we compare test statistics for overlapping phenotypes, which show high concordance between our results and their independent analyses. We include a citation to this July preprint in our revised manuscript.

Comparison of Schmidt *et al.* associated loci. Both manuscripts focus analyses on the characterization of genetic loci that are associated with EBV DNA. Using the effect size statistics that are accessible in the preprint, we compared the 27 independent loci reported by Schmidt *et al.* with the same variants in our results (**Reviewer Fig. 25**). The association statistics were near-perfectly correlated for the accessible statistics, showing the independent reproducibility of the statistical analyses reported in our work.

Reviewer Figure 25. Comparison of 27 genome-wide significant loci reported in Schmidt *et al.* (Supp. Table S4).

Response to reviewer references

1. Giunco, S. *et al.* Cross talk between EBV and telomerase: the role of TERT and NOTCH2 in the switch of latent/lytic cycle of the virus. *Cell Death Dis.* **6**, e1774 (2015).
2. Pender, M. P. & Burrows, S. R. Epstein-Barr virus and multiple sclerosis: potential opportunities for immunotherapy. *Clin. Transl. Immunology* **3**, e27 (2014).
3. Levine, K. S. *et al.* Virus exposure and neurodegenerative disease risk across national biobanks. *Neuron* **111**, 1086 (2023).
4. Hammer, C. *et al.* Amino acid variation in HLA class II proteins is a major determinant of humoral response to common viruses. *Am. J. Hum. Genet.* **97**, 738–743 (2015).
5. Scepanovic, P. *et al.* Human genetic variants and age are the strongest predictors of humoral immune responses to common pathogens and vaccines. *Genome Med.* **10**, 59 (2018).
6. Gruffat, H., Marchione, R. & Manet, E. Herpesvirus late gene expression: A viral-specific pre-initiation complex is key. *Front. Microbiol.* **7**, 869 (2016).
7. Lareau, C. A. *et al.* Latent human herpesvirus 6 is reactivated in CAR T cells. *Nature* **623**, 608–615 (2023).
8. Mentzer, A. J. *et al.* Identification of host–pathogen–disease relationships using a scalable multiplex serology platform in UK Biobank. *Nature Communications* **13**, 1–12 (2022).
9. Conacher, M. *et al.* Epstein-Barr virus can establish infection in the absence of a classical memory B-cell population. *J. Virol.* **79**, 11128–11134 (2005).
10. Burren, O. S. *et al.* Genetic feature engineering enables characterisation of shared risk factors in immune-mediated diseases. *Genome Med.* **12**, 106 (2020).
11. Balandraud, N. & Roudier, J. Epstein-Barr virus and rheumatoid arthritis. *Joint Bone Spine* **85**, 165–170 (2018).
12. Draborg, A. H., Duus, K. & Houen, G. Epstein-Barr virus and systemic lupus erythematosus. *Clin. Dev. Immunol.* **2012**, 370516 (2012).
13. Klatka, M. *et al.* Effect of Epstein-Barr virus infection on selected immunological parameters in children with type 1 diabetes. *Int. J. Mol. Sci.* **24**, 2392 (2023).

14. Vita, R. *et al.* The Immune Epitope Database (IEDB): 2024 update. *Nucleic Acids Res.* **53**, D436–D443 (2025).
15. Damania, B., Kenney, S. C. & Raab-Traub, N. Epstein-Barr virus: Biology and clinical disease. *Cell* **185**, 3652–3670 (2022).
16. Dowd, J. B., Palermo, T., Brite, J., McDade, T. W. & Aiello, A. Seroprevalence of Epstein-Barr virus infection in U.S. children ages 6-19, 2003-2010. *PLoS One* **8**, e64921 (2013).
17. Winter, J. R. *et al.* Predictors of Epstein-Barr virus serostatus in young people in England. *BMC Infect. Dis.* **19**, 1007 (2019).
18. Finucane, H. K. *et al.* Partitioning heritability by functional annotation using genome-wide association summary statistics. *Nat. Genet.* **47**, 1228–1235 (2015).
19. Schep, A. N., Wu, B., Buenrostro, J. D. & Greenleaf, W. J. chromVAR: inferring transcription-factor-associated accessibility from single-cell epigenomic data. *Nat. Methods* **14**, 975–978 (2017).
20. Avsec, Ž. *et al.* AlphaGenome: advancing regulatory variant effect prediction with a unified DNA sequence model. *Bioinformatics* (2025).
21. Dilthey, A. *et al.* Multi-population classical HLA type imputation. *PLoS Comput. Biol.* **9**, e1002877 (2013).
22. Song, L., Bai, G., Liu, X. S., Li, B. & Li, H. Efficient and accurate KIR and HLA genotyping with massively parallel sequencing data. *Genome Res.* **33**, 923–931 (2023).
23. Kulski, J. K., Suzuki, S. & Shiina, T. Human leukocyte antigen super-locus: nexus of genomic supergenes, SNPs, indels, transcripts, and haplotypes. *Hum. Genome Var.* **9**, 49 (2022).
24. Fogdell-Hahn, A., Ligiers, A., Grønning, M., Hillert, J. & Olerup, O. Multiple sclerosis: a modifying influence of HLA class I genes in an HLA class II associated autoimmune disease: MHC class I associations in multiple sclerosis. *Tissue Antigens* **55**, 140–148 (2000).
25. Wu, J.-S. *et al.* HLA-DRB1 allele heterogeneity influences multiple sclerosis severity as well as risk in Western Australia. *J. Neuroimmunol.* **219**, 109–113 (2010).
26. Miles, J. J. *et al.* CTL recognition of a bulged viral peptide involves biased TCR selection. *J. Immunol.* **175**, 3826–3834 (2005).
27. Brooks, J. M. *et al.* Early T cell recognition of B cells following Epstein-Barr virus infection: Identifying

- potential targets for prophylactic vaccination. *PLoS Pathog.* **12**, e1005549 (2016).
28. Nudel, R. *et al.* A large-scale genomic investigation of susceptibility to infection and its association with mental disorders in the Danish population. *Transl. Psychiatry* **9**, 283 (2019).
 29. COVID-19 Host Genetics Initiative. A second update on mapping the human genetic architecture of COVID-19. *Nature* **621**, E7–E26 (2023).
 30. Wong, Y., Meehan, M. T., Burrows, S. R., Doolan, D. L. & Miles, J. J. Estimating the global burden of Epstein-Barr virus-related cancers. *J. Cancer Res. Clin. Oncol.* **148**, 31–46 (2022).
 31. Moustafa, A. *et al.* The blood DNA virome in 8,000 humans. *PLoS Pathog.* **13**, e1006292 (2017).
 32. Babcock, G. J., Hochberg, D. & Thorley-Lawson, A. D. The expression pattern of Epstein-Barr virus latent genes in vivo is dependent upon the differentiation stage of the infected B cell. *Immunity* **13**, 497–506 (2000).
 33. Babcock, G. J., Decker, L. L., Freeman, R. B. & Thorley-Lawson, D. A. Epstein-barr virus-infected resting memory B cells, not proliferating lymphoblasts, accumulate in the peripheral blood of immunosuppressed patients. *J. Exp. Med.* **190**, 567–576 (1999).
 34. Latour, S. Human immune responses to Epstein-Barr virus highlighted by immunodeficiencies. *Annu. Rev. Immunol.* **43**, 723–749 (2025).
 35. Schneiderova, P. *et al.* The SARS-CoV-2 trigger highlights host interleukin 1 genetics in Epstein-Barr virus reactivation. *Cell Rep.* **44**, 115859 (2025).
 36. Yazar, S. *et al.* Single-cell eQTL mapping identifies cell type-specific genetic control of autoimmune disease. *Science* **376**, eabf3041 (2022).
 37. Sausen, D. G., Bhutta, M. S., Gallo, E. S., Dahari, H. & Borenstein, R. Stress-induced Epstein-Barr virus reactivation. *Biomolecules* **11**, 1380 (2021).
 38. Majerciak, V., Yang, W., Zheng, J., Zhu, J. & Zheng, Z.-M. A genome-wide Epstein-Barr virus polyadenylation map and its antisense RNA to EBNA. *J. Virol.* **93**, (2019).
 39. McManus, T. E. *et al.* High levels of Epstein-Barr virus in COPD. *Eur. Respir. J.* **31**, 1221–1226 (2008).
 40. Spargo, T. P. *et al.* Haploinsufficiency of ITSN1 is associated with a substantial increased risk of Parkinson's disease. *Cell Rep.* **44**, 115355 (2025).
 41. Nag, A. *et al.* Human genetics uncovers MAP3K15 as an obesity-independent therapeutic target for

- diabetes. *Sci. Adv.* **8**, eadd5430 (2022).
42. Georgakis, M. K., Bernhagen, J., Heitman, L. H., Weber, C. & Dichgans, M. Targeting the CCL2-CCR2 axis for atheroprotection. *Eur. Heart J.* **43**, 1799–1808 (2022).
 43. Schlosser, P. *et al.* Genetic studies of paired metabolomes reveal enzymatic and transport processes at the interface of plasma and urine. *Nat. Genet.* **55**, 995–1008 (2023).
 44. Traylen, C. M. *et al.* Virus reactivation: a panoramic view in human infections. *Future Virol.* **6**, 451–463 (2011).
 45. Wheeler, H. E. & Eileen Dolan, M. Lymphoblastoid cell lines in pharmacogenomic discovery and clinical translation. *Pharmacogenomics* **13**, 55 (2012).
 46. International HapMap Consortium. The International HapMap Project. *Nature* **426**, 789–796 (2003).
 47. Mandage, R. *et al.* Genetic factors affecting EBV copy number in lymphoblastoid cell lines derived from the 1000 Genome Project samples. *PLoS One* **12**, e0179446 (2017).
 48. Lappalainen, T. *et al.* Transcriptome and genome sequencing uncovers functional variation in humans. *Nature* **501**, 506–511 (2013).
 49. Wang, Q. *et al.* Rare variant contribution to human disease in 281,104 UK Biobank exomes. *Nature* **597**, 527–532 (2021).
 50. Brown, K. L., Ramlall, V., Zietz, M., Gisladdottir, U. & Tatonetti, N. P. Estimating the heritability of SARS-CoV-2 susceptibility and COVID-19 severity. *Nat. Commun.* **15**, 367 (2024).
 51. Gusev, A. *et al.* Partitioning heritability of regulatory and cell-type-specific variants across 11 common diseases. *Am. J. Hum. Genet.* **95**, 535–552 (2014).
 52. Briercheck, E. L. *et al.* Geographic EBV variants confound disease-specific variant interpretation and predict variable immune therapy responses. *Blood Adv.* **8**, 3731–3744 (2024).
 53. Zhang, J., Chen, H., Weinmaster, G. & Hayward, S. D. Epstein-Barr virus BamHi-a rightward transcript-encoded RPMS protein interacts with the CBF1-associated corepressor CIR to negatively regulate the activity of EBNA2 and Notch1C. *J. Virol.* **75**, 2946–2956 (2001).
 54. Feng, F.-T. *et al.* A single nucleotide polymorphism in the Epstein-Barr virus genome is strongly associated with a high risk of nasopharyngeal carcinoma. *Chin. J. Cancer* **34**, 563–572 (2015).
 55. Sagou, K. *et al.* Epstein-Barr virus lytic gene BNRF1 promotes B-cell lymphomagenesis via IFI27

- upregulation. *PLoS Pathog.* **20**, e1011954 (2024).
56. Schmidt, A. *et al.* Host control of latent Epstein-Barr virus infection. *Infectious Diseases (except HIV/AIDS)* (2025).
57. Brenner, N. *et al.* Validation of Multiplex Serology detecting human herpesviruses 1-5. *PLoS One* **13**, e0209379 (2018).
58. Sausen, D. G., Poirier, M. C., Spiers, L. M. & Smith, E. N. Mechanisms of T cell evasion by Epstein-Barr virus and implications for tumor survival. *Front. Immunol.* **14**, 1289313 (2023).
59. Nyeo, S. S. *et al.* Population-scale sequencing resolves correlates and determinants of latent Epstein-Barr Virus infection. *bioRxiv.org* (2025) doi:10.1101/2025.07.18.665549.

Dear _____,

Thank you once more for the prompt handling of our manuscript and the constructive feedback from the six referees. We were encouraged that our revisions have further elevated our manuscript.

The notable changes are as follows:

1. In response to the reviewer comments, we have ensured consistency of terminology throughout our manuscript, including the following:
 - a. We have now limited the use of the term “latent” in reference to our data only in a specific section (*Multi-modal characterization of EBV in peripheral blood*) to avoid strong claims about the nature of the viral infection. Further, we have revised the title of our manuscript, which no longer has the word “latent” based on this feedback.
 - b. We have now removed the “EBV DNA+” label from individuals for parsimony with the “EBV DNAemia” term that we now use throughout the manuscript.
2. We introduce and describe new analyses of WGS performed from 48,899 donors in the All of Us cohort that provided saliva instead of blood, which are summarized in a new **Supplemental Note 1**. Related to the feedback on other reservoirs of the virus, this additional analysis provides a new dimension to the manuscript, showing a separate reservoir but one with limited host genetic control, reinforcing the findings in our blood-derived associations.
3. We have added a comprehensive assessment of coding variation associated with EBV DNAemia at the suggestion of Referee #3, including highlighting a specific non-HLA, chromosome 6 association to refine the strong associations at this chromosome, which has been added to **Extended Data Fig. 5**.
4. We provide additional discussion to contextualize that our measurement is a snapshot of EBV DNA, which is likely to change over time and due to environmental factors. We emphasize here and in the text that our associations are a *lower bound*, as more thorough, longitudinal measurements would facilitate a clearer exposure and, in turn, greater sensitivity for associations.

Overall, our revised version now contains **5 main text figures, 9 Extended Data Figures, and 1 Supplemental Note**. The changes to the main manuscript text, as well as our responses to the reviewer comments, are noted in **red** with the changes from our first revision retained in **blue**. We anticipate that our revised manuscript now satisfies the remaining reviewer comments for publication in *Nature*.

Sincerely,
Slavé, Ryan, and Caleb
on behalf of all co-authors

Referees' comments:

Referee #1 (Remarks to the Author):

I appreciate the authors careful consideration of my comments. I have no further comments and support publication of the revised manuscript.

We appreciate the re-review of our manuscript and the supportive feedback.

Referee #2 (Remarks to the Author):

The revised manuscript has added significant new analyses and remains an important study. However, the revision may have compounded some of the concerning issues by making stronger, yet not fully substantiated claims.

We appreciate the continued support of our manuscript. We do not intend to overstate the implications of our claims, and the feedback on complicated terminology or specific verbiage has been helpful to ensure the appropriateness of our analyses and conclusions.

Major concerns include:

1. While the correlations of the many diseases with EBVemia is curious, the notion that EBV is potentially causative for 242 significant disease indications seems very unlikely and inconsistent with most other epidemiological studies. The authors should think of other, non-mechanistic reasons that this correlation exists, and may reflect a general state of immune insufficiency.

We agree that the phenotypic associations do not establish causality, and we have made sure to more explicitly state this in the revision. Additionally, the large number of associations requires greater contextualization. The binary phenotypes were derived from self-reported data as well as ICD-10 codes. Given the tree-like, parent-child hierarchy of ICD-10 billing codes, there is redundancy between many phenotypes. For example, rheumatoid arthritis and related sub-phenotypes are captured with 9 annotations (**Extended Data Table 3**), including:

- Union#M05#Seropositive rheumatoid arthritis
- Union#M0690#Rheumatoid arthritis unspecified Multiple sites
- Union#M0694#Rheumatoid arthritis unspecified Hand

We have now modified the main text to directly point this out:

[Page 7, Lines 201-203] We emphasize that these associations do not prove causality of EBV DNAemia and may instead reflect a general state of immunosuppression, and the large number of associated traits reflects redundancy in ICD-10 mapping.

We have taken extra precautions to remove all causal language from the PheWAS section, and further add perspective to our findings with additional text:

[Page 8, Lines 236-238] While further work is required to refine the role of EBV in these phenotypes, our scalable approach enables systematic associations with a large set of conditions, including rare diseases that require large sample sizes, as in UKB and AoU.

2. The authors appear to have made even stronger claims that the DNAemia reflects “latent” infection only. For EBV, the terminology of latency is complicated, as latent reservoirs likely reflect on-going lytic activity and repopulation of latently infected B-cells. It is not yet known, and not shown by any of the data, whether increased EBV DNAemia may be due to increase lytic activity in some reservoir (e.g. oropharynx). This could reflect important differences in underlying biological mechanism for any disease correlation. For this reason, I think the manuscript is better served with a more conservative and cautious use of the term “latent”.

To encompass the multitude of outcomes following EBV primary infection, we have elected to use “persistence” to describe the detected DNA (note: we claim persistent DNA rather than persistent infection) in line with prior definitions^{1,2}, as we believe the term most adequately captures the complexity in harboring this virus³.

We have changed the paper title and revised the manuscript to better describe this nuance:

- Our title now reads **Population-scale sequencing resolves correlates and determinants of persistent Epstein-Barr Virus DNA**.
- Our abstract now introduces the molecular feature we are characterizing as persistent EBV DNA, and concludes that existing WGS data can derive the genetic architecture of viral DNA persistence.

- We echo this in the introduction:

[Page 3, Lines 69-70] *As modern biobanks perform WGS on peripheral blood rather than LCLs, we posited that EBV DNA reflecting persistence in circulating cells could be captured and quantified in these libraries.*

- We describe individuals with EBV DNAemia more cautiously in the section **Biobank WGS data harbors EBV DNA**:

[Page 5, Lines 117-121] *As the proportion of individuals with EBV DNAemia (9.7%) is lower than the rate of seropositivity (~95%) in UKB, we interpret our measure to reflect the tail of individuals with the highest levels of circulating EBV DNA. This approach does not detect every individual who has had previous EBV infection, but identifies those with the greatest EBV DNA burden detectable in peripheral blood at the time of WGS sampling.*

- We now constrain the discussion of latency with respect to our data analysis to a single section: **Multi-modal characterization of EBV in peripheral blood**. We deliberately chose this language to be inclusive of alternative interpretations, including spontaneous infection and reactivation.
- We revised the interpretation of EBV DNAemia in the section **Polygenic variation underlies EBV DNAemia**:

[Page 10, Lines 331-334] *Namely, serostatus reflects a history of infection, which is largely independent of genetic variation, whereas our EBV DNAemia biomarker identifies a subset of individuals with high levels of persistent EBV DNA at the time of sampling, including those in the absence of lytic reactivation.*

- We include the complexity of EBV infection and changes in detectable DNA levels in the discussion:

[Page 14, Lines 494-498] *Further, as viral DNA levels can fluctuate longitudinally, these WGS measurements represent a snapshot of a complex process that marked individuals with potentially transiently high levels of EBV DNA. Thus, longitudinal profiles would refine our DNAemia phenotyping and power further discovery of genetic loci that underlie population variation in viral latency and persistence.*

- We have removed the word “latent” from figures and captions, including in **Extended Data Fig. 1**.

We also agree about the importance of other reservoirs (e.g., oropharynx) to understand how our approach can be used to study population variation. We note **Reviewer #2, Point 4** below, where we provide new analyses that help contextualize the primary findings from blood-derived analyses by contrasting with saliva EBV DNA detection (new **Supplementary Note 1**, and see response to the Reviewer’s comment #4).

3. Are individuals that have high DNAemia consistently high over time in longitudinal studies. While some longitudinal data was provided for serology (reviewer figure 7), it was not clear whether a similar longitudinal study was available for EBV DNAemia. This should be clarified as individuals with high EBV DNA loads may reflect recent reactivations in some anatomical locations that resolve over time, explaining why RNAseq rarely detects a tissue-resident reactivations. Similarly, the state of EBV serology may also not reflect a chronic reactivation status.

We emphasize that the longitudinal data added in the past revision (**Reviewer Fig. 7; reproduced as Reviewer Fig. 1 below**) were strictly longitudinal serology measurements. Unfortunately, longitudinal WGS was not available in UKB, GTEEx, AoU, or any other biobank that we have been able to ascertain, but we anticipate that our study, as well as recent work evaluating somatic evolution in peripheral blood^{4,5}, will motivate longitudinal measurements, which we emphasize in our revised **Discussion**.

Reviewer Figure 1 v2 (previous Reviewer Fig. 7; manuscript Extended Data Fig. 3j,k). Longitudinal serology analyses from UKB. (j) Depiction of longitudinal serology measurements of four EBV antigens for an exemplar EBV DNAemia donor. **(k)** Characterization of EBV longitudinal antigen titers across UKB. Statistical test: two-sided Student’s *t*-test.

We also agree that the longitudinal dynamics of EBV persistence are a useful heuristic to contextualize our data, and that it is challenging to determine the viral state from the available EBV serology measurements in UKB. We have revised our text in several places to reflect this nuance:

- In the section, **Multi-modal characterization of EBV in peripheral blood:**
[Page 6, Lines 176-182] *We analyzed the change in titers against all four EBV antigens, finding no significant difference in longitudinal fold changes of antibody titers when stratified by EBV DNAemia (Extended Data Fig. 3j,k). Additionally, we observed that 0 individuals with EBV DNAemia had seroconversion or seroreversion over longitudinal samples for any of the four antigens. The lack of resolution in antibody isotypes means that seropositivity does not necessarily indicate ongoing or recent reactivation (Methods), and we observed no evidence of variable antibody responses that would result from recent reactivation or infection in this population.*
- In the **Discussion:**

[Page 14, Lines 494-498] Further, as viral DNA levels can fluctuate longitudinally, these WGS measurements represent a snapshot of a complex process that marked individuals with potentially transiently high levels of EBV DNA. Thus, longitudinal profiles would refine our DNAemia phenotyping and power further discovery of genetic loci that underlie population variation in viral latency and persistence.

- In the **Methods** (*Corroboration of EBV latency*):

[Page 34, Lines 807-819] Of the 277 individuals with longitudinal serology, 262 had: a) WGS profiled from the baseline sample (“instance 0” in UKB data field 32056); b) serology assessed at the baseline sample; and c) a replicate serology measure from a separate blood draw taken 2–6 years following the baseline (note: the precise dates were not available). There were at most two time points available for serology measurements, and no longitudinal WGS samples were available in either biobank... We note that while serology can be used to distinguish between active, latent, and recently reactivated EBV states, the UKB serology measurements do not provide the detailed isotype information needed to fully distinguish between these states in the individuals measured.

4. The revised Introduction is more problematic since it more emphatically concludes that the EBV state is “Latent” and also that the reservoir is in the PBMC, which may not be correct, as there is reservoir in lymphoid tissue in oropharynx and elsewhere.

Related to our response to the Reviewer’s comment #2 above, we have struck “latent” from the title, the abstract, and key parts of the main text (including the Introduction) in an effort to be inclusive of this possibility. We believe our revised text now appropriately handles the nuance in these terms.

We also agree about the importance of studying reservoirs aside from PBMCs to help in understanding the results of our characterization in blood. To these ends, we have provided a new set of analyses that helps clarify the distinct nature of the blood-derived EBV that is the focus of this study. Namely, in the recent release of the All of Us (AoU) version 8 dataset, WGS was derived from 48,889 saliva samples in AoU, enabling analogous analyses to discriminate the biology of EBV DNA persistence in a reservoir aside from blood. We include a new **Supplemental Note** that now compares and contrasts saliva-derived EBV from blood-derived EBV to provide a new context that bolsters our original findings. Though zero individuals were assayed by both blood and saliva WGS, we observe starkly different patterns from considering these two reservoirs. Namely, despite using identical thresholds, we observed a marked difference in EBV DNAemia between the input sources. Specifically, ~51% of donors were positive for EBV DNAemia from saliva, as opposed to ~10% from our two cohorts of peripheral blood. While the male increase in EBV persistent DNA is similar to peripheral blood, we did not observe an age association, and overall, we observed considerably higher levels of detected EBV DNA among those with any detectable viral DNA. **Reviewer Figure 2** summarizes the major findings from this additional analysis:

Reviewer Figure 2. (Supplemental Note Figure 1). Quantification of EBV DNA from AoU saliva WGS samples. (a) Schematic of the approach. **(b)** Sum of per-base read coverage of high-confidence EBV-mapping reads. **(c)** Partition of AoU saliva WGS participants by EBV DNA detection after accounting for biased regions. “Biased only” refers to participants with reads mapping to only the two repetitive regions indicated in the main text **Fig. 1b**. “Valid and low count” refers to participants with EBV DNA detected after masking the two biased regions. “DNAemia” refers to participants who pass the threshold of 1.2 EBV copies per 10^4 human cells. **(d)** Empirical cumulative distribution of detected EBV DNA across all saliva samples, compared to blood. **(e)** Percent saliva EBV DNAemia resolved by sex and age in AoU. Statistical test: two-sided proportion test comparing sex in the associated age bin. Error bars: standard error of the mean. **(f)** Manhattan plot summarizing the genome-wide association statistics for saliva EBV DNAemia for 32,745 individuals of EUR ancestry. The only genome-wide significant association region in the HLA locus ($P < 5 \times 10^{-8}$) is annotated. **(g)** Number of genome-wide significant variants with each GWAS model, in chromosome six or not. “0018” refers to the threshold of 1.2 EBV copies per 10^4 human cells. “Top 10%” refers to a threshold of $\sim 1,800$ copies per 10^4 cells. “Continuous” refers to providing the EBV DNA load as a quantitative trait. **(h)** Results of allele-level regression, showing the z-score of the Wald test for individual HLA class I alleles, subsetted to $P < 0.1$ in AoU blood regression results. **(i)** Same as (h) but for class II alleles.

While the full text outlines our methods, results, and interpretations, we highlight a few conclusions particularly pertinent to the concerns raised:

- 1) Our method of identifying EBV DNA reads from existing WGS libraries will scale to other input materials and allow for future study and comparisons to other biological reservoirs.
- 2) EBV DNA detection is qualitatively and quantitatively different in peripheral blood than saliva, as we see substantially more positive donors when assaying saliva (**Reviewer Fig. 2d**). This likely reflects the distinct biological dynamics of the infection, reactivation, and shedding of EBV in the oropharynx compared to peripheral blood.
- 3) When performing genetic association mapping in three different ways for saliva, we only observed one genome-wide significant association (HLA locus) compared to 45 in our final meta-analysis from peripheral blood. Considering these 45 associations are enriched at regulatory and coding regions of genes critical for immune regulation, we suggest that these additional analyses reinforce that the EBV detected in peripheral blood is distinct from the reservoir identified in saliva.

5. Regarding Reviewer Fig. 5. While the computational simulation supports the possibility that the detection levels are consistent with latent EBV at 1 in 10^4 per cell, the counter-argument is that most healthy individuals (~90%) should be detected, while that was not the case. The variation in detection is potentially interesting, but the authors do not consider the longitudinal variation of EBV load, whether from increase latently infected cells or lytic activation in the oropharynx and repopulation of the latent reservoir. This is likely a normal cycle that may be perturbed in some individuals and disease states. While the overall analyses and correlations may hold, the underlying mechanism and interpretation of the data is open to question. The authors should have a better appreciation for the variation in EBV load over the normal course of individual life-time, including age and environment related changes.

We agree about the importance of the potential fluctuations of EBV load due to age and environmental changes. We have now substantially modified our text with additional text for these possibilities in the following lines:

[Page 14, Lines 494-498] Further, as viral DNA levels can fluctuate over time, these WGS measurements represent a snapshot of a complex process that marked individuals with potentially transiently high levels of EBV DNA. Thus, longitudinal profiles would refine our DNAemia phenotyping and power further discovery of genetic loci that underlie population variation in viral latency and persistence.

Otherwise, noting the comment above with additional saliva-derived analyses, our revised manuscript provides a clear distinction between the lack of genetic associations (outside of HLA) with saliva-derived EBV levels, providing stronger evidence that our characterization in blood reflects distinct biology specific to persistence in peripheral blood (particularly B cells).

Finally, we emphasize one last important point. Though our measurement is almost certainly imperfect (i.e., some fraction of individuals with EBV DNAemia from PBMCs is actually undergoing lytic infection, or EBV is derived from another source), our statistically significant associations reflect a lower bound. Conceptually, the noise introduced into our EBV detection will weaken the associations.

We appreciate the suggestions to synthesize these new analyses and text changes that now capture the variation in EBV load across tissues and time.

6. The longitudinal data shown in Reviewer Fig. 7 may further support the sporadic nature of the EBV DNA copy number and transcript detection. While this may not nullify the overall value of the study, it may change the interpretation of the data.

The longitudinal data in **Reviewer Fig. 7** (reproduced above as **Reviewer Fig. 1**) tested the hypothesis that, if individuals with EBV DNAemia were only categorized as positive sporadically (e.g., when experiencing lytic reactivation), we would expect larger fluctuations in these individuals' serological measurements between the two timepoints. However, the figure shows that there were no obvious differences in EBV serology levels for four different EBV antigens across the two timepoints, for both individuals with and without EBV DNAemia. This finding that individuals may experience high viral DNA loads in peripheral blood irrespective of anti-EBV serology levels is in line with prior studies that have used paired longitudinal serology and PCR analyses and came to similar conclusions⁶.

We clarify the seropositivity measurement interpretation in the section **Multi-modal characterization of EBV in peripheral blood**:

[Page 6, Lines 176-185] We analyzed the change in titers against all four EBV antigens, finding no significant difference in longitudinal fold changes of antibody titers when stratified by EBV DNAemia (Extended Data Fig. 3j,k). Additionally, we observed that 0 individuals with EBV DNAemia had seroconversion or seroreversion over longitudinal samples for any of the four antigens. The lack of resolution in antibody isotypes means that seropositivity does not necessarily indicate ongoing or recent reactivation (Methods), and we observed no evidence of variable antibody responses that would result from recent reactivation or infection in this population. In sum, while a proportion of the 735,954 donors in UKB and AoU were likely experiencing lytic infection or reactivation during blood sampling, these analyses suggest that EBV DNA captured by WGS of population biobanks likely reflects latent infection for most individuals.

We have also noted the potential fluctuations of EBV load in the **Discussion**:

[Page 14, Lines 494-498] Further, as viral DNA levels can fluctuate longitudinally, these WGS measurements represent a snapshot of a complex process that marked individuals with potentially transiently high levels of EBV DNA. Thus, longitudinal profiles would refine our DNAemia phenotyping and power further discovery of genetic loci that underlie population variation in viral latency and persistence.

7. Given the new findings shown in Reviewer Fig. 10 that immunosuppressive drugs may contribute to EBV DNAemia, how might that affect the overall conclusions. Rather than the diseases associated with the EBV DNAemia, are the drug treatments for those diseases contributing EBV copy number control?

The interpretation of the analysis in the prior Reviewer Fig. 10 (reproduced below as **Reviewer Fig. 3**) is that annotated treatment with immunosuppressive drugs does not change the interpretation or conclusion of these associations. Explicitly, the comparison is showing two different statistical models. On the X-axis is the model reported in main manuscript **Fig. 2**. On the Y-axis is an additional model with prior immunosuppressive drug use as a covariate (the full set of drugs is reported in **Extended Data Table 2**). As the dots (each dot is an individual ICD-10 code) lie on the y=x line, this demonstrates that the annotation of individuals taking immunosuppressive drugs has a very limited impact on the results or conclusions of these analyses. In other words, the EBV associations with these ICD-10 codes cannot be attributed to drug treatments. Hence, our analyses conclude that the reported associations are robust to the potential EBV DNAemia that follows immunosuppressive drug use.

Reviewer Figure 3 (previous Reviewer Fig. 10). Robustness of PheWAS analyses to drug treatments.

Per-phenotype summary statistics from the manuscript (x-axis) and with adding immunosuppressive drugs as a covariate (y-axis). The left panel shows the $-\log_{10} P$ values; the right panel shows the beta coefficient from the PheWAS regression. Statistical test: logistic regression with relevant covariates documented in the PheWAS methods.

Ultimately, we agree that general states of immune insufficiency, which can be drug-induced but insufficiently captured in our data, could explain some of the EBV associations. Hence, we have further revised our reporting of these phenotypic associations to mitigate any language that could be perceived as causal:

[Page 7, Lines 201-203] *We emphasize that these associations do not prove causality of EBV DNAemia and may instead reflect a general state of immunosuppression, and the large number of associated traits reflects redundancy in ICD-10 mapping.*

[Page 8, Lines 236-238] *While further work is required to refine the role of EBV in these phenotypes, our scalable approach enables systematic associations with a large set of conditions, including rare diseases that require large sample sizes as in UKB and AoU.*

8. The authors conflate latency with the lack of viral transcripts, but this is not correct. Viral latency in LCLs is a type III latency with significant RNA, including very high levels of some non-coding RNAs, such as EBERS. Many of these RNAs are not readily detected in RNAseq due to technical issues, that may include lack of non-polyadenylation or other reasons not yet clear. The authors may want to suggest that the majority of latency is type 0 latency, where no viral RNA is expressed. This is possible in memory B-cells in peripheral blood. Either way, the authors should have a better understanding of EBV latency types.

We appreciate the discussion about the distinct latency states and have significantly revised this section to reflect this. Indeed, the lack of detected EBV transcripts reflects EBV latency 0, and LCLs are latency III. We have updated the discussion of EBV RNA detection in population-scale sequencing datasets (within the section **Multi-modal characterization of EBV in peripheral blood**) to convey the nuance:

[Page 5, Lines 147-149] *In memory B cells, EBV predominantly remains in a transcriptional state (termed latency 0) with a limited viral gene expression program. In contrast, viruses in lytic reactivation express a wide range of viral proteins that enable immune evasion.*

[Page 6, Lines 163-165] *To confirm EBV transcript detection sensitivity with scRNA-seq, we reanalyzed scRNA profiles of LCL, in which EBV expresses a latency III program, and kidney transplant samples assayed with the same sequencing workflow as OneK1K.*

9. Extended Figure 3 and the longitudinal study (4H). How many individuals could be followed for how many time points?

For the longitudinal characterization, there were 277 individuals with longitudinal EBV serology samples provided in UKB, 262 of whom had WGS and serology profiled from the baseline sample, along with a replicate serology measure from a separate blood draw taken 2–6 years following the baseline. There were at most two time points available. We make this point clearer in our revised **Methods** section:

[Page 34, Lines 807-812] *Of the 277 individuals with longitudinal serology, 262 had: a) WGS profiled from the baseline sample (“instance 0” in UKB data field 32056); b) serology assessed at the baseline sample; and c) a replicate serology measure from a separate blood draw taken 2–6 years following the baseline (note: the precise dates were not available). There were at most two time points available for serology measurements, and no longitudinal WGS samples were available in either biobank.*

While the overall study is fine, the authors over-interpret the EBV latency issue. EBV virus is frequently detected in saliva of most individual at regular intervals, indicating virus is chronically and perpetually reactivating in the oropharynx and replenishing the latent reservoir. It remains unclear to me whether some individuals have long-term chronic high loads of EBV in PBMC, or if these loads vary over time with those having very high loads correlating with poor immune control and a vast number of other disease indications?

We appreciate the feedback on EBV latency, and as noted in the points above, we have significantly revised our manuscript, including our title, to avoid any potential mischaracterization of the latency of the virus.

Regarding saliva as a latent reservoir, as noted above, we have now extended our dataset to also include quantification of the EBV DNA load in 48,889 saliva-based WGS samples in AoU, as well as downstream GWASes using various thresholds. We see very distinct abundances (total EBV loads and overall positivity), further supporting our assertion that the ~10% of donors that we annotate with EBV DNAemia reflect a distinct subset of the population where genetic loci in part regulate the persistence of this viral DNA.

For the longitudinal component, we have added text qualifying how our measurement is a snapshot, as noted above. For greater context, prior studies of healthy individuals have tracked EBV DNA loads over months using qPCR, including individuals who experience persistently high loads for months after initial measurements^{6,7}. We rely on the interpretation of these past studies, as the lack of longitudinal data in any biobank limits our ability to answer this question using our WGS approach.

Synthesizing these elements, we would suggest this interpretation: there are indeed some individuals with high persistent viral DNA loads, which are heavily mediated by host genetics. A fraction of individuals at any given time point may indeed have a recent infection or reactivation event, and these individuals do limit the strength of our associations. In saliva, close to the site of infection and productive replication⁸, we observe a ~5x higher incidence of EBV DNAemia but cannot identify genetic loci aside from HLA that mediate this abundance, reflecting that the mechanisms that underlie persistence in peripheral blood either do not apply or have effect sizes that cannot be identified due to the high noise of what is happening in saliva. We hope that these additional analyses, modified discussion, and revised terminology provide an appropriate consideration of the nuance.

Other comments

Line 22. “persist for a lifetime in peripheral blood...” is misleading

We have now revised this line to mitigate potential confusion:

[Page 2, Lines 20-21] Although primary infection typically resolves with subclinical symptoms, long-term complications can arise following immune dysregulation associated with persistent EBV infection.

Line 26. “ biomarker for latent EBV infection.” Is also misleading, as it is a biomarker for EBV viral load, which is a complex sum of both latent and sporadic lytic activity.

We have now revised this line to mitigate potential confusion, including removing the word latent:

[Page 2, Lines 24-26] Here, we demonstrate that existing whole genome sequencing (WGS) data contain ample non-human sequences to reconstruct a molecular feature of persistent EBV DNA consistent with orthogonal phenotypes, including viral serostatus from prior EBV infections.

Line 169. “0 individuals with EBV DNAemia had seroconversion or seroreversion over longitudinal samples.” Understood that serology does not change over time, but it is surprising that the same individuals remain stable DNAemia over longitudinal samples. This should be stated more clearly- that the DNAemia individual remain at the same DNA copy number over time. If this DNAemia is not stable over time for the same individuals, this should be stated more clearly.

We apologize for the confusion. The longitudinal measurements shown are serology measures for these donors. In UKB, EBV DNAemia status is based on the WGS sample at the first visit, which is a single time point. To the best of our knowledge, no longitudinal WGS is available in large biobanks, either considered here or otherwise published. We have made this point clearer in the relevant sections:

*[Page 6, Lines 174-176] Here, 262 individuals, including 17 individuals with EBV DNAemia, had baseline serology profiles and WGS, with additional serology measurements taken 2–6 years following the initial sample collection (**Methods**).*

[Page 34, Lines 807-812] Of the 277 individuals with longitudinal serology, 262 had: a) WGS profiled from the baseline sample (“instance 0” in UKB data field 32056); b) serology assessed at the baseline sample; and c) a replicate serology measure from a separate blood draw taken 2–6 years following the baseline (note: the precise dates were not available). There were at most two time points available for serology measurements, and no longitudinal WGS samples were available in either biobank.

Referee #2 (Remarks on code availability):

I was not able to review the code and not my area of expertise.

Referee #3 (Remarks to the Author):

The authors provided an extremely thorough revision and response to the review, which has substantially improved the paper.

On the question of whether the authors measure “EBV DNAemia” is measuring latent EBV infection, they provide quite a lot of additional analysis, which is encouraging. I agree that the data support the notion that what is being measured is not lytic virus. However, I still think that it is confusing, and inaccurate, to describe it simply as a measure of latency, which implies NO latency in the remaining ~90% of subjects. The authors do state that they believe that they are measuring the tail end of the distribution of latent virus DNA, but at other times again imply that only those that are detected are experiencing actual latency. I think it’s important to clarify that while nearly all of the subjects who are EBV seropositive will have some latent virus, those with measurable EBV DNA are experiencing, for whatever reasons, a higher latent load.

We appreciate the reviewer’s careful consideration of the EBV DNAemia biomarker and agree that the manuscript could improve in clarity. To encompass the multitude of outcomes following EBV primary infection, we have elected to use “persistence” to describe the detected DNA (note: we claim persistent DNA rather than persistent infection) in line with prior definitions^{1,2}, as we believe the term best captures the complexity in harboring this virus³. We note the following changes:

- Our title now reads **Population-scale sequencing resolves correlates and determinants of persistent Epstein-Barr Virus DNA**.
- Our abstract now introduces the molecular feature we are characterizing as persistent EBV DNA, and conclude that existing WGS data can derive the genetic architecture of viral DNA persistence.
- This is echoed in the revised introduction:
[Page 3, Lines 69-70] As modern biobanks perform WGS on peripheral blood rather than LCLs, we posited that EBV DNA reflecting persistence in circulating cells could be captured and quantified in these libraries.
- To emphasize the idea that individuals without EBV DNAemia do not necessarily mean there is *no* detectable EBV DNA in these individuals, we describe individuals with EBV DNAemia more cautiously in the section **Biobank WGS data harbors EBV DNA**:

[Page 5, Lines 117-121] As the proportion of individuals with EBV DNAemia (9.7%) is lower than the rate of seropositivity (~95%) in UKB, we interpret our measure to reflect the tail of individuals with the highest levels of circulating EBV DNA. This approach does not detect every individual who has had previous EBV infection, but identifies those with the greatest EBV DNA burden detectable in peripheral blood at the time of WGS sampling.
- We now constrain the discussion of latency with respect to our data analysis to a single section: **Multi-modal characterization of EBV in peripheral blood**. We deliberately chose this language to be inclusive of alternative interpretations, including spontaneous infection and reactivation.
- We revised the interpretation of EBV DNAemia in the section **Polygenic variation underlies EBV DNAemia**:

[Page 10, Lines 331-334] Namely, serostatus reflects a history of infection, which is largely independent of genetic variation, whereas our EBV DNAemia biomarker identifies a subset of individuals with high levels of persistent EBV DNA at the time of sampling, including those in the absence of lytic reactivation.

- In the **Discussion**, we note that increasing the sequencing depth may help us identify EBV DNA carriers below our current limit of detection. We also note that if there are high levels of EBV DNA present in WGS samples (such as from saliva), we do indeed detect it using our approach:

[Page 14, Lines 480-487] *Despite ~90% EBV seropositivity among adults in the UK and US, we identify a distinct population of 9.7-11.9% of individuals with detectable EBV DNA in peripheral blood, suggesting that past infection is necessary but not sufficient for DNAemia. Instead, simulations imply that the EBV DNAemia population reflects a tail of exposed individuals with the highest EBV DNA levels (Fig. 1e,f; Extended Data Fig. 1d-g). We hypothesize that others in these cohorts are carriers of EBV DNA from past infections, but at levels below our limit of detection. We estimate that sequencing ~100x deeper per whole genome would detect EBV DNA from peripheral blood ~90% of individuals, noting that analyses of WGS from saliva show markedly higher levels of EBV, including DNAemia in 51% of donors.*

- We include the complexity of EBV infection and changes in detectable DNA levels in the discussion:

[Page 14, Lines 494-498] *Further, as viral DNA levels can fluctuate longitudinally, these WGS measurements represent a snapshot of a complex process that marked individuals with potentially transiently high levels of EBV DNA. Thus, longitudinal profiles would refine our DNAemia phenotyping and power further discovery of genetic loci that underlie population variation in viral latency and persistence.*

- We have removed the word “latent” from figures and captions, including in **Extended Data Fig. 1**.

Ultimately, we welcome any specific feedback to best represent the EBV DNAemia biomarker we characterize in the manuscript.

It would be helpful for the authors to double check the databases they downloaded the RNA data from (GTEx, OneK1K) and ensure that the publicly accessible transcript data have not been pre-filtered to exclude non-human sequences.

We appreciate this important suggestion. To verify this, we re-quantified 10 selected samples for explicit reprocessing with the default Cell Ranger mapping for the human reference genome. **Reviewer Fig. 4** shows a representative output, where ~92.4% of reads map to the hg38 reference genome. These results are consistent with our experience mapping PBMCs using the 10x Genomics scRNA-seq chemistry and verify that non-human reads were indeed present before re-mapping.

Mapping ?	
Reads Mapped to Genome	92.4%
Reads Mapped Confidently to Genome	90.6%
Reads Mapped Confidently to Intergenic Regions	2.4%
Reads Mapped Confidently to Intronic Regions	25.1%
Reads Mapped Confidently to Exonic Regions	63.1%
Reads Mapped Confidently to Transcriptome	77.5%
Reads Mapped Antisense to Gene	10.2%

Reviewer Figure 4. Mapping statistics of selected OneK1K run. Shown is the Cell Ranger v8 output for the SRR18029486.

We explicitly clarify this analysis in the revised methods section, with the pertinent text noted below. We appreciate the excellent suggestion to ensure confidence in our negative result.

[Page 34, Lines 796-799] After verifying that non-human reads were provided as part of the upload (i.e., ~90-95% of the reads mapped to the hg38 reference), we reanalyzed a total of 53,872,337,003 paired-end sequencing reads from this consortium, identifying only 1 UMI that was classified as uniquely mapping to the EBV transcriptome (the BARF0 EBV gene).

Similarly, they should double check that the version of the hg38 reference sequence used to make transcript alignment .bam files does include the chrEBV. If they have already done this, it would be easy to add a statement to the manuscript. Line 757-759 could be clarified to explicitly address this in their methods: “In GTEx, we obtained the mapped alignment files (.bam) from both WGS and RNA-seq of PBMCs for 681 donors, in which both were profiled for each individual. We downloaded .bam files from Anvil after obtaining permissions through dbGaP and subsequently extracted chrEBV for all sequencing datasets.”

This is indeed the case, and we have revised the noted lines to reflect this:

[Page 33, Lines 778-782] For reanalysis of EBV reads from GTEx, we obtained the mapped alignment files (.bam) from both WGS and RNA-seq of PBMCs for 681 donors, in which both were profiled for each individual. The GTEx consortium workflow hosts reads aligned to the hg38 reference, which included chrEBV (verified to be in all libraries). We downloaded .bam files from Anvil after obtaining permissions through dbGaP and subsequently extracted any reads mapping to the chrEBV contig for all sequencing libraries.

The manuscript is not completely clear in the use of EBV DNA+, EBV DNAemia and EBV UMIs (which has now been introduced). Perhaps minor, but they seem to be used interchangeably? Some clarification here would be helpful.

“Using this threshold, we classified 47,452 (9.7%) individuals as EBV DNA+ for subsequent analyses (Extended Data Fig. 1c). Though the proportion of individuals with EBV DNAemia (9.7%) is lower than the rate of seropositivity, we interpret our measure to reflect the tail of individuals for whom persistent EBV DNA is highest”

We have removed any terminology of “EBV DNA+” from the revised manuscript. EBV DNAemia now refers to individuals with detectable EBV DNA above our determined threshold of 1.2 EBV genomes per 10,000 cells. We define this more clearly:

[Pages 4-5, Lines 116-117] We classified 47,452 (9.7%) individuals with EBV DNAemia (defined as detectable EBV DNA levels >1.2 genomes per 10⁴ cells) for subsequent analyses (Extended Data Fig. 1c).

On the other hand, EBV UMIs refer to individual EBV RNA molecules from the analyzed 10x scRNA-seq datasets. We have added this in the relevant section:

[Page 6, Lines 165-168] In both RNA-seq datasets, we detected 100s-1,000s of EBV unique molecular identifiers (UMIs; indicating individual EBV RNA molecules), including rare cells with high transcriptional activity (consistent with HHV-6; Extended Data Fig. 3f,g).

I appreciate the reviewers examining the question of whether increased numbers of B cells could be responsible for the increased EBV DNA load. From their additional analysis, it seems, yes. The fact that this doesn't appear to be due to active infection seems to me beside the point – increased numbers of B cells may simply be indicative of an inflammatory response. While I appreciate the authors' reluctance to include these data with no replication set, I think that it provides important context and could be mentioned and included in supplemental data.

We agree that the analysis may be of broader interest and have now added this to the revised **Extended Data Fig. 3** (reproduced as **Reviewer Fig. 5**), noting that panels **d** and **e** have been added.

Reviewer Figure 5 (Extended Data Figure 3). Characterization of EBV latency in large genomics datasets. (a) Schematic for the GTEEx consortium, highlighting donors where matching WGS and RNA-seq data are available from peripheral blood. **(b)** Sum of per-base read coverage of high-confidence EBV-mapping reads. Two repetitive regions with inflated coverage are noted in red and purple. **(c)** Characterization of EBV detection from 681 GTEEx donors with paired WGS and RNA-seq from peripheral blood. Three RNA-seq samples were positive, all with one paired-end (PE) read. **(d)** Estimated B cell fraction using RNA-seq deconvolution and cell type abundance estimation. Statistical test: Mann Whitney U Tests with two-sided hypothesis testing for $n = 681$ GTEEx donors. **(e)** B cell activation (Panther Pathway) module score per donor. Statistical test: Mann Whitney U Tests with two-sided hypothesis testing for $n = 681$ GTEEx donors. **(f)** Schematic of data generation from the OneK1K cohort. **(g)** Summary of EBV quantification from the OneK1K cohort, showing total read number from the consortium (top) and number of EBV-assigned UMIs. **(h)** Summary of scRNA-seq results from a lymphoblastic cell line (LCL), a positive control for EBV transformation (SAMN34277123). **(i)** Summary of scRNA-seq results from a kidney transplant, a positive control for EBV reactivation (SAMN35232564). **(j)** Depiction of longitudinal serology measures of four EBV antigens for an exemplar EBV DNA+ donor. **(k)** Characterization of EBV longitudinal antigen titers across UKB, stratified by EBV DNAemia. Statistical test: two-sided Student's t -test.

We have also added the following text in the Main Text and Methods:

[Page 6, Lines 156-159] Paired analyses of EBV DNAemia and RNA-seq profiles showed elevated estimated B cell proportions and lower B cell activation states for individuals with EBV DNAemia, reflecting that peripheral blood cell composition may mediate viral DNA persistence (Extended Data Fig. 3d,e; Methods).

[Page 34, Lines 788-791] Further analyses of the GTEx RNA-seq data included an estimation of cell type composition using CIBERSORT, and the B cell activation module was estimated using the AddModuleScore from Seurat of the B cell activation genes in the PANTHER database.

I appreciate that the authors performed a full regression on the imputed HLA data. However, it's unfortunate that these data are not really discussed or highlighted here, given the clear importance of these variants on the trait in question. It's very difficult to even discern what the strongest associations are here; the extended data table where they are presented is not even sorted in any obvious way, not by p-value or even by allele name. Also, while not major, the allele names are not shown correctly anywhere that they are given, where we should be seeing two colon-delimited fields at this resolution.

We agree that the supplementary table of the HLA regression results (Extended Data Table 6) could be clearer. We performed the HLA associations to the allele name level (e.g., "HLA-DPB1*81:01"), for both UKB and AoU. To this end, we have separated Class I and II alleles, added the specific allele names as an additional column, and sorted the allele names by alphabetical order in the updated **Extended Data Table 6**. The values are now sorted alphabetically based on the overall name (column B). **Reviewer Fig. 6** shows the new appearance of the table, which we hope will be acceptable for rapid interpretation of these associations.

A	B	C	D	E	F	G	H	I	J	K
allele	HLA_allele	class	Estimate_UKB	StdErr_UKB	PValue_UKB	ZStatistic_UKB	Estimate_AoU	StdErr_AoU	PValue_AoU	ZStatistic_AoU
A_0101	HLA-A*01:01	Class_I	0.160	0.084	0.056	1.913	0.405	0.054	0.000	7.491
A_0102	HLA-A*01:02	Class_I	0.192	0.245	0.432	0.785	0.738	0.323	0.023	2.282
A_0103	HLA-A*01:03	Class_I	0.340	0.815	0.676	0.418	1.010	0.472	0.032	2.141
A_0201	HLA-A*02:01	Class_I	-0.241	0.084	0.004	-2.873	0.013	0.054	0.816	0.233
A_0202	HLA-A*02:02	Class_I	-0.099	0.179	0.580	-0.554	0.335	0.220	0.128	1.521
A_0203	HLA-A*02:03	Class_I	-0.245	0.154	0.111	-1.592	-10.479	238.035	0.965	-0.044
A_0205	HLA-A*02:05	Class_I	0.026	0.095	0.786	0.271	0.243	0.092	0.008	2.643
A_0206	HLA-A*02:06	Class_I	-0.148	0.131	0.259	-1.129	0.217	0.180	0.228	1.204
A_0207	HLA-A*02:07	Class_I	0.324	0.589	0.582	0.550	-0.545	1.214	0.653	-0.449
A_0210	HLA-A*02:10	Class_I	-29.922	842.759	0.972	-0.036	-0.891	0.589	0.131	-1.512
A_0211	HLA-A*02:11	Class_I	-0.064	0.167	0.702	-0.383	0.195	0.403	0.629	0.484
A_0214	HLA-A*02:14	Class_I	-17.391	733.626	0.981	-0.024	1.062	1.132	0.348	0.938
A_0216	HLA-A*02:16	Class_I	-12.191	595.420	0.984	-0.020	-11.734	200.061	0.953	-0.059
A_0220	HLA-A*02:20	Class_I	-0.308	0.467	0.509	-0.660	-0.320	0.524	0.541	-0.612
A_0264	HLA-A*02:64	Class_I	-26.907	1072.703	0.980	-0.025	-0.925	1.024	0.366	-0.904
A_0301	HLA-A*03:01	Class_I	0.230	0.084	0.006	2.749	0.535	0.052	0.000	10.270
A_0302	HLA-A*03:02	Class_I	0.339	0.132	0.010	2.577	0.487	0.130	0.000	3.755
A_1101	HLA-A*11:01	Class_I	0.183	0.085	0.030	2.169	0.439	0.057	0.000	7.672
A_1102	HLA-A*11:02	Class_I	-0.074	0.803	0.927	-0.092	-10.926	252.328	0.965	-0.043
A_2301	HLA-A*23:01	Class_I	-0.005	0.089	0.958	-0.053	0.275	0.075	0.000	3.668
A_2402	HLA-A*24:02	Class_I	0.087	0.085	0.305	1.026	0.337	0.055	0.000	6.165
A_2409	HLA-A*24:09	Class_I	0.169	0.185	0.186	0.687	0.688	0.149	0.000	1.888

Reviewer Figure 6. Screenshot of Extended Data Table 6. Revisions showing the appropriate nomenclature for all HLA alleles are explicitly shown.

We have revised our manuscript, including the naming and explicit citation of specific alleles, where these analyses are introduced to clarify this result:

[Page 12, Lines 388-395] One of the strongest risk alleles for EBV DNAemia was HLA-A*03:01 (UKB: $Z = 2.6$, $P = 0.0060$; AoU: $Z = 10.3$; $P = 9.63 \times 10^{-25}$), an allele linked to increased risk of MS. Conversely, a protective allele against EBV DNAemia, HLA-DRB1*12:01 (UKB: $Z = -8.7$, $P = 4.6 \times 10^{-18}$; AoU: $Z = -3.6$; $P = 3.9 \times 10^{-4}$), has been associated with less severe MS. We also observed that two other negatively associated HLA alleles, such as HLA-B*35:01 (UKB: $Z = -11.1$, $P = 1.3 \times 10^{-28}$; AoU: $Z = -8.7$; $P = 2.6 \times 10^{-18}$) and HLA-B*55:01 (UKB: $Z = -11.2$, $P = 7.3 \times 10^{-29}$; AoU: $Z = -7.61$; $P = 2.8 \times 10^{-14}$) that present known immunodominant epitopes from the EBV proteome, reflecting that strong peptide presentation may underlie decreased EBV DNAemia.

Last, it seems the authors do not further interpret the ~600 exWAS loci that mapped to the MHC, many of which occur in non-HLA genes. It looks like they chose to just leave it open ended with their statement: "Specifically, as the MHC locus is a hyper-polymorphic region with strong linkage disequilibrium,

ascertaining causal variants at this locus is particularly challenging. Hence, we first focused our efforts on interpreting associated loci outside the MHC locus.” This is a fair compromise, but it is a bit disappointing.

We appreciate the emphasis on further characterization of potentially causal variants in the MHC locus, as this additional analysis was indeed informative. In our revised manuscript, we now report both a global and anecdotal analysis that we believe will be of interest and refine our understanding of the chromosome 6 locus. Specifically, we applied AlphaMissense⁹, which assigns predicted pathogenicity, either benign, ambiguous, or pathogenic, based on amino acid changes from each protein-altering variant. The results are shown in **Reviewer Figure 7**:

Reviewer Figure 7 (manuscript Extended Data Fig. 5). Characterization of non-HLA, chromosome 6 associated ExWAS variants. (c) AlphaMissense characterization of significant missense variants implicated in the ExWAS association analyses. Left: AlphaMissense score and annotation; right: genomic positions of ExWAS variants. **(d)** Zoom plot of the *HFE* locus in the EBV DNAemia GWAS with the UKB NFE cohort, highlighting the rs1800562 variant.

The variant with the highest AlphaMissense pathogenicity score and OR > 1 was *HFE* C282Y (rs1800562), which has been previously implicated as a causal variant in hemochromatosis, an iron storage disorder. We contextualize this anecdote by linking the previously described role of iron metabolism in EBV infection¹⁰. We selected this anecdote to highlight that non-HLA, chr6 associations may indeed be functional, but a full table of the variants shown in **Extended Data Fig. 5c** is now provided as part of our revised **Extended Data Table 4**. We note any variants with missing annotations are a function of altered ref/alt from our database to AlphaMissense, either the protein or nucleotide.

[Pages 8-9; Lines 271-275] Annotation of ExWAS-associated variants with AlphaMissense helped refine candidate causal variants on chromosome 6 outside of HLA (**Extended Data Fig. 5c; Extended Data Table 4**). Plausible functional variants included a missense mutation in *HFE* (rs1800562; C282Y; OR = 1.1; P = 2.16 x 10⁻¹³; **Extended Data Fig. 5d**) linked to iron storage disorders, consistent with prior reports of EBV modulating iron metabolism.

We anticipate that the additional characterization of this variant highlights the value in further resolving potential functions of non-HLA ExWAS loci mapped to the MHC locus. We appreciate the emphasis on this point, as it has made interpreting these genetic association results much clearer.

Referee #3 (Remarks on code availability):

The authors addressed all computational/reproducibility concerns. The inclusion of a Terra workflow that generates EBV DNA counts for 1000GenomesProject individuals was an excellent addition and significantly increases usability.

We appreciate the time spent ensuring our computational resources could be reproduced and made usable.

Referee #4 (Remarks to the Author):

I co-reviewed this manuscript with one of the reviewers who provided the listed reports.

Referee #5 (Remarks to the Author):

Nyeo et al have revised their manuscript to include additional analyses and rewrite sections that lacked explanations and context. I am satisfied with the revised paper, as the authors have done great work in addressing my (and other reviewers') comments, resulting in a much stronger manuscript.

We appreciate the strong support of our manuscript and the careful reading of our revision.

Some minor typos:

Line 155: "same" presumably missing from "the same sequencing workflow"

This is indeed the case and has been updated.

Line 709: "with a 10% forced 0 rate for 500,00" : should be 50K ie mistake in placement of comma.

This has been corrected.

Referee #6 (Remarks to the Author):

I co-reviewed this manuscript with one of the reviewers who provided the listed reports.

REFERENCES

1. Kane, M. & Golovkina, T. Common threads in persistent viral infections. *J. Virol.* **84**, 4116–4123 (2010).
2. La Frazia, S., Pauciullo, S., Zulian, V. & Garbuglia, A. R. Viral oncogenesis: Synergistic role of genome integration and persistence. *Viruses* **16**, 1965 (2024).
3. Boldogh, I., Albrecht, T. & Porter, D. D. Persistent viral infections. in *Medical Microbiology* (University of Texas Medical Branch at Galveston, Galveston (TX), 1996).
4. Bao, E. L. *et al.* Inherited myeloproliferative neoplasm risk affects haematopoietic stem cells. *Nature* **586**, 769–775 (2020).
5. Bernstein, N. *et al.* Analysis of somatic mutations in whole blood from 200,618 individuals identifies pervasive positive selection and novel drivers of clonal hematopoiesis. *Nat. Genet.* **56**, 1147–1155 (2024).
6. Maurmann, S. *et al.* Molecular parameters for precise diagnosis of asymptomatic Epstein-Barr virus reactivation in healthy carriers. *J. Clin. Microbiol.* **41**, 5419–5428 (2003).
7. Conacher, M. *et al.* Epstein-Barr virus can establish infection in the absence of a classical memory B-cell population. *J. Virol.* **79**, 11128–11134 (2005).
8. Hayman, I. R. *et al.* New insight into Epstein-Barr virus infection using models of stratified epithelium. *PLoS Pathog.* **19**, e1011040 (2023).
9. Cheng, J. *et al.* Accurate proteome-wide missense variant effect prediction with AlphaMissense. *Science* **381**, eadg7492 (2023).
10. Zhao, X. *et al.* Epstein-Barr virus modulates iron metabolism and ferritin expression to promote tumorigenesis in gastric cancer. *J. Mol. Histol.* **56**, 231 (2025).

Please respond to these point-by-point:

1. Please reduce the length of the title to 75 characters (with spaces) or fewer, so that it fits on two lines in the final layout.

This has been corrected. Our new title reads "Population-scale sequencing resolves determinants of persistent EBV DNA".

2. Please reduce the Abstract to ~230 words. Currently there are 278 words.

We have revised our Abstract to be 227 words.

3. For a regular research article, we recommend ~60 main text references.

We have now revised the main text to contain 59 references as suggested.

4. Please split the main text and methods references into two separate, but continuously numbered, lists that follow the discussion and the methods section, respectively.

These have now been split as recommended.

5. Please re-supply the Extended data figures and tables individually in EPS, JPEG or TIF format.

We have supplied these as high-resolution JPEG files.

6. Please ensure that the text size in all figures is at least 5 pt Arial, keeping in mind that the maximum size of a main text figure is 18cm width by 17cm height.

We have modified relevant figures accordingly.

7. Please reduce subheadings to 40 characters (with spaces) or fewer.

This has been corrected.

8. You have more than one account on the manuscript system. Please contact Nature Manuscripts (naturemanuscripts@nature.com) to confirm all accounts which belong to you, and your primary email address.

We appreciate the note and confirm that slav.petrovski@astrazeneca.com is the primary email address.

9. Please provide a supplementary information guide as a separate word document.

This has been supplied as part of the upload.

10. There are potential third party rights issues in the figures. Please check the sources of all illustrations and clarify whether permissions are needed to adapt or reproduce them. Please make sure to include the relevant details in third party rights table when you resubmit (more information below). If Biorender or a similar software has been used, please also ensure to provide relevant licenses. In particular please check figure/s: Figures 1a, 5(d,f) Extended data figures 1h, 2a, 3(a,f), 8e, and SI Figure 1a.

11. Please make sure to provide a third party rights table (more information below) when you resubmit. If Biorender or similar software has been used, please also ensure to provide relevant licenses.

For points 10 and 11, we confirm that our figures comply with the policy, including the noted subpanels.

12. The Chief Editor has approved 5,000 words for your paper. As you are aware, the current manuscript exceeds this limit by quite some margin. We ask that you make the necessary revisions to bring the article to budget. We recommend that you aim for more economical writing and to move more technical and/or peripheral descriptions and discussions of results to Supplementary Notes.

We have revised our Main text to be 4,997 words as suggested.

13. Please label the 8 “Extended Data Tables” as “Supplementary Tables” instead and change the call-out to these tables accordingly throughout.

This has been corrected.

13. For any Supplementary Figures, please check and confirm that:

- * If data is presented as bar charts, individual data points are shown using overlaid dot plots.
- * The n number (i.e. the sample size used to derive statistics) is provided and defined as a precise value (not a range), using the wording “n=X samples/cells/independent experiments” etc. where applicable.
- * Any chart axis, error bars, scale bars, symbols and colour scales are defined.
- * Any statistical tests used for data analysis are specified and exact p-values are provided either on the figures themselves, in the legend or in the Source Data file.
- * Wherever representative data such as micrographs are shown, the legend indicates how many times the experiment was repeated with the same results.

We have revised our figures as necessary, including showing exact *P* values and clarifying legends where noted.

Referees' comments:

Referee #2 (Remarks to the Author):

The authors have provided an excellent revision and comprehensive rebuttal. The revised manuscript is substantially improved and provides a compelling correlative analyses of high EBV viral load in PBMC with various inflammatory disease and gene loci. The study is of great interest to the EBV community, and others studying inflammatory disease and GWAS studies of large databases.

We appreciate the support of our revised manuscript and the very helpful feedback ensuring the accuracy of our terminology and results.

One minor comment:

Lines 143-145.

In memory B cells, EBV predominantly remains in a transcriptional state (termed latency 0) with a limited viral gene expression program¹¹. In contrast, viruses in lytic reactivation express a wide range of viral proteins that enable immune evasion¹¹.

While EBV type 0 latency may be one common state for EBV in some resting memory B-cells, other more sensitive methods find some viral transcripts in memory B-cell subpopulations in healthy donors. The failure to detect EBV transcripts in RNAseq datasets may be due to some technical issues relating to the failure to capture low level EBV transcripts in RNAseq libraries. The point being that the lack of EBV reads in RNAseq datasets does not necessarily indicate a type 0 latency. The authors should be cautious to not over-interpret the EBV latency transcriptional state, and concluding type 0 latency is not fully justified based on the existing data.

We appreciate the reviewer's attentiveness to detail and have now included this nuance in our interpretation, as well as a very recent publication with a sensitive methodology that specifically measured EBV transcription in peripheral blood B cells. In the "Multimodal characterization of EBV in peripheral blood" section (now in **Supplemental Note 1**), it now reads:

These findings indicate that EBV DNA presence in peripheral blood is overwhelmingly not accompanied by detectable viral RNA expression, supporting that the viral DNA detected in UKB and AoU likely reflects latent carriage rather than active replication.

We emphasize that recent sensitive methods specific for EBV have detected very low amounts of EBV transcripts in peripheral blood B cells of healthy donors⁸. The failure to detect EBV transcripts in the analyzed RNA-seq datasets may reflect that previous methods are not sensitive enough to capture low levels of EBV transcripts, if present, and further work is needed to fully determine the latency state of the virus. We note, however, that in the same paper, the 10 healthy controls had 0-3 EBV-infected B cells detected per 10,000 cells, and no subpopulation of B cells with EBV lytic reactivation gene expression was detected⁸, corroborating our observations of largely latent EBV in the blood of healthy individuals.

Referee #3 (Remarks to the Author):

The authors have thoroughly responded to all critiques, and the manuscript is substantially improved. I have no further comments and commend the authors for their responsiveness to all reviews and for this interesting work.

We appreciate the helpful feedback and confirmation of the value of our manuscript revisions.

Referee #3 (Remarks on code availability):

Code was previously reviewed and found to be adequate

Referee #4 (Remarks to the Author):

I co-reviewed this manuscript with one of the reviewers who provided the listed reports.

We are grateful for the time spent verifying our software is applicable for a broad audience.